# eddy4R 0.2.0: A DevOps model for community-extensible processing and analysis of eddy-covariance data based on R, Git, Docker and HDF5

**Stefan Metzger[1,2], David Durden[1], Cove Sturtevant[1], Hongyan Luo[1], Natchaya Pingintha-Durden[1], Torsten Sachs[3], Andrei Serafimovich[3], Jörg Hartmann[4], Jiahong Li[5], Ke Xu[2], Ankur R. Desai[2]**

[1]National Ecological Observatory Network, Battelle, 1685 38[th] Street, Boulder, CO 80301, USA

[2]University of Wisconsin-Madison, Dept. of Atmospheric and Oceanic Sciences, 1225 West Dayton Street, Madison, WI 53706, USA

[3]GFZ German Research Centre for Geosciences, Telegrafenberg, 14473 Potsdam, Germany

[4]Alfred Wegener Institute – Helmholtz Centre for Polar and Marine Research, Am Handelshafen 12, 27570 Bremerhaven, Germany

[5]LI-COR Biosciences, 4647 Superior Street, Lincoln, NE 68504, USA

Correspondence to: Stefan Metzger (smetzger@battelleecology.org)

**Keywords**: computing, container, continuous development, continuous integration, devOps, eddy4R, eddy-covariance, EddyPro, EdiRe, EddyUH, image, NEON, REddyProc, reproducibility, science code, TK3

**Abstract**

Large differences in instrumentation, site setup, data format, and operating system stymie the
adoption of a universal computational environment for processing and analyzing eddy-
covariance (EC) data. This results in limited software applicability and extensibility in addition
to often substantial inconsistencies in flux estimates. Addressing these concerns, this paper
presents the systematic development of portable, reproducible, and extensible EC software
achieved by adopting a Development and Systems Operation (DevOps) approach. This
software development model is used for the creation of the eddy4R family of EC code packages
in the open-source R Language for Statistical Computing. These packages are community-
developed, iterated via the Git distributed version control system, and wrapped into a portable
and reproducible Docker filesystem that is independent of the underlying host operating
system. The HDF5 hierarchical data format then provides a streamlined mechanism for
highly compressed and fully self-documented data ingest and output.
The usefulness of the DevOps approach was evaluated for three test applications. First, the
resultant EC processing software was used to analyze standard flux tower data from the first
EC instruments installed at a National Ecological Observatory (NEON) field site. Second,
through an aircraft test application we demonstrate the modular extensibility of eddy4R to
analyze EC data from other platforms. Third, an intercomparison with commercial-grade
software showed excellent agreement ($R^2$=1.0 for $CO_2$ flux). In conjunction with this study,
a Docker image containing the first two eddy4R packages and an executable example
workflow, as well as first NEON EC data products are released publicly. We conclude by
describing the work remaining to arrive at the automated generation of science-grade EC fluxes,
and benefits to the science community at large.
This software development model is applicable beyond EC, and more generally builds the
capacity to deploy complex algorithms developed by scientists in an efficient and scalable
manner. In addition, modularity permits meeting project milestones while retaining
extensibility with time.

# 1   Introduction

Answering grand challenges in earth system science and ecology requires combining information from hierarchies of environmental observations (tower, aircraft, satellite; Raupach et al., 2005; Running et al., 1999; Turner et al., 2004). Eddy-covariance (EC) measurements serve as crucial observations in this hierarchy to study landscape-scale surface-atmosphere exchange processes that both inform and anchor earth system models. Networks of EC towers such as FLUXNET (Baldocchi et al., 2001), AmeriFlux (Law, 2007), ICOS (Sulkava et al., 2011), and others are vital for providing the necessary distributed observations covering the climate space, with the longest running towers now reaching two decades of observations.

A current challenge for EC tower networks in informing regional and continental scale processes is instrument and computational compatibility. The computations involved in EC processing are complex and developmentally dynamic, making code portability, extensibility, and documentation paramount. Much progress has been made in developing community standards for processing algorithms and workflows (Aubinet et al., 2012; Papale et al., 2006). Many authors have included code in publication, or have developed sharable tools (e.g. EddyPro and TK3 by Fratini and Mauder (2014), EddyUH by Mammarella et al. (2016), EdiRe by Clement et al. (2009), despite the significant and often unfunded effort required to adequately document and generalize code. Still, large differences in instrumentation, site setup, data format, and operating systems stymie the adoption of a universal EC processing environment: one that is portable, reproducible, and extensible to allow tailored workflows that incorporate additional data streams, to automate and scale processing across large compute facilities, or to inject additional algorithms that address specific needs or synergistic research questions. In 50% of published scientific code, one cannot even replicate the necessary software dependencies (Collberg et al., 2014), and even widely used and well-documented EC processing software packages have shown substantial inconsistencies in flux estimates (e.g. Fratini and Mauder, 2014). A universal EC processing environment that enables these capabilities would better allow research groups to tailor existing software to their needs (and contribute new algorithms) instead of re-creating code or kludging together multiple software outputs to realize an algorithmic chain for their data analytics.

The U.S.-based National Ecological Observatory Network (NEON), once fully operational, will represent the largest single-provider EC tower network globally, with a standardized measurement suite designed explicitly for cross-site comparability and analysis of continental-scale ecological change (Schimel et al., 2007). This capability is accompanied by a strong need for a flexible and scalable processing framework that can incorporate specific data streams, take advantage of close alignment of hardware and software for problem tracking and resolution, provide traceability and reproducibility of outputs, and seamlessly integrate distributed and dynamic community-developed code (written by multiple people in multiple places) within existing cyberinfrastructure (CI). In sum, NEON needs what the EC community is currently lacking.

The question we ask in this paper is: How do we collaboratively create portable, reproducible, open-source, scalable, and extensible software that improves reliability and comparability of

EC data products? Here, we describe and demonstrate a developmental model that enables these capabilities by embracing a Development and Systems Operation (DevOps) approach. DevOps is a philosophy arising from the software development community that emphasizes collaboration among developers and operators to continuously iterate the development, building, testing, packaging, and release of software (Erich et al., 2014; Loukides, 2012). Tools are adopted that control and automate these processes, allowing distributed development and rapid iteration. Applied to the scientific community, developers are the multitude of scientists creating and improving the scientific algorithms that form the developmentally dynamic community standard. Operators are those deploying the algorithms to process and analyze data, and can be the same or different people as those creating the algorithms. A key aspect of DevOps is the recipe- or script-based generation and packaging of computation environments rather than abstracted documentation, which improves accessibility, extensibility, and reproducibility of scientific software (Boettiger, 2015; Clark et al., 2014). The recipe automates the loading of the software including all dependencies so that the most significant hurdle of reproducing the computational environment is overcome. At the same time, the recipe serves as explicit documentation, and can be easily extended (added to or changed), shared, and versioned. The entire computational environment including any necessary data are packaged into Docker images that work identically across different computers and operating systems, can be deployed at scale, and archived for ultimate reproducibility.

In the following we present this framework and demonstrate its success in producing EC data products via a family of modular, open-source R packages wrapped in Docker images. We emphasize that this paper is not a presentation of EC processing software (although this is the ultimate application). Rather, it is a presentation of the development model that facilitates portability, reproducibility, and extensibility of EC processing software. In the following, Sect. 2 describes the DevOps framework, and Sect. 3 provides three core tests of the applicability of this framework: 1) processing tower-based flux data, including NEON's first set of EC data, 2) processing and footprint modeling of aircraft-based flux data, and 3) a software cross-validation. Section 4 summarizes the work remaining to operationally produce EC fluxes from 47 NEON sites, and provides an outlook on future capabilities and science community benefits. Code and data availability information is provided in Sect. 5.

## 2   The development and operations (DevOps) model

DevOps promotes collaboration and tight integration between software development, testing, and operational deployment by following a core workflow (e.g., Wurster et al., 2015): Plan, Create, Verify, Package, Release, Configure, and Monitor. The text below describes these stages and **Error! Reference source not found.** shows the general sequence and overlap of these stages between software developers (Dev) and operators (Ops).

**Plan** involves focusing and prioritizing new software features or capabilities based on their enhancement of value. **Create** is the activity of designing and writing the code that delivers a new feature. **Verify** tests the new software feature against established standards for accuracy and performance (e.g. does it unexpectedly alter the output of pre-existing features? Does it produce the expected result?). **Package** involves the compilation of the code once it is ready

for deployment, including all data and software dependencies, and gathers necessary approvals.
The **Release** stage deploys the software into production. **Configure** involves supplying and
configuring the computational infrastructure required to operate the code at scale, including
storage, database operations, and networking. Finally, **Monitor** observes and tracks the use,
performance, and end-user impact of the release.

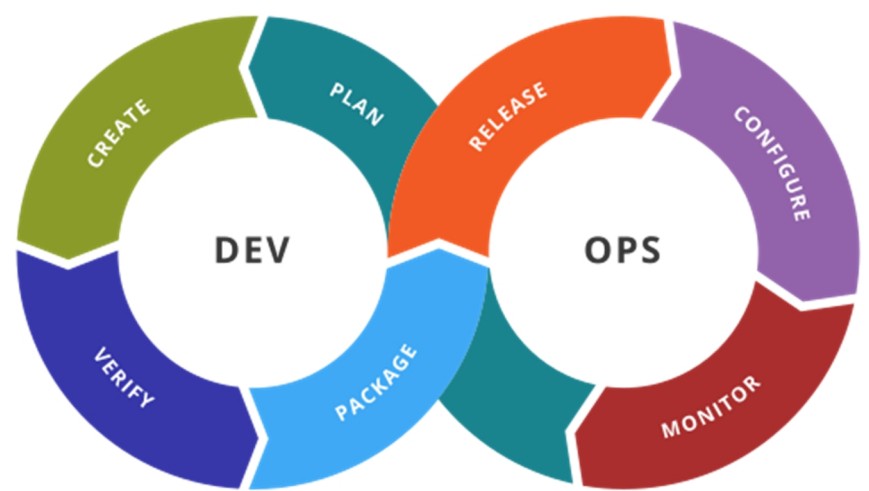


Figure 1. Stages of the general DevOps workflow (source: Kharnagy via Wikimedia
Commons [CC BY-SA 4.0]).

Variants of this workflow exist (e.g., Chen, 2015), but the general components and sequence
are retained. In addition, there is no single set of tools accompanying the DevOps approach.
Rather, many tools exist that facilitate the execution of one or more of these workflow steps,
often through automation.
NEON's DevOps framework consists of a periodic sequence (Figure 2) that incorporates these
workflow steps. For this purpose we define NEON Science as personnel working directly on
the NEON project, and the Science Community, regardless of whether they also work on the
NEON project, as anyone producing or using data, algorithms, or research products related to
the NEON data themes (Atmosphere; Biogeochemistry; Ecohydrology; Land Cover and
Processes; Organisms, Populations, and Communities): The science community contributes
algorithms and best practices (1). Implicitly or explicitly, this embodies the DevOps: Plan stage
– the algorithms most valued by the community are being incorporated. Together with NEON
Science (2), these algorithms are coded in the open-source R computational environment
(DevOps: Create stage). DevOps: Verify (testing) and Package (packaging) are performed as
the code is compiled into eddy4R packages via the GitHub distributed version control system
(3). NEON Science releases an eddy4R version from GitHub, which automatically builds an
eddy4R-Docker image on DockerHub as specified in a "Dockerfile" (4; DevOps: Release stage).
The eddy4R-Docker image is immediately available for deployment by NEON CI (5; DevOps:
Configure & Monitor stages), the Science Community (1) and NEON Science (2) alike. Here
the DevOps: Configure (computational resource allocation) & Monitor stages occur.
Monitoring of end-user experience is also performed in GitHub (3) via issue-tracking. This
DevOps cycle can be repeated for continuous development and integration of requests and
future methodological improvements by the scientific community, resulting in the next release.
Two principal types of releases are provided: stable versions are tagged with "0.2.0", "0.2.1"
etc., and the most recent development built is tagged with "latest". Thus, the DevOps model
serves as the framework within which the scientific community can efficiently and robustly
collaborate to produce, manage, and iterate software. Through choosing appropriate tools to
implement the DevOps workflow steps, the reproducibility, scalability and extensibility needs
of software development communities (including EC) can be met.

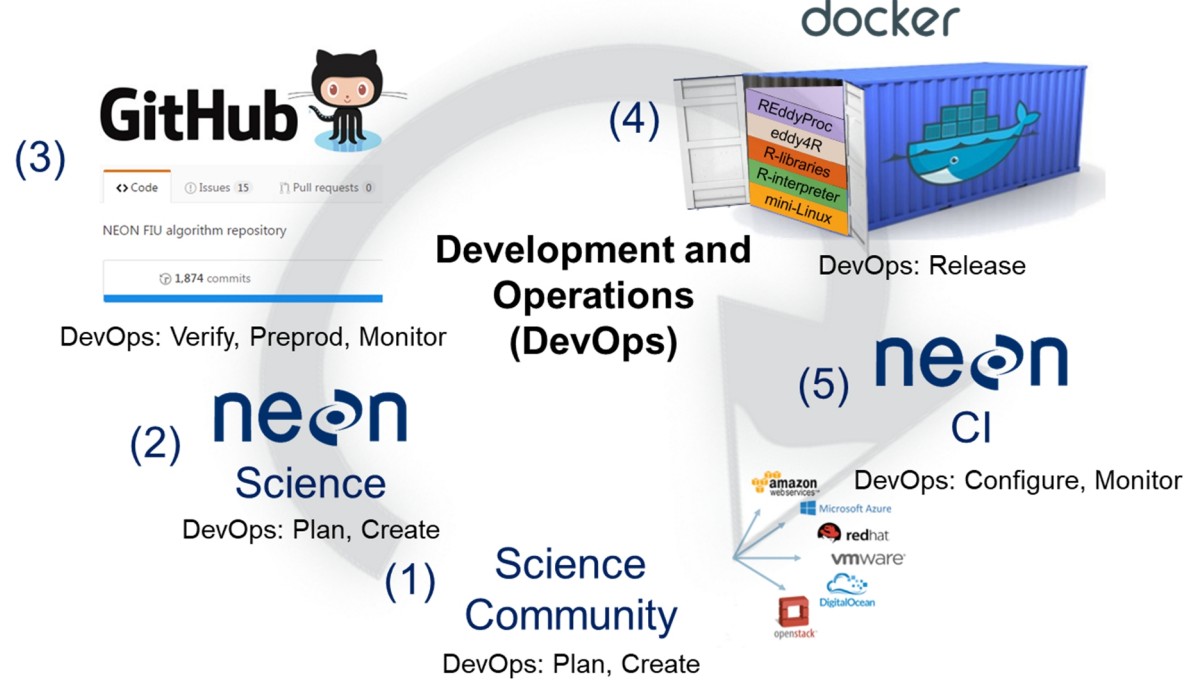

Figure 2. NEON-specific DevOps workflow. DevOps workflow steps are called out in
parentheses. Please see text in Sect. 2 for detailed explanation.

In the following we describe the key components and tools of this NEON-specific DevOps
model, namely the eddy4R family of code packages (Sect. 2.1), Git-based distributed code
development (Sect. 2.2), packaging of the computational environment in Docker images
(Sect. 2.3), hierarchical data formats (Sect. 2.4), integration with NEON's CI (Sect. 2.5), and
installation and deployment (Sect. 2.6).
## 2.1 The eddy4R family of R-packages (DevOps: Plan & Create)
eddy4R is a family of open-source packages for EC raw data processing, analyses and modeling
in the R Language for Statistical Computing (R Core Team, 2016). Forming the DevOps: Plan
& Create stages, it is being developed by NEON scientists with wide input from the
micrometeorological community (e.g., De Roo et al., 2014; Kohnert et al., 2015; Lee et al.,
2015; Metzger et al., 2012; Metzger et al., 2013; Metzger et al., 2016; Sachs et al., 2014; Salmon
et al., 2015; Serafimovich et al., 2013; Starkenburg et al., 2016; Vaughan et al., 2015; Xu et al.,
2017). eddy4R currently consists of four packages eddy4R.base, eddy4R.qaqc, eddy4R.turb,
and eddy4R.erf. Of these, eddy4R.base and eddy4R.qaqc are published here in conjunction with
NEON's release of EC Level 1 data products ([https://w3id.org/smetzger/Metzger-et-](https://w3id.org/smetzger/Metzger-et-al_2017_eddy4R-Docker/portal/0.2.0)
[al_2017_eddy4R-Docker/portal/0.2.0](https://w3id.org/smetzger/Metzger-et-al_2017_eddy4R-Docker/portal/0.2.0)): descriptive statistics of calibrated instrument output. In
addition, previews of eddy4R.turb and eddy4R.erf are provided, which will be published along
NEON's upcoming release of EC Level 4 data products (derived quality-controlled fluxes and
related variables). Development of two additional R-packages eddy4R.stor and eddy4R.ucrt has
started, which provide functionalities for storage flux computation and uncertainty
quantification, respectively. These packages are not covered here, and will be published once
available.
Each eddy4R package consists of a hierarchical set of reusable definition functions, wrapper
functions and workflows. Following best practices, eddy4R is written in controlled and strictly
hierarchical terminology consisting of base names and modifiers, which ensures modular
extensibility over time. Interactive documentation is provided through the use of Roxygen tags
([http://roxygen.org/](http://roxygen.org/)) during development, and follows the Comprehensive R Archive Network
(CRAN; [https://cran.r-project.org/](https://cran.r-project.org/)) guidelines for package dissemination. In addition,
expanded documentation is available in the form of Algorithm Theoretical Basis Documents
from the NEON data portal ([https://w3id.org/smetzger/Metzger-et-al_2017_eddy4R-](https://w3id.org/smetzger/Metzger-et-al_2017_eddy4R-Docker/portal/0.2.0)
[Docker/portal/0.2.0](https://w3id.org/smetzger/Metzger-et-al_2017_eddy4R-Docker/portal/0.2.0)).
EC data processing consists of employing a sequence of model algorithms. These often
originate from scientific sub-fields with corresponding publications, and eddy4R provides an
integrative, yet modular and extensible framework for their concerted application and continued
development: eddy4R.base provides natural constants and basic functions for usability,
regularization, transformation, lag-correction, aggregation and unit conversion ensuring
consistency of internal units at any point in the workflow. Next, eddy4R.qaqc provides the
general quality assurance and quality control (QA/QC) tests of Taylor and Loescher (2013),
along the Smith et al. (2014) model for tracking quality information in large datasets, and
functions for de-spiking (Brock, 1986; Fratini and Mauder, 2014; Mauder et al., 2013; Mauder
and Foken, 2015; Metzger et al., 2012; Vickers and Mahrt, 1997). eddy4R.turb provides
standard, Reynolds-decomposed turbulent flux calculation (Foken, 2017), accompanied by
models for planar fit transformation (Wilczak et al., 2001) and spectral correction (Nordbo and
Katul, 2012). Additional functionalities include Fourier transform, the determination of
detection limit (Billesbach, 2011), integral length scales and statistical sampling errors
(Lenschow et al., 1994), and flux-specific QA/QC models (Foken and Wichura, 1996; Vickers
and Mahrt, 1997). Also, basic scaling variables, atmospheric stability and roughness length
(Stull, 1988), as well as the flux footprint (Kljun et al., 2015; Kormann and Meixner, 2001;
Metzger et al., 2012) can be determined. Lastly, edd4R.erf provides time-frequency de-
composed flux processing and data mining functionalities to determine an environmental
response function model and project the flux fields underlying the EC observations (Metzger et
al., 2013; Xu et al., 2017).
eddy4R can be used with a fully adaptive single-pass workflow (Sect. 3.1), which makes it
computationally efficient compared to the multiple passes required by other flux processing
schemes. In addition, eddy4R is fully parallelized and memory efficient leveraging R's snowfall
parallelization (https://cran.r-project.org/package=snowfall) and ff file-backed object
(https://cran.r-project.org/package=ff) facilities, respectively. This makes eddy4R seamlessly
scalable from local laptop development to deployment across massively parallel computing
facilities. Lastly, its unique modularity permits straightforward adjustments (extensibility) and
versioning as science and/or hardware progresses.

## 2.2   Git distributed version control (DevOps: Verify & Package)

The eddy4R source code resides on a version-controlled Git repository on the hosting service
GitHub (https://github.com/). In general, a developer community uses a version control system
to manage and track different states of their works over time. GitHub provides distributed
version control and has become widely used by scientific research groups because it is free,
open-source, and provides several features that make it useful for managing artifacts of
scientific research (Ram, 2013).

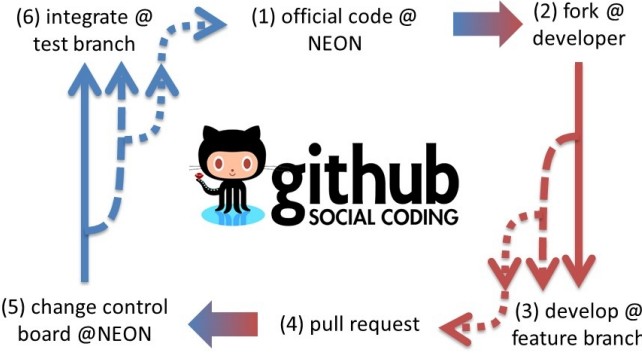

Figure 3. NEON's Git workflow. Please see text in Sect. 2.2 for detailed explanation.

Git allows multiple users and developers to simultaneously access and collaborate on a remote
repository by means of independent 'forks' or replicas of the entire repository (Paarsch and
Golyaev, 2016). Figure 3 shows NEON's Git workflow: At any given time (1) the official,
stable eddy4R source code resides on NEON's GitHub repository. A user can install the eddy4R
packages directly from there, and (2) a developer can 'fork' or copy the repository and create
'branches' for modification. After (3) 'committing' or creating a new feature, the developer (4)
can propose the feature for inclusion in the official eddy4R source code by issuing a pull request
to (5) NEON's change control board. After (6) thorough review and all prior test cases
reproducing benchmark results (DevOps: Verify stage), the feature can be 'merged' or
integrated into the next release of (1) the official, stable eddy4R source code (DevOps: Package

stage). This cycle can be repeated to accommodate requests and future developments, resulting in subsequent releases. Including a test case for new code is strongly encouraged to ensure sustainability over time, but is not mandatory. The developers can periodically update their 'forks' from the remote repository, ensuring that they always work on basis of the most recent eddy4R source code.

The ultimate advantages of Git are provenance, reproducibility and extensibility: Every copy of the code repository includes the complete history of all changes and authorship that can be viewed and searched by anyone (Ram, 2013). This allows developers to build from any stage of the versioned project and makes it easy to collaborate as an integrated scientific community. We note that the DevOps workflow is robust to the business viability of the particular tools used for implementation. Git is simply one instance of a version control system, which could be replaced with another similar tool should Git fail at some point in the future.

## 2.3   Docker image build and deployment (DevOps: Release)

Facilitating the DevOps: Release stage, Docker images (https://www.docker.com/what-docker) wrap a piece of software in a complete filesystem that contains only the minimal context an application needs to run: code, runtime, system libraries and tools. This guarantees that it always performs the same, regardless of the compute environment it is deployed in (i.e. ultimate reproducibility). Compared to the similar but more cumbersome virtual machine approach, a Docker image is an order or magnitude smaller (eddy4R-Docker: 2 GB without example data). Also, by running as native processes it bypasses the virtual machine overhead. Docker is used by many organizations (e.g., National Center for Atmospheric Research, National Snow and Ice Data Center, NSF Agave API), and widely supported across large-scale cloud compute environments (e.g., Amazon EC2 Container Service, Google Container Engine, NSF Xsede). It is particularly well suited to NEON's DevOps strategy: combining development, operation and quality assurance to enable creating, testing, deploying and updating scientific software rapidly and reliably (Figure 2).

Docker can build images automatically by reading the instructions from a Dockerfile. A Dockerfile is a text document that contains all the instructions a user would call on the command line to assemble an image. Using e.g. a cloud hosting platform like DockerHub (https://hub.docker.com/), the image build, versioning and distribution can be automated. This is realized through executing the series of command-line instructions defined in the Dockerfile whenever a new eddy4R source code version is available on GitHub. A key feature of eddy4R-Docker is that it builds upon "Rocker" pre-built Docker images, maintained by the rOpenSci group (https://ropensci.org/). This ensures access to stable, up-to-date base images containing R and a variety of packages commonly used. The eddy4R-Docker image (0.2.0) released in this study was built based on the rocker/ropensci/latest image containing R (3.4.0; (https://hub.docker.com/r/rocker/ropensci/builds/). As specified in the eddy4R Dockerfile, our R packages eddy4R.base (0.2.0) and eddy4R.qaqc (0.2.0) and their dependencies were automatically built on top of this base image. To complete the eddy4R-Docker processing, analysis and modeling environment, the NEON data portal API Client nneo (0.1.0) as well as

the REddyProc (1.0.0) high-level utilities for aggregated EC data were also included. In
addition, the user can install any desired R packages to customize the environment.
Docker's benefits to scientific software development are described in detail in Boettiger (2015).
For NEON's purposes, several Docker properties are particularly important:

- **Portability**: Docker images are portable and independent of the underlying operating
  system. This enables scientists to develop code on local computers or virtual machines
  without worrying about the deployment architecture.
- **Reproducibility**: The DevOps principles are ingrained into the Docker build process,
  thus ensuring a fully traceable and documented Docker image.
- **Streamlined interface between Science and CI**: Defined inputs, outputs and
  instructions provide an ideal framework to isolate and package algorithmic services for
  operational deployment.
- **Continuous development and integration**: Docker provides a modular and extensible
  framework, permitting NEON's data processing to remain up-to-date with the latest
  algorithmic developments. As shown be the nneo and REddyProc examples, it enables
  directly leveraging community-developed code. In this way eddy4R-Docker is
  functionally extensible, while making it easy for the community to incorporate NEON-
  developed code into their own data processing.

## 2.4  Hierarchical Data Format version 5 (DevOps: Configure)

The capability to process large data sets is reliant upon efficient input and output of data, data
compressibility to reduce compute resource loads, and the ability to easily package and access
metadata. The Hierarchical Data Format (HDF5) is a file format that can meet these needs, and
is a key tool aiding the DevOps: Configure (computational resource allocation) stage. A NEON
standard HDF5 file structure and metadata attributes allow users to explore larger data sets in
an intuitive "directory-like" structure that is based upon the NEON data product naming
convention (see Figure 4). Group level 1 separates data by site and site level metadata are
attributed at that level. Group level 2 separates data by data product level (DPL) and DPL
metadata are attributed at that level, where DPLs correspond to the amount of processing
performed. DPL1 are calibrated descriptive statistics, DPL2 are temporally interpolated, DPL3
are spatially interpolated, and DPL4 are further-derived quantities. Group level 3 are the
individual data products, for instance $CO_2$ concentration. Lastly, replicates in the horizontal and
vertical are separated as individual data tables.
This provides a streamlined data-delivery mechanism for the eddy4R-Docker processing
framework. For the tower datasets analyzed in this study, including sonic anemometer, infrared
gas analyzer and mass flow controller data, file sizes ranged from 1 GB for the uncompressed
data in comma-delimited ASCII files to 0.1 – 0.2 GB in HDF5 format, depending on the amount
of missing data. The HDF5 files can be written in a simple format where data are stored as
single 1-dimensional arrays to maximize compression and efficiency, or the data can be stored
as compound datatables that allow multiple datatypes to be written together in columnar format
for ease of navigation when data size is not an issue.

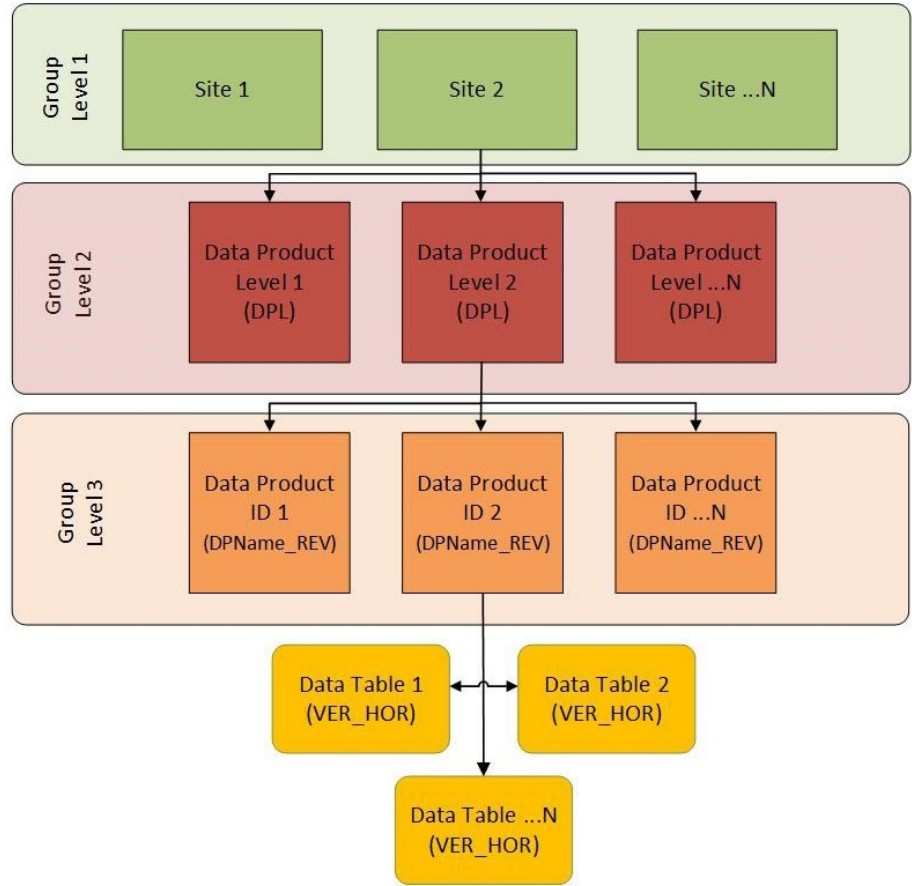


Figure 4 The NEON HDF5 file structure based on the NEON data product naming convention.

Another important function of the HDF5 file format is the ability to attach metadata as attributes,
further promoting reproducibility. The data in this study has the units and variable names as
metadata attached to the data tables in the HDF5 file. Additional metadata are attributed to
various hierarchical groups throughout the file, including environmental parameters, sensor
metadata, and processing parameters. As a result, HDF5 and similar self-documenting
hierarchical data formats are gaining traction in a community that has traditionally relied on
ASCII text column or comma-delimited files, especially as tools for viewing, manipulating, and
extracting data from HDF5 become more commonplace. The utility of HDF5 file format is
demonstrated in the executable example workflow that accompanies this manuscript (see
Sects. 2.6, 5).
**2.5  Modular compatibility with existing compute infrastructure (DevOps:**
**Configure & Monitor)**
To perform a defined series of processing steps, a Docker image is called with a workflow file,
resulting in a running instance called Docker container (Figure 5). Through this mechanism, an
arbitrary number of Docker containers can be run simultaneously performing identical or
different services depending on the workflow file. This provides an ideal framework for scaled
deployment using e.g. high-throughput compute architectures, cloud-based services etc.

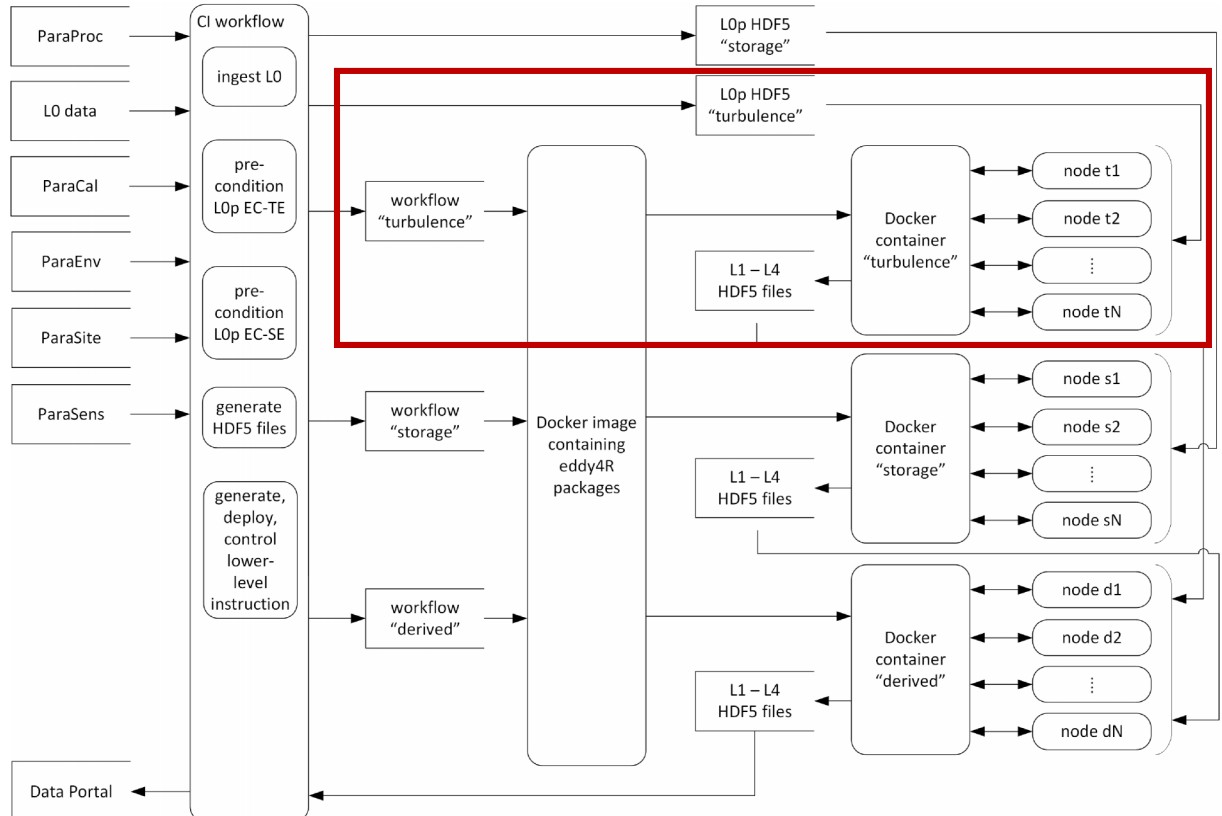


Figure 5. NEON's eddy4R-Docker EC processing framework. The red box visualizes the scope
of the present study, and individual components are described in the text.

Embodying the DevOps: Configure stage, NEON's eddy4R-Docker EC processing framework
begins with ingesting information from various data sources on a site-by-site basis (Figure 5
top left). This includes EC raw data (Level 0, or L0 data) alongside contextual information on
measurement site (ParaSite), environment (ParaEnv), sensor (ParaSens), calibration (ParaCal),
as well as processing parameters (ParaProc). Next, the raw data is preconditioned and all
information is hierarchically combined into a compact and easily transferable HDF5 file
(Figure 5 panel "CI workflow"). Each file contains the calibrated raw data (L0 prime, or L0p)
and metadata for one site and one day, either for EC turbulent exchange or storage exchange.
In this manuscript, we focus on demonstrating the turbulence data process and analysis in the
red box of Figure 5. Together with the "turbulence" workflow file the HDF5 L0p data file is
passed to the eddy4R-Docker image, where a running Docker container is spawned that scales
the computation over a specified number of compute nodes (Figure 5 top right). The resulting
higher-level data products (Level 1 – Level 4, or L1-L4) are collected from the compute nodes
and, together with all contextual information, are combined into a daily L1-L4 HDF5 data file
that is served on the data portal (Figure 5 bottom left). In addition to the daily output files,
monthly concatenated files are also available for download from the NEON data portal
(https://w3id.org/smetzger/Metzger-et-al_2017_eddy4R-Docker/portal/0.2.0). This sequence
is performed analogously for different combinations of workflows and data, and it is possible
for the workflow instruction sets to interact with each other. For example, the "turbulence" and
"storage" containers are processing in parallel, and starting the "derived" container once all
intermediary results are available (Figure 5 bottom right). It should be noted that the
"turbulence", "storage" and "combined" Docker containers (Figure 5 right) are all spawned
from the same eddy4R-Docker image (Figure 5 center): each container includes the same
underlying functionality (eddy4R packages), but serves a different purpose by being fed the
"turbulence", "storage" or "combined" workflow files.
This eddy4R-Docker EC processing framework modularly integrates into pre-existing data
processing pipelines, such as NEON's CI (Figure 6): in NEON's pre-existing framework the
CI group encoded simple algorithms (e.g. temporal means) in Java, based on algorithm
documentation provided by Science staff. The key difference of the eddy4R-Docker EC
processing framework is that instead of algorithm documentation, NEON Science staff now
provide documented algorithms that perform a complex series of processing steps, which can
be directly deployed by CI. Not only does this adoption of the NEON-DevOps workflow
(Figure 2) streamline end-to-end operational implementation and efficiency, it empowers the
Science community at large by putting the key to the scientific algorithms into the hand of
scientists.

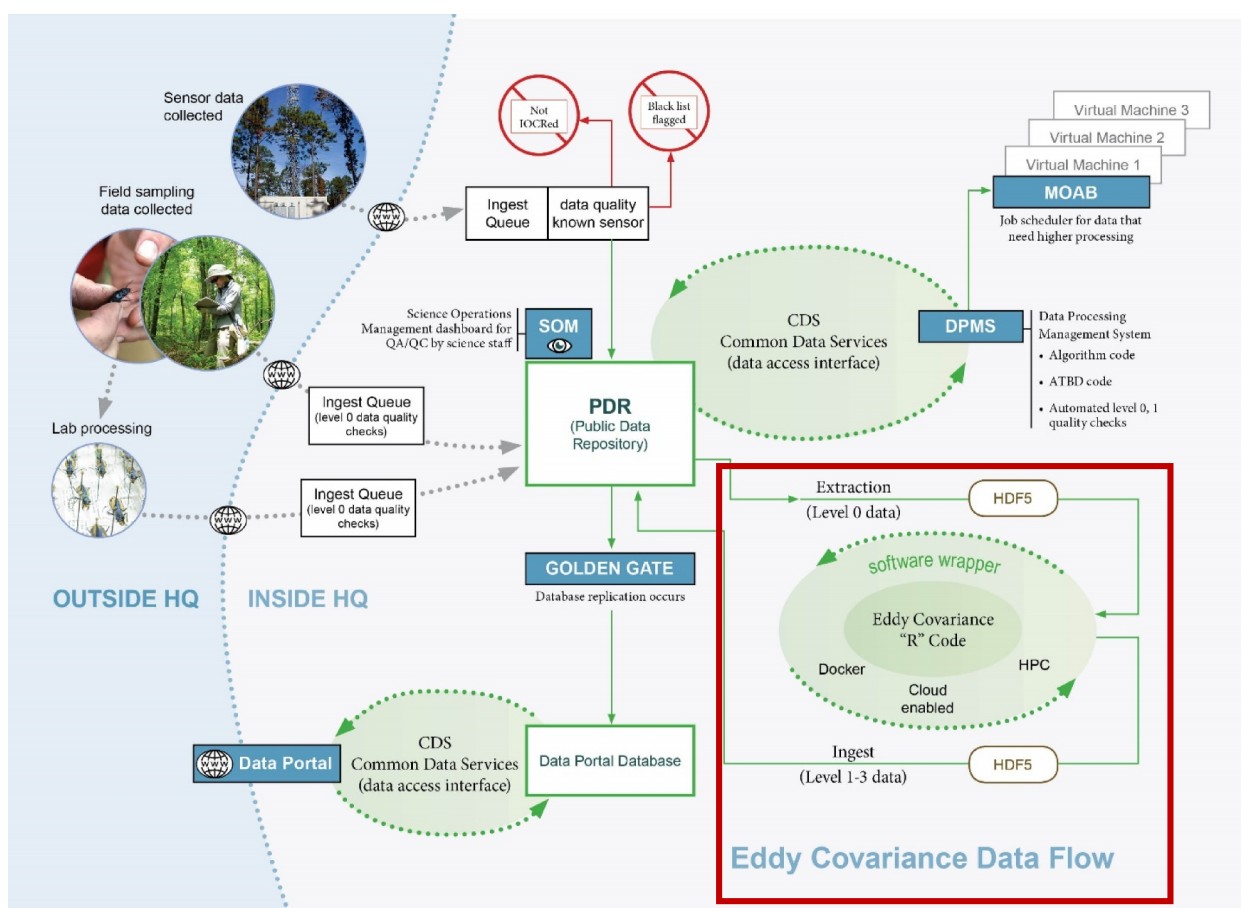


Figure 6. NEON's CI for streaming data processing. The red box visualizes the eddy4R-Docker
EC processing framework within the overall CI.

To address the DevOps: Monitor stage, the computational resource load and performance statistics of operating eddy4R-Docker can easily be monitored with standard profiling procedures within NEON's CI or other compute infrastructures. eddy4R-Docker further utilizes the R logging package (0.7-103) to provide hierarchical logging, multiple handlers and formattable log records. Finally, end-user experience is monitored via the Issues feature in GitHub, where users can report code bugs, deployment problems, etc.

## 2.6  Installation and operation

One source of resistance to reproducible research is the initial burden of learning a new workflow. The eddy4R-Docker image aims to reduce the initial setup effort and learning requirements. This is achieved by providing a computational environment that contains all the necessary software dependencies, the Rstudio graphical development environment (https://www.rstudio.com/), and a code base consisting of example workflows and easily accessible functions. Combined with a simple and thoroughly documented installation procedure it provides a similar feel to working locally.

To work with the eddy4R-Docker image, one first needs to sign up at DockerHub (https://hub.docker.com/) and install the Docker host software following the Docker installation instructions (https://docs.docker.com/engine/getstarted/step_one/). Next, the download of the eddy4R-Docker image and subsequent creation of a container can be performed by two simple commands in an open shell (Linux/Mac) or the Docker Quickstart Terminal (Windows):

```
docker login
docker run -d -p 8787:8787 stefanmet/eddy4r:0.2.0
```

The first command will prompt for the user's DockerHub ID and password. The second command will download the latest eddy4R-Docker image and start a Docker container that utilizes port 8787 for establishing a graphical interface via web-browser. The release version of the Docker image can be specified, or alternatively the specifier `latest` provides the most up-to-date development image. In addition, it is possible to download and run a specific digest using the `docker run stefanmet/eddy4r@sha256:` command. If data is not directed from/to cloud hosting, a physical file system location on the host computer (`local/dir`) can be mounted to a file system location inside the Docker container (`docker/dir`). This is achieved with the `docker run` option `-v local/dir:docker/dir`.

The interactive Rstudio Server session running inside the Docker container can then be accessed via a web browser at http://host-ip-address:8787, using the IP address of the Docker host machine. The IP address of the Docker host can be determined by typing `localhost` in a shell session (Linux/Mac) or by typing `docker-machine ip default` in cmd.exe (Windows). Lastly, in the web browser the user can log into the RStudio session with username and password `rstudio` (see Figure 7 top left panel). Figure 7 also shows the Rstudio integrated development environment and interactive help for the eddy4R.base package in the bottom and top left panels, respectively. Additional information about the use of Rstudio and eddy4R packages in Docker containers can be found on the rocker-org/rocker website

(https://github.com/rocker-org/rocker/wiki/Using-the-RStudio-image) and the eddy4R Wiki
pages.

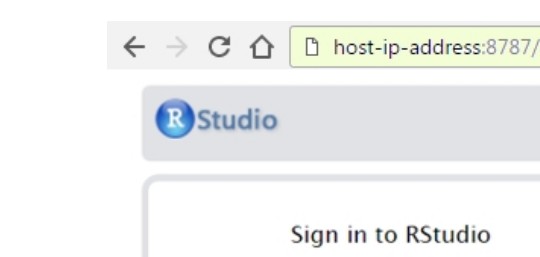

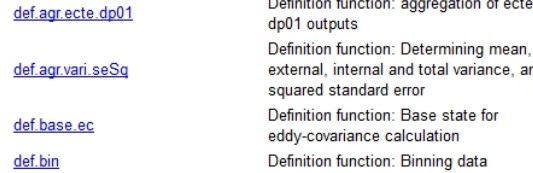

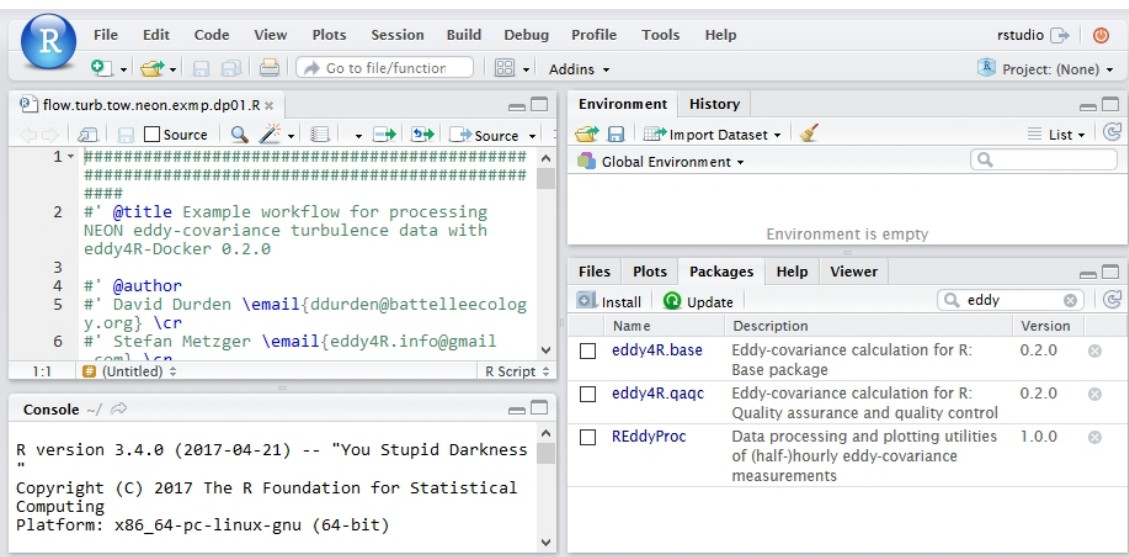

Figure 7 Docker-based Rstudio server session login via web-browser. Top left panel: Sign-in
screen with highlighted areas showing information to input by the user. Top right panel:
Interactive help for the eddy4R.base package. Bottom panel: Integrated development
environment with workflow template, R console, Git staging area and eddy4R packages.

To demonstrate the ease of "Docker-assisted" data analysis and provide a template for potential
eddy4R-Docker users, an executable example workflow and data are included in the eddy4R-
Docker image. Once the eddy4R container is started, the example workflow, input data (NEON
dp0p HDF5 file) and output data (NEON dp01 HDF5 file) are available from the Docker-
internal directory `/home/eddy/`. The example workflow is located at
`/home/eddy/flowExmp/flow.turb.tow.neon.exmp.dp01.R`, and provides a selection of the
processing steps that yield the EC dp01 data on the NEON data portal
([https://w3id.org/smetzger/Metzger-et-al_2017_eddy4R-Docker/portal/0.2.0](https://w3id.org/smetzger/Metzger-et-al_2017_eddy4R-Docker/portal/0.2.0)). The example
workflow is fully documented to guide readers through the various processing steps, and
employs key functionalities of the eddy4R.base and eddy4R.qaqc packages. These include data
and metadata import from the input HDF5 file, data assignment to file-backed objects,
processing of 1 minute and 30 minute data statistics and data quality, and writing the output
HDF5 file. In addition, outputs from the quality flag and quality metric model are visualized.
As described above, the eddy4R Docker image can be used for code development (DevOps:
Create stage) through accessing a running eddy4R Docker container via a web browser.
Alternatively, the eddy4R Docker image can be used from the command line to perform scaled
batch processing (DevOps: Configure & Monitor stages). Deployment from the command line
consists of passing the R workflow file to the Docker image. This is achieved by using the
`docker run` command with the additional argument `Rscript docker/dir/filename.R`, with
`filename.R` being the desired workflow. Thus, the eddy4R Docker image can be used to
simultaneously deploy multiple Docker containers to process data for multiple days or sites to
the capacity of the computational platform.

## 3   Test applications

In the following we present three test applications of eddy4R-Docker to evaluate whether the
NEON DevOps model can indeed produce collaborative, portable, reproducible, and extensible
EC software. Code development, packaging, release, and operation followed the NEON
DevOps model presented in this paper. Code modules have been contributed by order 10
individuals, distributed across multiple institutions and utilizing various computer systems.
Nevertheless, each contributor achieved identical results per validation scripts during the
DevOps: Verify & Package stage (Sect. 2.2), emphasizing the achieved portability and
reproducibility. The majority of the calculations presented here were performed on 12 Intel
Xeon X5550 2.67GHz CPUs, 32 GB memory with 10 Mbit interconnects and 10 Mbit access
to 8 TB storage on an Oracle Zettabyte File System. The software specifications were CentOS 7
(3.10.0-327.el7.x86_64) with docker-engine (1.11.0). In Sect. 3.1, results of processing 12 days
of EC data from a fixed tower at a NEON field site are shown. Next, in Sect. 0, we present the
processing of EC fluxes from a 1-hour recording of a moving platform: airborne observations
in a convectively mixed boundary layer. Lastly, a validation via software intercomparison is
provided in Sect. **Error! Reference source not found.**.

## 3.1   Tower eddy-covariance measurements

Here, we use tower EC measurements to test a typical implementation of the eddy4R processing
framework. The Smithsonian Environmental Research Center (SERC) in Edgewater, MD, USA
is located on the Rhode and West Rivers, and hosts the NEON SERC tower (38°53'24.29" N,
76°33'36.04" W; 30 m a.s.l.). The ecosystem at SERC is a closed-canopy hardwood deciduous

forest dominated by tulip popular, oak and ash, with a mean canopy height of approximately 38 m (Figure 8). EC turbulent flux sensors are mounted at the tower top at 62 m above ground or 24 m above the forest canopy.

An enclosed infrared gas analyzer (IRGA, LI-COR Biosciences, Lincoln, NE, USA, model: LI-7200, firmware version 7.3.1.) was used to measure the turbulent fluctuations of $H_2O$ and $CO_2$. A mass flow controller (Alicat Scientific, Burlington, VT, USA, model: MCRW-20 SLPM-DS-NEON) was used to maintain a constant flow rate of 12 SLPM through the IRGA cell. A sonic anemometer (Campbell Scientific, Logan, UT, USA, model: CSAT3, firmware version 3) was used to measure the 3-dimensional turbulent wind components. Data from the IRGA and the sonic anemometer was synchronized using triggering and network timing protocol, and collected simultaneously at 20 Hz sampling rate.

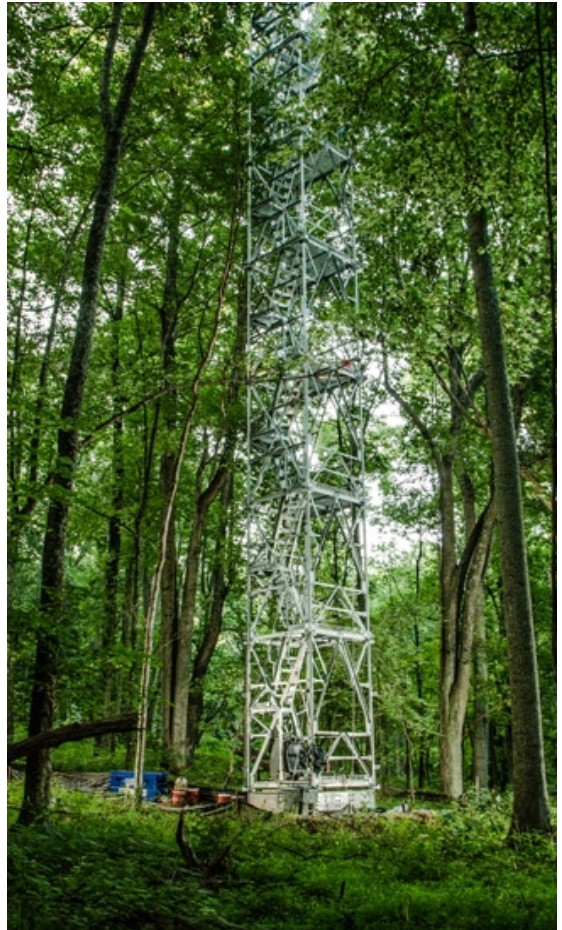
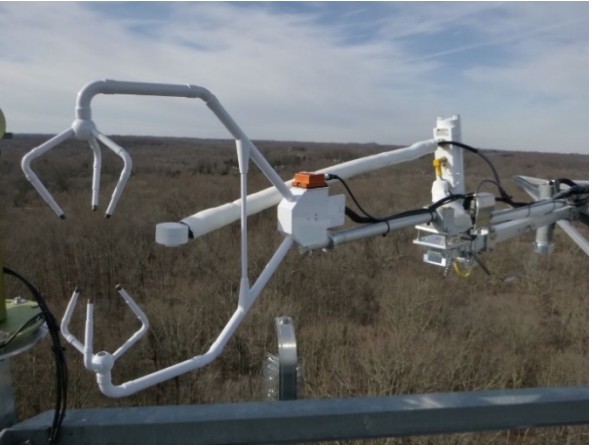
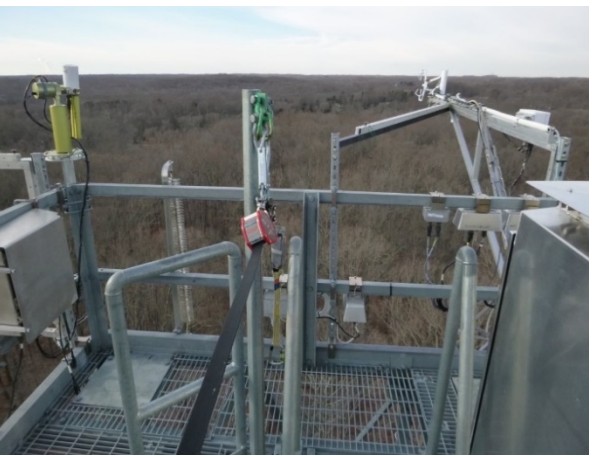

Figure 8. Left panel: Ecosystem at the NEON SERC tower (credit: Stephen Voss Photography; http://www.stephenvoss.com/blog/neon-tower-smithsonian). Right panels: EC instrumentation on top of the NEON SERC tower. Right top panel: Campbell Scientific CSAT-3 three-dimensional sonic anemometer (front) and LI-COR Biosciences LI-7200 infrared gas analyser (back) on the retracted tower-top boom. Right bottom panel: Same instrumentation but with the tower-top boom extended at 230° from true north.

Here, data from April 22 to May 3, 2016 were used. The mean temperature during this time
period was 15°C, with a maximum temperature of 29°C and a minimum of 8°C. A total of
15 mm of precipitation was observed at nearby Annapolis Naval Academy.

### 3.1.1 Algorithm settings and profiling

The eddy4R workflow file was configured to ingest on the order of 50 data streams at 20 Hz,
including 3-D wind components, sonic temperature, and $H_2O$ and $CO_2$ concentrations. The data
were processed to half-hourly L1 data products and turbulent fluxes. The L1 data products are
essentially state variables (wind, temperature, concentrations) with basic statistical products
derived, i.e. mean, minimum, maximum, standard error of the mean and variance. The
algorithmic processing for the L4 flux calculations requires additional scientific and procedural
complexity to test assumptions of the EC theory. The resultant fluxes represent half-hourly
vertical turbulent exchanges between the earth's surface and the atmosphere corresponding to
these state variables.
For the datasets analyzed in this study, the L0p input file sizes ranged from 0.1 – 0.2 GB in
HDF5 format depending on the amount of missing data, with metadata attached as attributes.
We used the simple data format for our HDF5 files, as opposed to compound data type, this
resulted in reduced read in time from 60 seconds to 3 seconds for 20 Hz IRGA data. Elementary
testing indicates that in this framework 6 CPU-minutes were required to process 1 day of 20 Hz
L0 data, and 1.2 CPU-minutes per 1 day of L0p data (100,000,000 observations). This
difference arises mainly from application of plausibility tests per Taylor and Loescher (2013)
in the transition from L0 to L0p. No reduction in efficiency was observed between direct
software deployment and its Docker implementation. Once flux QA/QC and uncertainty budget
is implemented, the computational expense will likely increase by a factor of two to three. This
suggests that eddy4R performs comparably to other flux processors. Memory usage is kept
below 2 GB through the use of fast access file-backed objects, enabling more sophisticated
scientific analyses through access to multiple days of data without overloading random access
memory (RAM) resources. Additionally, the snowfall R package allows for logical
parallelization frameworks to be implemented in the processing framework, even at low-level
analysis steps.

### 3.1.2 Results and discussion

The time series ranging from April 22 to May 3, 2016 was processed to deliver both state (L1)
and flux (L4) quantities; however, the initial eddy4R package release will only contain
functions necessary to report state variables or L1 data products in the NEON data product
description. During the processing of the proof-of-concept results averaging periods with >10%
missing data (incl. bad sensor diagnostic flags) were removed, and dedicated flux QA/QC and
uncertainty quantification were disabled.
Figure 9 shows the resultant time series of shear stress (friction velocity), sensible heat, latent
heat and $CO_2$ flux. The derived values fall into typical ranges for mid-latitude hardwood forests
in spring. As expected, fluxes follow the general trends in the scalar quantities. Good data
coverage can be seen for the LI-7200 measurements even during the rainy period at the end of
the analysis. A footprint analysis revealed that 90% of the flux measurement signals were
sourced within 800 m from the tower, and 80% were within 500 m from the tower at our site.
Data coverage was reduced after day of year (DOY) 120 due to inclement weather conditions.

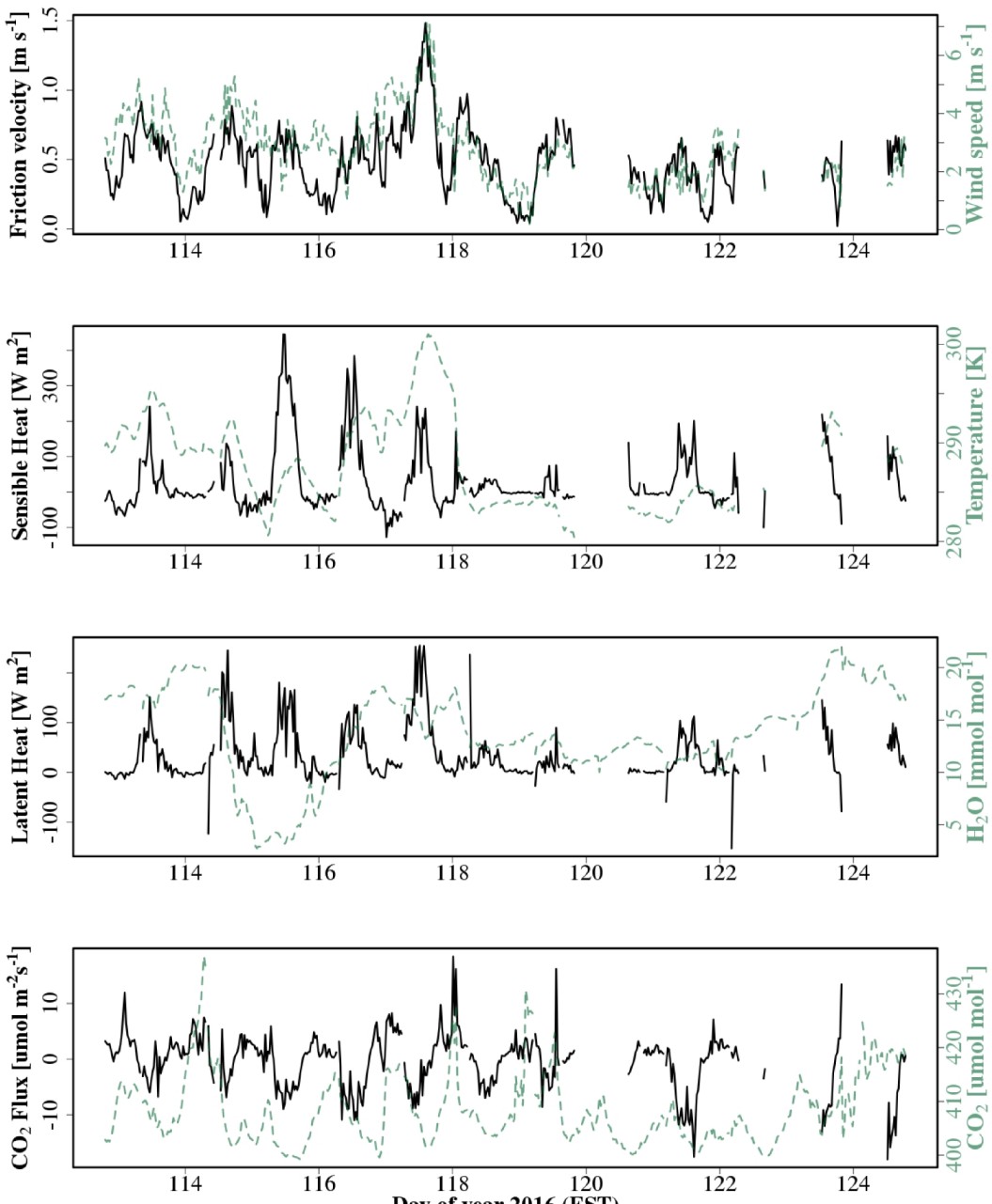


Figure 9. Time-series of turbulent fluxes derived from EC measurements atop the NEON SERC
tower. Top to bottom: Vertical turbulent exchange of shear (friction velocity) and wind speed,
sensible heat and temperature, latent heat and $H_2O$ dry mole fraction and $CO_2$ flux and $CO_2$ dry
mole fraction.
The spiky results preceding and following periods with >10% invalid data highlight the need
for enabling the full flux QA/QC and uncertainty budget to subset science-grade fluxes. This
implementation of eddy4R in a Docker image, as it will interact with NEON CI, clearly
demonstrates the applicability of the DevOps model for generating EC L1-L4 data products.

## 3.2 Aircraft eddy-covariance measurements

Here, we use aircraft EC measurements to test more advanced scientific capabilities of the
eddy4R processing framework. Airborne turbulent flux observations were performed along
more than 3100 km of low level (i.e. 50 m above ground level) flights across the North Slope
of Alaska, USA in July 2012, using the research aircraft Polar 5 (Tetzlaff et al., 2015). The
example data used in this manuscript were recorded during a SSW-NNE flight line near the
village of Atqasuk, Alaska, above tundra dominated by sedges and emerging herbaceous
wetland vegetation. Large, often oriented lakes and the meandering Meade River characterize
the surrounding landscape.
The aircraft was equipped with a 3 m nose boom holding a 5-hole probe for wind measurements,
an open wire Pt100 in an unheated Rosemount housing for air temperature measurements, and
an HMT-330 (Vaisala, Helsinki, Finland) in a Rosemount housing for relative humidity.
Sample air was drawn from an inlet above the cabin at about 9.7 l s$^{-1}$, analysed in an RMT-200
cavity-ringdown trace gas sensor (Los Gatos Research Inc., Mountain View, California, USA)
and recorded at 20 Hz. Aircraft position, velocity and attitude was provided by several Global
Positioning Systems (NovAtel Inc., Calgary, Alberta, USA) and an Inertial Navigation System
(Laseref V, Honeywell International Inc., Morristown, New Jersey, USA). Height above ground
was determined by a radar altimeter (KRA 405B/ Honeywell International Inc., Morristown,
New Jersey, USA) and a laser altimeter (LD90/ RIEGL Laser Measurements Systems GmbH,
Horn, Austria). The input data used in this study included the pre-derived 3-D wind vector from
5-hole probe and aircraft position, velocity and attitude. After spike removal the sampling
frequency of the original data was reduced from 100 Hz to 20 Hz resolution using block
averaging. These steps were performed prior to import into eddy4R processing, but could
equally well be performed therein.

### 3.2.1 Algorithm settings and profiling

Here, aircraft-measured vertical wind speed and $CH_4$ dry mole fraction were analysed to
determine $CH_4$ emissions by means of a time-frequency-resolved version of the EC method
(Metzger et al., 2013). For this purpose a combination of settings were chosen in the eddy4R
workflow file that differ from Sect. 3.1: Initially the small (<1 MB) EC raw data file consisting
of 17 variables and 12,800 data points (or 42 km flight data) was read in ASCII Gzip format –
standard R capabilities for data ingest can be used to read data in various formats, frequencies
and units. Aircraft-measured vertical wind speed and $CH_4$ dry mole fraction were then
correlated using a Wavelet transform (Metzger et al., 2013). This process includes ranging and
de-spiking of unphysical raw data values (Mauder et al., 2013; Metzger et al., 2012), fast dry
mole fraction derivation (e.g., Burba et al., 2012) and spectroscopic correction (Tuzson et al.,
2010) of $CH_4$ trace gas observations, and high-frequency spectral correction (Ammann et al.,
2006) by means of applying a sigmoidal transfer function (Eugster and Senn, 1995) directly in
Wavelet space. This permits estimating turbulent fluxes with improved spatial discretization
and determining ~100 biophysically relevant surface properties in the flux footprint. The
analysis took 56 minutes with 8-fold parallelization and consumed <3 GB RAM thanks to the
use of fast access file-backed objects.

### 3.2.2 Results and discussion

The resulting Wavelet cross-scalogram (Figure 10) is integrated in frequency over transport
scales up to 20 km, and along the flight path over a 1000 m moving window with 100 m step
size, similar to the resolution of the land surface data. The result is an in-situ observed space-
series of the $CH_4$ surface-atmosphere exchange at 100 m spatial resolution. Analogously,
turbulence statistics characterizing shear stress and buoyancy are determined for characterizing
the atmospheric transport between the emitting land surface and the aircraft position.

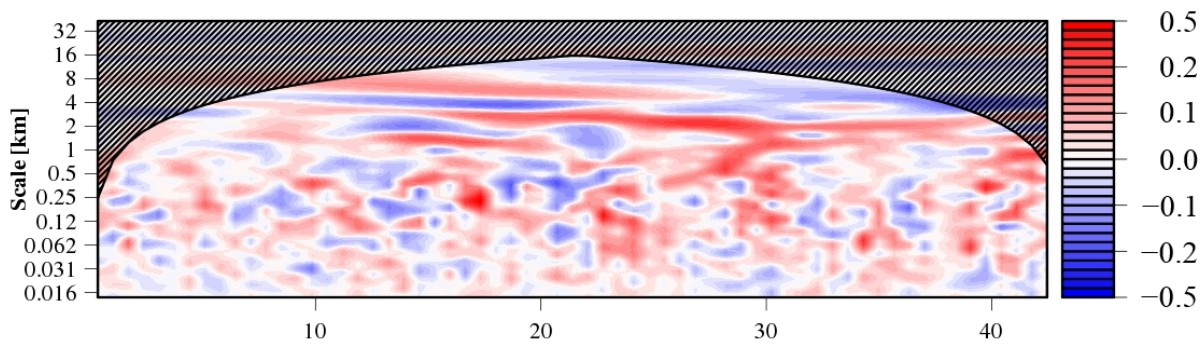

Figure 10. Wavelet cross-scalogram of the $CH_4$ flux equivalent to a time (x-axis) frequency (y-
axis) resolved version of EC. For each combination of aircraft position and eddy size, blue and
red areas indicate transport toward and away from the surface, respectively.

Corresponding systematic and random statistical errors are calculated following Lenschow and
Stankov (1986) and Lenschow et al. (1994), and the flux detection limit is calculated after
Billesbach (2011).
The relationship between the aircraft-observed $CH_4$ surface-atmosphere exchange and land
surface properties is established through an atmospheric transport operator, the so-called flux
footprint function (e.g., Schmid, 1994). Here we use a computationally efficient one-
dimensional parameterization of a Lagrangian particle model for the along-wind footprint
extent (Kljun et al., 2002; Kljun et al., 2004), combined with an analytical approach to
determine cross-wind surface contributions to each 100 m aircraft measurement, depending on
aircraft position (Figure 11; Metzger et al., 2012).
For each 100 m observation of the $CH_4$ surface-atmosphere exchange an individual footprint
weight matrix derived from the footprint parameterization is convolved with the land surface
drivers. The results are space-series of land surface contributions accompanying the CH₄
measured surface-atmosphere exchange (Figure 12).

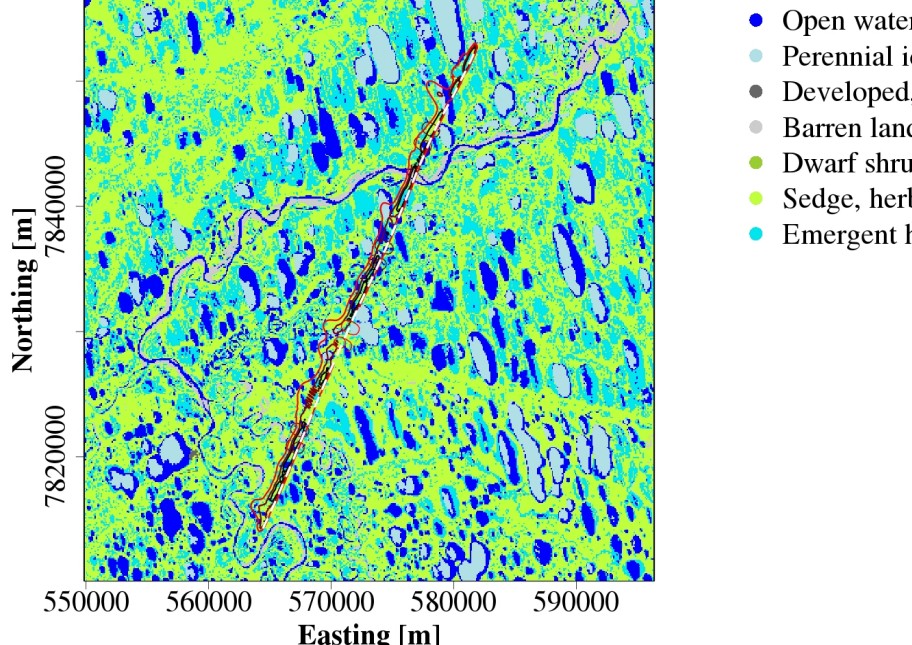

Figure 11. The composite flux footprint along the flight line (30 %, 60 %, 90% contour lines)
superimposed over the National Land Cover Database. The white dashed line represents the
aircraft flight track.

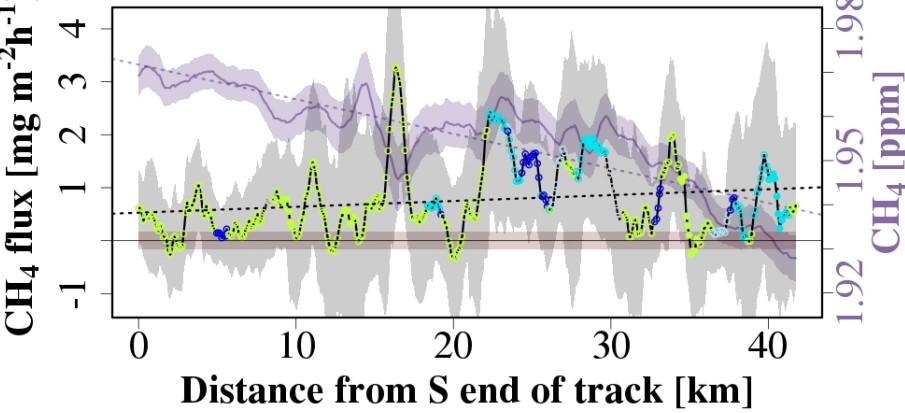


Figure 12. Space-series of 399 CH₄ concentration (purple line) and flux (black line)
observations each 100 m, averaged over 1000 m windows. The random sampling errors are
indicated by the shaded areas enveloping each line, and the flux detection limit is shown as
salmon envelope around the abscissa. Circles indicate the dominating land cover in the footprint
of each observation (Figure 11) with full circles corresponding to 'pure' fluxes (>80% surface
contribution).

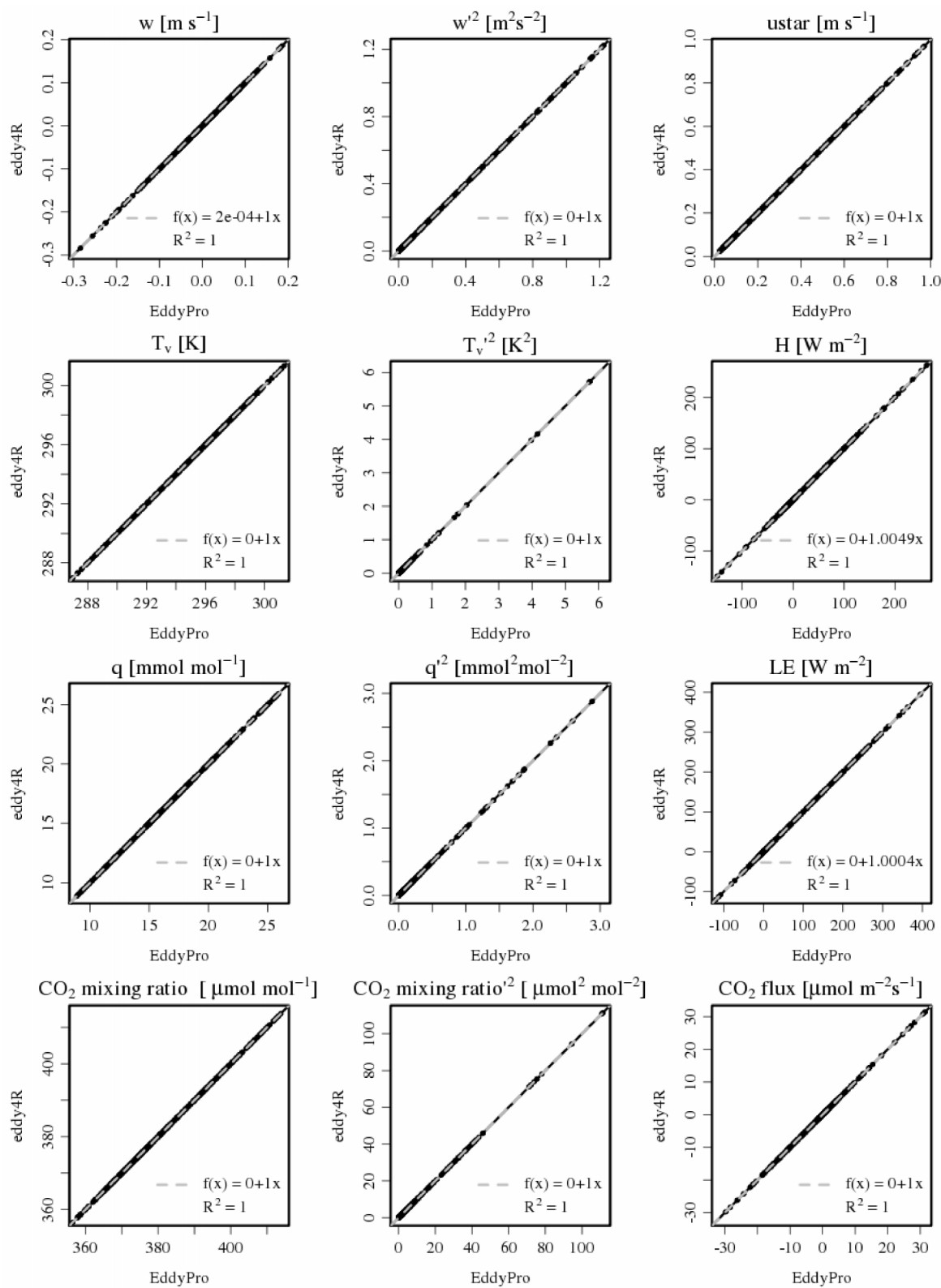


Figure 13. Scatterplot of means (vertical wind speed, w; sonic temperature, $T_v$; $H_2O$ dry mole fraction (q); and $CO_2$ dry mole fraction), variances and fluxes (friction velocity, ustar; sensible heat flux, H; latent heat, LE; $CO_2$ flux). Data are generated from 2011 July to Aug WLEF data in EddyPro and eddy4R. Each point represents a one-hour averaging period. Black lines are 1:1 lines, and dashed lines are robust regressions (Salibian-Barrera and Yohai, 2006).

The successful application of eddy4R-Docker to both, basic tower and advanced aircraft EC
data analyses, highlights how the DevOps model promotes modular extensibility.

## 3.3 Validation and verification

eddy4R includes a verification script which automatically processes subsets of the tower and
aircraft data introduced in Sect. 3.1 and Sect. 3.2, and verifies the results against a reference,
e.g. generated with a different software.
Here, we demonstrate such approach at the Park Falls, Wisconsin WLEF very tall tower
Ameriflux site (US-PFa). The 447 m tall WLEF television tower (45.946∘N, 90.272∘W) has
been instrumented for EC measurements in 1996, and is part of the AmeriFlux network. Flux
measurements at 30 m, 122 m and 396 m sample a mixed landscape of forests and wetlands
(Desai et al., 2015). The surrounding forest canopy has approximately 70% deciduous and 30%
coniferous trees, and a mean canopy height of 20 m. The site has an interior continental climate.
Instrumentation at each level consists of fast response wind speed and temperature from a sonic
anemometer (Applied Technologies., Inc., Seattle, USA, ATI Type K). 10 Hz dry mole fraction
of $CO_2$ and $H_2O$ at the 122 m level used here were measured by a closed-path infrared gas
analyzer (LI-COR, Inc., Lincoln, USA, LI-6262) located on the tower.
A data set from July 27 to August 19, 2011 was used in the intercomparison between eddy4R
and the reference software EddyPro (LI-COR, Inc., Lincoln, USA, version 6.2.0). EddyPro was
released in April 2011 and is widely used in the EC community.

### 3.3.1 Algorithm settings

Several preprocessing steps were applied, and the resulting data and settings were used in both,
eddy4R and EddyPro: (i) The raw data was pre-cleaned in eddy4R using the Brock (1986) de-
spiking algorithm with a filter width of 9 data points for all variables. (ii) EddyPro was used to
calculate the planar-fit rotation parameters (Wilczak et al., 2001) over the entire dataset (offset
$= -0.06$ ms$^{-1}$, pitch $= -5.27°$, roll $= -1.81°$). (iii) Time lags for dry mole fractions of $CO_2$ (0.8 s
behind vertical wind) and $H_2O$ (0.1 s behind vertical wind) were calculated in eddy4R using
maximum correlation (median lag time over entire dataset).
Because $CO_2$ and $H_2O$ fluxes were calculated from dry mole fractions, the Webb et al. (1980)
density correction was not necessary and therefore not applied (Burba et al., 2012). Frequency
response correction was not considered in this validation and therefore not applied. Means,
variances and fluxes were calculated on the basis of one-hour block averages. Based on
Schotanus et al. (1983), sensible heat flux was calculated from point-by-point conversion of
sonic temperature in eddy4R, and with the half-hourly statistical correction in EddyPro.

### 3.3.2 Results and discussion

eddy4R and EddyPro produce nearly identical results (Figure 13), and the gain error is within
0.04% for most outputs. Sensible heat flux values produced by eddy4R have slightly larger
magnitude compared to EddyPro, by 0.49%. This is likely a result of the different methods
applied when converting sonic temperature to air temperature. This intercomparison confirms
that applying the DevOps model to scientific EC software achieved results comparable to
commercial-grade software. A detailed end-to-end intercomparison considering additional
processing steps and EC software is planned for a separate manuscript accompanying NEON's
release of flux data products.
## 4 Summary and conclusions
Adopting a DevOps philosophy has facilitated the creation of a universal processing
environment for producing NEON's EC data products. Portable, reproducible, and extensible
software is reliably and efficiently created by incorporating the DevOps workflow steps of Plan,
Create, Verify, Package, Release, Configure, and Monitor into a NEON-specific DevOps model
based on the tools R, Git, HDF5, and Docker. Git-distributed version control facilitates
simultaneous internal-external collaboration on scientific algorithms, the outcome being a
modular family of open-source R packages. The use of Hierarchical Data Format allows for
efficient, self-describing data input and output. Docker images package the entire processing
environment for robust, scalable, and portable deployment. The capability of this framework
was demonstrated with cross-validated tower and aircraft fluxes.
The results presented here are from a file-based implementation of the eddy4R Docker
workflow, with EC instrument data accessed directly e.g. from the NEON site and manually
processed into the HDF5 ingest format (Sect. 2.5). The subsequent focus is the operational
implementation of the eddy4R-Docker workflow for reporting means and variances. This
includes: (i) Automated ingest of streaming raw data into the NEON database; (ii) Processing
of raw data into the standard, defined inputs required by the eddy4R-Docker in HDF5 format,
and (iii) Developing the software and hardware infrastructure to pass data and instructions back
and forth to the eddy4R-Docker workflow, and control program execution in a distributed
computing framework.
Remaining scientific algorithms are being integrated into eddy4R-Docker for producing
turbulent exchange data products. These algorithms include on-the-fly de-spiking, lag
correction, planar-fit and spectral correction, flux QA/QC, and uncertainty budget estimation.
Finally, eddy4R-Docker is being expanded to include "storage" and "derived" workflows
(Figure 6) for generating reproducible net ecosystem exchange data products in 2018. Lessons
learned here will profit the community at large, e.g. through enabling streaming processing
directly at an EC site or over cellular modems with the same eddy4R-Docker open-source
software as used for sophisticated analyses (Sect. 3.2). Already now, the executable example
workflow and data included in eddy4R-Docker image invite the reader to realize their own end-
to-end data analysis and apply it to their data (Sects. 2.6, 5).
While our sole focus in developing and implementing this model has been to generate EC data
products with the unique capabilities and constraints of NEON, it has become clear that the
NEON DevOps model enables the implementation of a suite of complex processing algorithms,
such as temporal gap filling of sensor time series data or modeling re-aeration rates. There exist
many potential synergies between NEON, other tower networks, and the user community for
producing high level EC data products. We hope this framework can serve as a model for
implementing community-sourced, distributed-development scientific code while combatting

the deficiencies of current computational frameworks that limit accessibility, reproducibility, and extensibility.

## 5 Code and data availability

The source code packages eddy4R.base (0.2.0) and eddy4R.qaqc (0.2.0) used in this study are archived at https://w3id.org/smetzger/Metzger-et-al_2017_eddy4R-Docker/code/0.2.0, under the GNU Affero General Public License (GNU AGPLv3). Similarly, the corresponding eddy4R-Docker image (0.2.0), including an executable example workflow and data, is available at https://w3id.org/smetzger/Metzger-et-al_2017_eddy4R-Docker/docker/0.2.0. In addition, a data supplement is provided at https://w3id.org/smetzger/Metzger-et-al_2017_eddy4R-Docker/data/0.2.0, including an extended abstract and all NEON SERC raw data used in this study, accompanied by variable documentation. Lastly, NEON EC data products generated with eddy4R-Docker are available at https://w3id.org/smetzger/Metzger-et-al_2017_eddy4R-Docker/portal/0.2.0.

## Acknowledgements

Many colleagues at Battelle Ecology supported this study. In particular, Santiago Bonarrigo provided pre-parsed high-frequency data from the SERC site, and Andrew Fox (now: National Center for Atmospheric Research), Mike SanClements and David Hulslander commented on an earlier version of the manuscript. Henry Loescher and Leslie Goldman designed Figure 6, and Andrea Thorpe (now: Washington Natural Heritage Program), Thomas Gulbransen and Michael Kuhlman helped shepherding this study and its publication through required administrative procedures. Special thanks goes to Timothy Brown at the National Oceanic and Atmospheric Administration, for numerous discussions and invaluable advice on scientific computing. The National Ecological Observatory Network is a project sponsored by the National Science Foundation and managed under cooperative agreement by Battelle Ecology, Inc. This material is based upon work supported by the National Science Foundation under the grant DBI-0752017. Any opinions, findings, and conclusions or recommendations expressed in this material are those of the author(s) and do not necessarily reflect the views of the National Science Foundation. Ankur Desai acknowledges support from NSF DBI-1457897 and DOE Office of Science Ameriflux Network Management Project core site support to the ChEAS cluster. Torsten Sachs and Andrei Serafimovich are supported by the Helmholtz Association of German Research Centres through a Helmholtz Young Investigators Group grant to Torsten Sachs (grant VH-NG-821).

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
