# Peer review of "eddy4R 0.2.0: A DevOps model for community-extensible"

_Geoscientific Model Development, 2016_

## Short Comment (SC1) · 24 Feb 2017

Dear authors,

in my role as Executive editor of GMD, I would like to bring to your attention our Editorial version 1.1:

http://www.geosci-model-dev.net/8/3487/2015/gmd-8-3487-2015.html

This highlights some requirements of papers published in GMD, which is also available on the GMD website in the 'Manuscript Types' section:
http://www.geoscientific-model-development.net/submission/manuscript_types.html

In particular, please note that for your paper, the following requirement has not been met in the Discussions paper:

- "The main paper must give the model name and version number (or other unique identifier) in the title."

Yourselves state in the abstract "While this proof-of-concept represents a significant advance, substantial work remains to arrive at the automated framework needed for the streaming generation of science-grade EC fluxes." So further developments of the framework are to expected. Therefore the described current state of eddy4R should be labelled by a version number in the title upon your revised submission to GMD.

Yours,

Astrid Kerkweg

---

## Referee Comment (RC1) · Anonymous Referee #1 · 3 Mar 2017

**1   General**

- The paper describes the development of an open source framework for the processing of eddy-covariance data. The framework is developed to deal with the wealth of data that will collected within the National Ecological Observatory Network (which is located in the US, a geographical specification not given in the manuscript).

- The paper both addresses the development process as well as some higher level

details of the framework being developed. Little specific information is provided about the actual processing algorithms.

- Three case studies are presented in which the processing framework, in its present state, has been used.

- My general impression is that the work presented in itself is very worthwhile, and -at least in our field- quite novel.

However, I do have some comments:

1. The Development and Operations (DevOps) framework is mentioned as an essential characteristic of the work discussed. However, the DevOps framework is not clearly introduced to the reader (except for one paragraph in the introduction). Even there, most emphasis is placed on the tools, rather than the essential steps or processes that are part of DevOps. I would have expected some sort of a list of steps, or a schematic that shows general characteristics of the workflow in a generic DevOps development process (Wikipedia already shows some colorful examples). Then these generic characteristics could be translated into the specific characteristics of the project that is the subject of the paper. Furthermore, although a number of problems in current practice of EC-data processing are identified (lines 48-62), it remains unclear why DevOps would be the answer.

2. The description of the DevOps framework in section 2 is in fact a description of the collection of tools. It is hard to recognize the DevOps characteristics of it.

3. The description of the tools in section 2 remains rather vague on one hand (2.1, 2.2), and becomes very (too) specific at other points (2.3). For example, to me the message of section 2.1 is that the authors have implemented regular EC-processing software in R. It remains unclear what are the special properties of

this implementation that make it so much better/easier/more flexible/.... than any existing EC-processing toolchain (except that in a number of places terms are used that at least suggest that something special happens in the coding, while it remains unclear what this means in terms of code quality, re-usability etc. e.g. lines 131-134). Similar comments would hold for the other parts of section 2 (see below).

4. The various sections in in section 2 (except 2.1) in my view insufficiently address what the role of the different tools is (on a conceptual level, not an implementation level): who/why are these tools part of the proposed system. Furthermore, some of the sections are not very specific for the proposed application of the tool (section 2.3 could be used in nearly any paper that desribes the use of Docker). The same holds to a large extent for sections 2.2 and 2.4.

5. To summarize my main concern: the paper misses a clear problem statement and as a result it is unclear why the presented software development would be the answer. The paper would be worth reconsidering for publication if the authors would be able to reformulate the paper in such a way that:

- – there is a clear problem statement (which may, or may not be related directly to the NEON network);
  – it is clear that the DevOps methodology is needed to tackle that problem (including a clear general introduction to DevOps, irrespective of the tools used);
  – the this set of proposed tools and methods would enable a DevOps methodology for the task at hand (EC-data processing);
  – the presented cases (section 3) clearly illustrate why a software infrastructure as proposed in the paper is needed.

Below I will provide detailed comments

*Note: in the comments below, the comment is preceded by the line number.*

**2  Detailed comments**

1. 56: I do not see why the ultimate goal should be a universal EC processing environment. As long as the software that is used in papers is described in open literature and the source code is openly available, readers will be able to assess the results of the EC-processing used in the described research. Research groups will always have reasons to do things their own way: because of instrument or site specifics, or due to specific requirements in output and analysis. And from software intercomparisons performed in the past it is quite clear in which aspects the main differences between different processing packages occur (e.g. Mauder et al., 2008; Fratini and Mauder, 2014).

2. 71: it is unclear why DevOps is being embraced: why is this methodology so suited for the problem at hand (the problem is not clearly stated, but I assume that the 'strong need' expressed in lines 66-70 is 'the problem')?

3. 83: indeed, I agree that the fact that the proposed work enables an exact reconstruction of the system that was used to construct certain derived data is an important asset (having the data and the software is -in some cases- insufficient to enable exact reproducibility). On the other hand, in the application at hand, bitwise identical results are usually not needed: the statistical errors in the derived quantities are usually orders of magnitude larger than the differences that occur do to differences in details of the computational environment.

4. 96-106: this 'cycle' suggests some regularity and pace of development. Is that what is intended, or does it simply mean that once someone has a suggestion

for a new feature or new implementation, the code has to go through the cycle? My own experience with code development is that many users are using very different versions of the code (from very old to cutting edge) which may yield impractical surprises when these different users supply new code. The advantage of the -apparently well-funded- NEON network is that paid personal is available to oversee new code submissions.

In this section it is not clearly defined who/what is the 'science community' on one hand, and 'NEON science' on the other hand.

5. As explained in my main comment: section 2.1 seems to mainly discuss 'just another EC-processing tool'. Please focus on those aspects that are new and make it particularly suited for NEON, and for use in the DevOps methodology. My impression is that the direct implementation of the option for parallel processing of EC-data is one important aspect (which may not be relevant if a group only operates a limited number of towers, but which *may* be relevant if in 10 years time NEON decides to reprocess all EC-data of all stations according to the latest insights). But if this parallelization is so central, I would like to see some more explanation/example/tests of it (performance, scalability). Another aspect would be indeed the modularity if it allows -as advertised- for the easy adaptation of new hardware. I suppose that you use some sort of instrument abstraction which makes it possible to describe the properties of any thinkable instrument in such a way that the data can be ingested and processed in a correct way. Again, more details on this potentially unique aspect would make the paper worthwhile to read.

6. 168: indeed git is open source and free, but Github is not (unless you use a public repository). That brings me to the question in which way eddy4R will be open-sourced. Will the github repository be opened up for everyone (by now I cannot find it)? Or will you publish certain stable versions to the public and keep the development closed?

Reliance on Github may seem a risk when planning for the processing of data from a project that will run for 30 years. But since git repositories are stored locally as well, no risk exists in case Github would go out of business.

7. 181: who will do the review and testing? Do you have full-time staff for that? Will the testing be automated in the sense that when the new code is included, all prior test cases should still run without errors, while the new code should also provide it's own test case (if it implements a new feature)?

8. 198: The phrase 'minimal context' suggests that Dockers are small. However, in my experience Docker files are usually quite bulky (as they contain a complete operating system + the software needed to run the scripts). Although storage is not an issue nowadays, the wording suggests something different than what readers might expect.

9. 211: it is unclear to what extent hub.docker.com provides versioning (in the sense that I would be able to exactly reproduce a docker that was produced 5 years ago (with the same Debian version and the same R version with the same package versions). In lines 324 and further it becomes clear that indeed you can specify a version of the docker.

10. I *do* understand that a docker image provides a machine that can do the data processing in a way that is independent of the underlying hardware and software (OS). But you do not specify how this machine (which I still would consider 'virtual') talks to the local file system (outside the docker) to read the raw EC data and write the results. Or is the data flow always supposed to pass through the NEON databases (which are accessed over a network connection).

11. 249: does 'uncompressed' mean 'ASCII', or does it mean 4 or 8 byte binary real values. In the first case a 1:10 compression ratio is not unexpected, in the second case I would be surprised.

12. 251: I am surprised that only the variable name and units are added as meta data. Since an important reason for the proposed methodology is reproducibility and traceability, I would expect that also metadata on the processing itself (software versions, tool chain, processing configurations) would be included. In that way a file with processed EC-data, when 'found in the wild' would still tell the story of how it was produced.

13. 253: I would say that text-based data storage of raw EC-data is already something of the distant past. NetCDF has gained significant usage since 15 years. When properly used, NetCDF files are self-descriptive as well (the same reservation would hold for HDF files: the meta data have to be filled). Otherwise many people use the binary file formats produced by their data logging infrastructure. So again, please indicate the real advantages of the HDF format as compared to others, vis-a-vis the requirements of your application, in combination with the DevOps framework. I could think of two aspects: compression of data (which is not possible in NetCDF) and the hierarchical data structure.

14. 258: in the figure it is unclear if this depicts one file, or multiple files. Furthermore, the terminology ('group', 'Level 1,2,3', 'DPL', . . . .) is not explained. It is insufficient to keep the reader in the dark and just refer to the 'NEON data product naming convention'. To summarize: I do not understand what I am looking at.

15. 268 and related text: in principle, the configuration shown here is not much different from the scripted use of a compiled program (or interpreted script): what is done by the docker images is described in the parameter files. What may be interesting is that the first (left-most) docker-image spawns the second (right-most) images, which are identical in implementation, but serve a different purpose because they are instructed to do so (either the 'turbulence' task or the 'storage' task). Another important asset of the workflow is that apparently meta-data on the processing is included in the HDF files (although line 251 suggests otherwise). It would be of interest to give more details about that (how self-contained are the HDF-files?).

16. 283: I wonder why apparently a single day is used as the unit of storage when it comes to storage of L1-L4 data. In many applications I would think that longer time series are needed. Or does the data-portal glue these daily datafiles together when the data-request to the portal asks for e.g. a full year of data?

17. 284: As it is shown here, the workflow is tightly integrated with the data portal. It seems to be the only use case. But what if a given researcher wants to (re-)process her/his data on her/his desktop for private use only? Is that possible as well? And how would that alter the use of the presented infrastructure?

18. 294-297: this is the first point where I clearly see a reference to DevOps that links the presented software infrastructure to the advantages that DevOps could provide.

19. 300: again, this figure contains a number of unexplained acronyms and terms. Furthermore, I wonder what the added value of this figure is for the paper as a whole. Finally, a part of the text is very hard to read.

20. 303: section 2.6 reads as a user manual, rather than the description of the logic of operation. To me it is unclear why you would need to login to the docker machine (unless you want to develop and test new R-code. On the other hand, figure 5 suggests that the docker-images can be instructed 'from the outside' using parameter files.

21. 341 and further: to me it is unclear what the function of the three case studies is. In the introductory paragraph details on the computing infrastructure is given. This suggests that some benchmarking in terms of performance will be done. However, this only makes sense to me if there is a standard against which the

new setup can be compared (e.g. current practice of reading ASCII files on a single processor machine?).

22. 349 and further: if indeed the performance of the software (in terms of speed, and perhaps also ease of use, flexibility . . . ) is the objective of section 3, I do not see why so much attention needs to be paid to the experimental setup (including photographs!) as well as the interpretation of the results (section 3.1.2). Any EC dataset with a length of 1 to 2 weeks would have sufficed.

23. 381: what kind of additional complexity? It could be that the presented processing infrastructure makes it easy to logically define different processing levels (L1 to L4). In that case it would be of interest to the reader to clarify which are the differences between the various processing levels, and how,conceptually, they can be defined and configured with the current setup. Perhaps this is an important added flexibility? In line 400 it becomes at least clear what is the distinction between L1 and L4.

24. 386: what is the difference between a simple data format and a compound data format. Since the use of the HDF5 file format is such an important aspect, I would expect a clearer description of the way in which the added possibilities of HDF5 files are used (which meta-data, mixes of L1, L2 . . . data . . . .).

25. 389-390: does this mean that the calibration step (to go from L0 to L0p) takes 4.8 PU minutes? Compared to the actual processing this seems long. But perhaps there is a good explanation for it. Please provide the reader with one (or perhaps my interpretation of the numbers is incorrect).

26. 399 and further: only include these results if they show something that only this software can do, and other EC-processing methodologies cannot.

27. 423: the fact that the presented methodology apparently is equally well able to deal with aircraft data as it is can deal with tower data is an interesting added

value, worth advertising! However, I would have appreciated it if the authors would have clearly shown which parts of the processing would work the same for tower and aircraft data, and which steps would be different (e.g. by referring to figure 5). Again, the details of the particular data set (conditions of collection) is less important than the type of data and instruments involved.

28. 442: it is unclear whether the 100Hz → 20Hz reduction is done by the same software or was part of the preparation of the L0 data (i.e. from L(-1) to L0.

29. 445: it is unclear whether the software is able to perform all the corrections that are related to the accelerations of the aircraft (the wind speeds derived from the pressure readings are not the wind speeds). I could imagine that this is part of the L0 → L0p conversion, but this is not specified.

30. 449: it was earlier suggested that the workflow is based on HDF5 files, whereas here gzipped ASCII files are used. I understand that R is able to read all kinds of files, but is reading the ASCII files a standard feature of the methodology or did you first convert the gzipped ASCII files to HDF5 (to produce real L0 data) before ingesting them into the processing software?

31. 451-460: This is an interesting, perhaps non-standard, chain of processing steps. For the reader this would be an interesting example to see how easily this can be configured in your software. Is it just a matter of setting a number of flags in your configuration files? How easily can you change the order of various steps that are applied to high rate data or reduced data?

32. Section 3.2.2: again (like for 3.1.2): presentation of the results is only relevant if your methodology can do something that existing software cannot (or if yours can do it better/faster/easier/more insightful). For instance, if figure 11 would be output that can be produced automatically it could be of added value.
In that respect it would also be interesting to know if the software could be used

in a scenario where certain processing has already been done, and the user decides that he/she wants additional output. Would the software be able to figure out which data (L1... L4) is already there, and which is the minimum set of additional processing steps needed to produce the additional output. Again, in that sense it would be worthwhile for the reader (and potential user) to know which output your software can produce, based on which type of input data (type as in L0, L0p, etc). A well-organized table would be more informative than three case studies. Similarly a clear definition of which types of input data can be ingested would be helpful.

33. 499: this is an interesting application! Does the script automatically select the EddyPro settings that would match the eddy4R settings? If so, providing the difference statistics with your L1-L4 output in the HDF5 file would be useful added value, since users of your L1-L4 data would have an indication how sensitive the resulting aggregated data are to details of the processing (he/she does not know who is wrong and who is right, but a sense of the sensitivity can be helpful).

34. 519: Do you run EddyPro in the same docker as the R-framework? It would be interesting to know what the performance difference (if any) is between EddyPro and your R-based software.

35. 550-560: it is OK for me that in the paper you present the workflow as it is envisaged. But it would be good to already clarify in the main text which steps have already been implemented and which are underway (for the eddy4R part you clearly made this distinction, but for the other parts not). Furthermore, it remains unclear to me to what extent the workflow you propose is strongly tied to the NEON infrastructure. Or could I, as a scientist not working within NEON, be able to adopt your methodology as well. To pose it differently: where does your methodology end and where does the NEON infrastructure start?

**3   Very detailed comments**

1. 63: please add that NEON is being developed in the US.

2. 68: what do you mean by 'tight hardware-software integration'? And which code is 'distributed' and where ?

3. 189: I think figure 3 is superfluous. The message conveyed by the caption could be included in the text. No illustration needed.

4. 277: for people working in the field of micrometeorology (and particularly when involved in CO2 exchange) the terms "turbulence" and "storage" will ring a bell. But for people somewhat more remote, but interested in the proposed collection of tools, it will be less clear (in the context of a strongly IT-oriented paper "storage" might mean something completely different to the reader). For the clarity of the paper the distinction between the turbulence and storage workflows is in fact irrelevant. The message is simply that based on the same code/machinery you can do different things with the same data  Unless you want to show that you can have various parallel workflows which could also interact (but then you have to clarify the potential interactions).

5. 317: I think that figure 7, apparently just a collection of screenshots, is superfluous.

6. 376: it is unclear to me how the current setup would lead to 50 high-rate data streams.

7. 385: the 0.1 to 0.2 Gb: I suppose that this refers to the *input* files (L0 or L0p).

8. 459: the performance is only relevant if it can be compared to something else

9. 460: what do you mean with 'file-backed objects'?

10. 565: what do you mean by defensible?

**References**

Mauder, M., Foken, T., Clement, R., Elbers, J. A., Eugster, W., Gr, T., ... & Kolle, O. (2008). Quality control of CarboEurope flux data–Part 2: Inter-comparison of eddy-covariance software. *Biogeosciences*, *5*, 451-462.

---

## Referee Comment (RC2) · Anonymous Referee #2 · 26 Mar 2017

Metzger et al. describe a data processing framework which is illustrated on ad-hoc examples from NEON's eddy covariance tower and airborne measurement datasets. Overall this technical concept seems potentially valuable for streamlining automation of specific data processing steps from different measurement stations but it is extremely difficult to recognize the broader scientific values in the current version of the paper as written. I must admit that I was rather disappointed to find the description of the tools to be fragmented and poorly supported by the scientific results and conclusions. The whole analysis is very descriptive and in many cases misleading as to what is possible. There is little effort to synthesize what

can be actually learnt from using the tool other than what its potential applications might be in the future. Most importantly, the paper does not specify scientific goals and does not even address the scope of modeling which is what the main interest of the journal's audience is. The manuscript seems to need much work to make the results and discussion useful for the scientific community but could be worthwhile to reconsider after major clarifications. The other reviewer has already provided a useful detailed guidance how this could be achieved and I agree with her/him. I also have other concerns which hopefully can be addressed in the revision.

**General issues:**

1) One fundamental issue in this paper intended for GMDD is that the work is not even connected to any model or modeling framework. The journal scope does not overlap with what paper is about or at least the connection is not made clear. Because there is no model, there is no model version – a requirement of the journal. There are only two words "model" in the whole paper, one of which is included in the last sentence of conclusions but probably in a different meaning: "We hope this framework can serve as a \*model\* for implementing community-sourced, distributed-development scientific code while combatting the deficiencies of current computational frameworks that limit accessibility, reproducibility, and extensibility."

2) It is not apparent how exactly this technical set of workflows adopted by NEON can be useful for a broader scientist/modeler community and what scientific problems it can solve as the idea wraps around different open-source products dedicated essentially to crunching of eddy covariance measurement data. In the abstract, it is promised that the framework is applicable beyond EC but it is completely unclear how. Maybe one way to overcome this issue would be to make a strong connection to a modeling framework where measurement and model outputs are evaluated together or elucidate aspects where this data processing framework would add to novelty and usefulness for the broader GMD community.

3) There are no clear scientific objectives of the paper and the title does not help either "eddy4R: A community-extensible processing, analysis and modeling framework for eddy-covariance data based on R, Git, Docker and HDF5". The use of "modeling framework" is misleading (see also comment 1) because the paper fails to present any modeling or prediction which could be achieved from this framework.

4) The story basically presents a rather ambitious idea of automating data processing including quality control. The latter is not shown yet that it already works well so the product is not yet ready to be fully useful for the community. Once QC is implemented it could be interesting to see how it is done and how flexible the options are for the user. For instance, on page 14 L32 it is concluded "Once scientific QA/QC and uncertainty budget is implemented, the computational expense will likely increase by a factor of two to three. This suggests that eddy4R performs comparably to other flux processors." As presented, the value from another EC flux processor tool is unclear in where it would really help but what is interesting is that the development is directed to a modeling audience who might also be able to use this tool if it was better explained. However, without clearly stated goals and sufficient supporting material to assess its quality and usefulness, it is difficult to evaluate the code framework for all its ambitious features. The paper is incoherent in its presentation (e.g. different components, datasets are presented separately without a clear thread creating multiple fragmented methods and results) and in many places the quality is diverging from the standards of a scientific paper.

**Specific issues**

5) The number of figures seems rather large and not all of them seem necessary. A heavy detail from different settings and configurations (e.g. Sect. 3.1.1, 3.2.1) could be nicely summarized in a table. The examples in Figures 9-13 require specific understanding of eddy covariance and do not help a modeler to adjust the framework for their needs. Even for the eddy covariance community, it might seem surprising that

airborne and tower data can be automatically compared, because there is a comprehensive quality control that needs to be performed on these data and it is not easy to automate such EC comparison at different scales (e.g. Mahrt, 1998), at least without multiple user interactions. For example, in Sect. 3.3 "Validation and verification" it is stated: "eddy4R includes a verification script which automatically processes subsets of the tower and aircraft data introduced in Sect. 3.1 and Sect. 3.2, and verifies the results against a reference, e.g. generated with a different software." Where is this validation shown? Do you actually mean that you duplicate the processing (e.g. also with Eddy Pro) or just check selected files for consistency? The agreement in Figure 13 definitely seems surprising. It almost looks like the same dataset was plotted against the same dataset? The significant figures inconsistently range from 1 to 5. $R^2=1$ is surprisingly good but not too meaningful (did you mean 1.0000, 0.9999 or 0.99?)? I am also confused why the measured variables (e.g. w, q, $CO_2$ mixing ratio) are compared with each other as they should have been the same unless the software interferes with the measurement data.

6) The data quality control does not seem careful. For example, in Figure 9 the periods of latent heat and $CO_2$ flux were not rejected when the friction velocities were at their minima which look definitely below 0.1 m/s on days 114, 115, 116, 119, and other. It is also unclear what the gaps correspond to (rejected data, power interruption). I would be surprised if it was not possible to choose an uninterrupted dataset in the NEON's large EC measurement network. It also seems weird in the same figure that the general temperature trend is anticorrelated with sensible heat flux (doy 115-118). Does the data output use normalized flux units for $CO_2$?

7) The results and discussion also do not focus on the science but rather on what the software can do before the QC/QA are implemented. The QC/QA are the most important component of any data processing, so I am a little bit shocked that this has not been done before the submission and only raw data are reported. It is all about QC/QA so if it does not work well, the whole infrastructure could be in vain. Was it not
possible to wait until the QC steps are implemented? When will it happen?

8) Sect. 3.2.1 "Algorithm setting and profiling". Can you define algorithm setting? There is no model algorithm here. By algorithm you probably mean the data processing routine which deals with technical issues of EC data handling such as "despiking of unphysical data". This and other similar sections can be confusing for the journal readers. It is also unclear what you mean by profiling in this context as it can also have different meanings (I suppose you meant vertical profiling of EC fluxes rather than algorithm profiling). The authors should be careful not to use ambiguous terms and define clearly what they mean by model, algorithm, and other terms where the meaning is not unambiguous in the modeling context.

9) The choice of example figures 10 and 11 is not optimal because they are not well explained or sufficiently informative for the story. The figures are described only superficially what they represent but are not interpreted scientifically. The blue areas represent high deposition of methane? I am not convinced it is fair to show these data without the discussion of uncertainties which are definitely different in airborne and ground fluxes. The results section 3.3.2 are less than a paragraph so it cannot be informative. This should be made more general or explained much better for general audience of GMD.

10) There are other issues which are uncommon to see in a peer-review paper. For example, on Page 7, L225-239 the information is shown as bullet points more like a web-based manual or technical report which almost feels like from a magazine advertising IT Systems. It is interesting, that not even one paper is cited from the GMD community, and majority of the references are authors' own papers published in specialist eddy covariance journals. I think it would make more sense to send the paper to one of those journals or make a better and balanced connection to the GMD literature realm. The EC data handling does seem promising but the approaches vary in various details among the groups so I found the author's EC method review particularly unbalanced.

[Figure]

Overall, I like the general concept of the real-time data/model processing framework, but I expected the paper would be much more than just a teaser of an EC data-processing framework in progress. The revised paper should be guided by clearly defined science question(s) through a coherent story thread throughout the paper. If the intention is to publish the science in GMD, I would strongly recommend the authors to refocus the story on a solid connection between measurement data and modeling. One example could be a model-measurement testbed to validate models on observation data which could be very novel and useful for a larger audience including GMD.

References:

Mahrt L. Flux sampling errors for aircraft and towers. Journal of Atmospheric and Oceanic technology. 1998 Apr;15(2):416-2

---

## Referee Comment (RC3) · Anonymous Referee #3 · 26 Mar 2017

This study present a radically new way to process eddy-covariance data. It combines R-coded EC software that are wrapped in a portable Docker image that can be used on various platforms. It is meant to be scalable and to make use of parallel processing of large quantities of data.

Major comments:

I line with the other reviewers I think that the paper currently lacks a clear scientific question. I could image that for GMD a clear description of a software environment would suffice, but this paper seems to describe "work in progress". I am a big fan of

[Figure]

Docker and directly downloaded the Docker image. I was disappointed in the fact that the image did not contain clear examples (e.g. the three examples outlined in the paper). I could see that the eddy4R.base and eddy4R.qaqc packages were part of the Docker image. I think it is a missed opportunity not to provide examples of (simple) data processing and plotting. Now the advantage of Docker images remains untraceable to the readers and remains rather theoretical. For instance, the HDF5 section (2.4) is clear but a rather standard description that is available on internet (meta-data, directory structure, self-documenting). Again this is a missed opportunity to guide users through an example (download raw data, process the data, and HDF5 output and visualization of results). You want to convince the "traditional ASCII" community. Section 2.5 presents the way NEON wants to deploy Docker images. Again, this remains rather high level, while the stated goal is to "empower the Science community at large by putting the key to the scientific algorithms into the hand of scientists". Again a clear running example in a Docker container would convince these scientists more than a NEON brochure. Section 2.6 would be an ideal starting point for further "Docker-assisted" data analysis, but unfortunately stops at a reference to the eddy4R wiki pages. In section 5 there is a reference to the raw data, but again unfortunately no examples are given in which a Docker image automatically reads, processes, and presents results. In the remainder of the paper, three examples are given, which is basically fine, but without a traceable and "hands-on" exercise does not add much. It is (and should be) part of the standard software testing. In summary, I very much like the concept presented in this paper. However, without more in depth possibilities for potential users of the software, the papers seems more suitable for internal documentation than convincing readers that this is a promising way for the community to process eddy covariance data.

Minor comments

Page 1: line 34: mention where the NEON site is and also where the aircraft data were collected.

Page 1, line 38: "streaming generation of science-grade EC fluxes": please explain better what this means.

Page 6, line 185: current recent

Page 6, Figure 3, introduced at line 191. This hardly adds anything. A link would do here. Also figure 4 and figure 7 seem illustrations that do not add much.

Page 7, line 231: CI?

––––––––––––––––––––––––––––––––

---

## Author Comment (AC1) · 13 Apr 2017

**Author reply to the comment by Executive Editor Astrid Kerkeweg of the manuscript gmd-2016-318**

**"eddy4R: A community-extensible processing, analysis and modeling framework for eddy-covariance data based on R, Git, Docker and HDF5"**

**by S. Metzger et al.**

We thank the Executive Editor for catching our oversight regarding "*The main paper must give the model name and version number (or other unique identifier) in the title.*" In response we intend to adjust the manuscript title accordingly: "eddy4R **v1.0.0**: A community-extensible processing, analysis and modeling framework for eddy-covariance data based on R, Git, Docker and HDF5".

We would value further feedback, and are happy to incorporate additional suggestions.

---

## Author Comment (AC2) · 13 Apr 2017

**"eddy4R: A community-extensible processing, analysis and modeling framework for eddy-covariance data based on R, Git, Docker and HDF5"**

**by S. Metzger et al.**

We thank Anonymous Referee #1 for the valuable feedback on this manuscript. To ensure the comments result in the intended improvements of the manuscript, we outline below our plans for addressing them in a revised version. The comments by the reviewer are recited in italics, followed by our reply in upright font.

We would value further feedback and specification, and are happy to incorporate additional suggestions.
* * *
Reviewer comments

*1 General*

⟩ *The paper describes the development of an open source framework for the processing of eddy-covariance data. The framework is developed to deal with the wealth of data that will collected within the National Ecological Observatory Network (which is located in the US, a geographical specification not given in the manuscript).*

⟩ *The paper both addresses the development process as well as some higher level details of the framework being developed. Little specific information is provided about the actual processing algorithms.*

⟩ *Three case studies are presented in which the processing framework, in its present state, has been used.*

⟩ *My general impression is that the work presented in itself is very worthwhile, and -at least in our field- quite novel.*

Author intentions for revision

We intend to include the requested geographic specification. Please find specific replies to the other points below.
* * *
Reviewer comments

*However, I do have some comments:*

1. *The Development and Operations (DevOps) framework is mentioned as an essential characteristic of the work discussed. However, the DevOps framework is not clearly introduced to the reader (except for one paragraph in the introduction). Even there, most emphasis is placed on the tools, rather than the essential steps or processes that are part of DevOps. I would have expected some sort of a list of steps, or a schematic that shows general characteristics of the workflow in a generic DevOps development process (Wikipedia already shows some colorful examples). Then these generic characteristics could be translated into the specific characteristics of the project that is the subject of the paper. Furthermore, although a number of problems in current practice of EC-data processing are identified (lines 48-62), it remains unclear why DevOps would be the answer.*

2. *The description of the DevOps framework in section 2 is in fact a description of the collection of tools. It is hard to recognize the DevOps characteristics of it.*

3. *The description of the tools in section 2 remains rather vague on one hand (2.1, 2.2), and becomes very (too) specific at other points (2.3). For example, to me the message of section 2.1 is that the authors have implemented regular EC- processing software in R. It remains unclear what are the special properties of this implementation that make it so much better/easier/more flexible/.... than any existing EC-processing toolchain (except that in a number of places terms are used that at least suggest that something special happens in the coding, while it remains unclear what this means in terms of code quality, re-usability etc. e.g. lines 131-134). Similar comments would hold for the other parts of section 2 (see below).*

4. *The various sections in in section 2 (except 2.1) in my view insufficiently address what the role of the different tools is (on a conceptual level, not an implementation level): who/why are these tools part of the proposed system. Furthermore, some of the sections are not very specific for the proposed application of the tool (section 2.3 could be used in nearly any paper that describes the use of Docker). The same holds to a large extent for sections 2.2 and 2.4.*

5. *To summarize my main concern: the paper misses a clear problem statement and as a result it is unclear why the presented software development would be the answer. The paper would be worth reconsidering for publication if the authors would be able to reformulate the paper in such a way that:*
   - *there is a clear problem statement (which may, or may not be related directly to the NEON network);*
   - *it is clear that the DevOps methodology is needed to tackle that problem (including a clear general introduction to DevOps, irrespective of the tools used);*
   - *that this set of proposed tools and methods would enable a DevOps methodology for the task at hand (EC-data processing);*
   - *the presented cases (section 3) clearly illustrate why a software infrastructure as proposed in the paper is needed.*

Author intentions for revision

We agree with the reviewer that more clarity is needed on the problem statement and exactly how the DevOps approach in general and specific tools implemented in the paper address the needs of the EC community. To address these concerns, we intend to make the following edits to the manuscript:

) In the introduction, explicitly identify the problem this paper addresses: The EC community currently lacks the capacity to develop & deploy complex algorithms collaboratively created by scientists in an efficient and scalable manner.

) Add a section presenting a general and more detailed introduction to DevOps and explicitly link the specific workflow steps presented in Figure 1 to DevOps concepts. Throughout the manuscript, we intend to tie these concepts and tools back to the problem statement to clarify how they solve the accessibility, extensibility, and reproducibility problems limiting EC software use and development.

) In each of the use cases presented in section 3, we intend to add discussion of the advantages of employing the DevOps approach, and how they address the paper's problem statement.

Reviewer comments

*Below I will provide detailed comments*

*Note: in the comments below, the comment is preceded by the line number.*

*2 Detailed comments*

1. *56: I do not see why the ultimate goal should be a universal EC processing environment. As long as the software that is used in papers is described in open literature and the source code is openly available, readers will be able to assess the results of the EC-processing used in the described research. Research groups will always have reasons to do things their own way: because of instrument or site specifics, or due to specific requirements in output and analysis. And from software intercomparisons performed in the past it is quite clear in which aspects the main differences between different processing packages occur (e.g. Mauder et al., 2008; Fratini and Mauder, 2014).*

Author intentions for revision

We intend to clarify in the manuscript: By universal processing environment, we mean the capability of processing software to be reliably and consistently applied at scale across most, if not all, computer platforms. As well as the flexibility to incorporate additional data streams and processing workflows. This would better allow research groups to tailor existing software to their needs instead of re-creating code or kludging together multiple software outputs to realize an algorithmic chain for data analytics. Even though source code is available, it does not guarantee reproducibility: The subsequent sentence in the manuscript (l. 58) points out that 50% of published scientific code cannot even be run due to missing software dependencies.

Reviewer comments

2.  *71: it is unclear why DevOps is being embraced: why is this methodology so suited for the problem at hand (the problem is not clearly stated, but I assume that the 'strong need' expressed in lines 66-70 is 'the problem')?*

3.  *83: indeed, I agree that the fact that the proposed work enables an exact reconstruction of the system that was used to construct certain derived data is an important asset (having the data and the software is -in some cases- insufficient to enable exact reproducibility). On the other hand, in the application at hand, bit- wise identical results are usually not needed: the statistical errors in the derived quantities are usually orders of magnitude larger than the differences that occur do to differences in details of the computational environment.*

4.  *96-106: this 'cycle' suggests some regularity and pace of development. Is that what is intended, or does it simply mean that once someone has a suggestion for a new feature or new implementation, the code has to go through the cycle? My own experience with code development is that many users are using very different versions of the code (from very old to cutting edge) which may yield impractical surprises when these different users supply new code. The advantage of the -apparently well-funded- NEON network is that paid personal is available to oversee new code submissions. In this section it is not clearly defined who/what is the 'science community' on one hand, and 'NEON science' on the other hand.*

5.  *As explained in my main comment: section 2.1 seems to mainly discuss 'just another EC-processing tool'. Please focus on those aspects that are new and make it particularly suited for NEON, and for use in the DevOps methodology. My impression is that the direct implementation of the option for parallel processing of EC-data is one important aspect (which may not be relevant if a group only operates a limited number of towers, but which may be relevant if in 10 years time NEON decides to reprocess all EC-data of all stations according to the latest insights). But if this parallelization is so central, I would like to see some more explanation/example/tests of it (performance, scalability). Another aspect would be indeed the modularity if it allows -as advertised- for the easy adaptation of new hardware. I suppose that you use some sort of instrument abstraction which makes it possible to describe the properties of any thinkable instrument in such a way that the data can be ingested and processed in a correct way. Again, more details on this potentially unique aspect would make the paper worthwhile to read.*

Author intentions for revision

We intend to utilize these specifications as concrete areas in the manuscript for revision per our replies to the general comments, and to utilize / address the provided examples to achieve the suggested content extension.

Reviewer comments

6.  *168: indeed git is open source and free, but Github is not (unless you use a public repository). That brings me to the question in which way eddy4R will be open-sourced. Will the github repository be opened up for everyone (by now I cannot find it)? Or will you publish certain stable versions to the public and keep the development closed? Reliance on Github may seem a risk when planning for the processing of data from a project that will run for 30 years. But since git repositories are stored locally as well, no risk exists in case Github would go out of business.*

Author intentions for revision

We intend to clarify in the manuscript: The current, stable snapshot of the GitHub repository is published and archived incl. DOI (links provided in Sect. 5 "Code and data availability"). As NEON Construction completes, the GitHub repository itself will be made publicly accessible. The central component with regard to contingency planning is distributed version control, of which Git is one implementation. Should Git fail in the next 30 years, it can be replaced with another version control system.

Reviewer comments

7.  *181: who will do the review and testing? Do you have full-time staff for that? Will the testing be automated in the sense that when the new code is included, all prior test cases should still run without errors, while the new code should also provide it's own test case (if it implements a new feature)?*

8.  *198: The phrase 'minimal context' suggests that Dockers are small. However, in my experience Docker files are usually quite bulky (as they contain a complete operating system + the software needed to run the scripts). Although storage is not an issue nowadays, the wording suggests something different than what readers might expect.*

9.  *211: it is unclear to what extent hub.docker.com provides versioning (in the sense that I would be able to exactly reproduce a docker that was produced 5 years ago (with the same Debian version and the same R version with the same package versions). In lines 324 and further it becomes clear that indeed you can specify a version of the docker.*

10. *I do understand that a docker image provides a machine that can do the data processing in a way that is independent of the underlying hardware and software (OS). But you do not specify how this machine (which I still would consider 'virtual') talks to the local file system (outside the docker) to read the raw EC data and write the results. Or is the data flow always supposed to pass through the NEON databases (which are accessed over a network connection).*

11. *249: does 'uncompressed' mean 'ASCII', or does it mean 4 or 8 byte binary real values. In the first case a 1:10 compression ratio is not unexpected, in the second case I would be surprised.*

12. *251: I am surprised that only the variable name and units are added as meta data. Since an important reason for the proposed methodology is reproducibility and traceability, I would expect that also metadata on the processing itself (software versions, tool chain,*

*processing configurations) would be included. In that way a file with processed EC-data, when 'found in the wild' would still tell the story of how it was produced.*

13. *253: I would say that text-based data storage of raw EC-data is already something of the distant past. NetCDF has gained significant usage since 15 years. When properly used, NetCDF files are self-descriptive as well (the same reservation would hold for HDF files: the meta data have to be filled). Otherwise many people use the binary file formats produced by their data logging infrastructure. So again, please indicate the real advantages of the HDF format as compared to others, vis-a-vis the requirements of your application, in combination with the DevOps framework. I could think of two aspects: compression of data (which is not possible in NetCDF) and the hierarchical data structure.*

14. *258: in the figure it is unclear if this depicts one file, or multiple files. Furthermore, the terminology ('group', 'Level 1,2,3', 'DPL', . . . .) is not explained. It is insufficient to keep the reader in the dark and just refer to the 'NEON data product naming convention'. To summarize: I do not understand what I am looking at.*

15. *268 and related text: in principle, the configuration shown here is not much different from the scripted use of a compiled program (or interpreted script): what is done by the docker images is described in the parameter files. What may be interesting is that the first (left-most) docker-image spawns the second (right-most) images, which are identical in implementation, but serve a different purpose be- cause they are instructed to do so (either the 'turbulence' task or the 'storage' task). Another important asset of the workflow is that apparently meta-data on the processing is included in the HDF files (although line 251 suggests otherwise). It would be of interest to give more details about that (how self-contained are the HDF-files?).*

16. *283: I wonder why apparently a single day is used as the unit of storage when it comes to storage of L1-L4 data. In many applications I would think that longer time series are needed. Or does the data-portal glue these daily datafiles together when the data-request to the portal asks for e.g. a full year of data?*

17. *284: As it is shown here, the workflow is tightly integrated with the data portal. It seems to be the only use case. But what if a given researcher wants to (re-)process her/his data on her/his desktop for private use only? Is that possible as well? And how would that alter the use of the presented infrastructure?*

18. *294-297: this is the first point where I clearly see a reference to DevOps that links the presented software infrastructure to the advantages that DevOps could provide.*

19. *300: again, this figure contains a number of unexplained acronyms and terms. Furthermore, I wonder what the added value of this figure is for the paper as a whole. Finally, a part of the text is very hard to read.*

20. *303: section 2.6 reads as a user manual, rather than the description of the logic of operation. To me it is unclear why you would need to login to the docker machine (unless you want to develop and test new R-code. On the other hand, figure 5 suggests that the docker-images can be instructed 'from the outside' using parameter files.*

21. *341 and further: to me it is unclear what the function of the three case studies is. In the introductory paragraph details on the computing infrastructure is given. This suggests that some benchmarking in terms of performance will be done. However, this only makes*

*sense to me if there is a standard against which the new setup can be compared (e.g. current practice of reading ASCII files on a single processor machine?).*

22. *349 and further: if indeed the performance of the software (in terms of speed, and perhaps also ease of use, flexibility . . . ) is the objective of section 3, I do not see why so much attention needs to be paid to the experimental setup (including photographs!) as well as the interpretation of the results (section 3.1.2). Any EC dataset with a length of 1 to 2 weeks would have sufficed.*

23. *381: what kind of additional complexity? It could be that the presented process- ing infrastructure makes it easy to logically define different processing levels (L1 to L4). In that case it would be of interest to the reader to clarify which are the differences between the various processing levels, and how, conceptually, they can be defined and configured with the current setup. Perhaps this is an important added flexibility? In line 400 it becomes at least clear what is the distinction between L1 and L4.*

24. *386: what is the difference between a simple data format and a compound data format. Since the use of the HDF5 file format is such an important aspect, I would expect a clearer description of the way in which the added possibilities of HDF5 files are used (which meta-data, mixes of L1, L2 . . . data . . . .).*

25. *389-390: does this mean that the calibration step (to go from L0 to L0p) takes 4.8 PU minutes? Compared to the actual processing this seems long. But perhaps there is a good explanation for it. Please provide the reader with one (or perhaps my interpretation of the numbers is incorrect).*

26. *399 and further: only include these results if they show something that only this software can do, and other EC-processing methodologies cannot.*

27. *423: the fact that the presented methodology apparently is equally well able to deal with aircraft data as it is can deal with tower data is an interesting added value, worth advertising! However, I would have appreciated it if the authors would have clearly shown which parts of the processing would work the same for tower and aircraft data, and which steps would be different (e.g. by referring to figure 5). Again, the details of the particular data set (conditions of collection) is less important than the type of data and instruments involved.*

28. *442: it is unclear whether the 100Hz    20Hz reduction is done by the same software or was part of the preparation of the L0 data (i.e. from L(-1) to L0.*

29. *445: it is unclear whether the software is able to perform all the corrections that are related to the accelerations of the aircraft (the wind speeds derived from the pressure readings are not the wind speeds). I could imagine that this is part of the L0    L0p conversion, but this is not specified.*

30. *449: it was earlier suggested that the workflow is based on HDF5 files, whereas here gzipped ASCII files are used. I understand that R is able to read all kinds of files, but is reading the ASCII files a standard feature of the methodology or did you first convert the gzipped ASCII files to HDF5 (to produce real L0 data) before ingesting them into the processing software?*

31. *451-460: This is an interesting, perhaps non-standard, chain of processing steps. For the reader this would be an interesting example to see how easily this can be configured in your software. Is it just a matter of setting a number of flags in your configuration*

*files? How easily can you change the order of various steps that are applied to high rate data or reduced data?*

32. *Section 3.2.2: again (like for 3.1.2): presentation of the results is only relevant if your methodology can do something that existing software cannot (or if yours can do it better/faster/easier/more insightful). For instance, if figure 11 would be output that can be produced automatically it could be of added value. In that respect it would also be interesting to know if the software could be used in a scenario where certain processing has already been done, and the user decides that he/she wants additional output. Would the software be able to figure out which data (L1. . . L4) is already there, and which is the minimum set of additional processing steps needed to produce the additional output. Again, in that sense it would be worthwhile for the reader (and potential user) to know which output your software can produce, based on which type of input data (type as in L0, L0p, etc). A well-organized table would be more informative than three case studies. Similarly a clear definition of which types of input data can be ingested would be helpful.*

33. *499: this is an interesting application! Does the script automatically select the EddyPro settings that would match the eddy4R settings? If so, providing the difference statistics with your L1-L4 output in the HDF5 file would be useful added value, since users of your L1-L4 data would have an indication how sensitive the resulting aggregated data are to details of the processing (he/she does not know who is wrong and who is right, but a sense of the sensitivity can be helpful).*

34. *519: Do you run EddyPro in the same docker as the R-framework? It would be interesting to know what the performance difference (if any) is between EddyPro and your R-based software.*

35. *550-560: it is OK for me that in the paper you present the workflow as it is envisaged. But it would be good to already clarify in the main text which steps have already been implemented and which are underway (for the eddy4R part you clearly made this distinction, but for the other parts not). Furthermore, it re- mains unclear to me to what extent the workflow you propose is strongly tied to the NEON infrastructure. Or could I, as a scientist not working within NEON, be able to adopt your methodology as well. To pose it differently: where does your methodology end and where does the NEON infrastructure start?*

*3 Very detailed comments*

1. *63: please add that NEON is being developed in the US.*
2. *68: what do you mean by 'tight hardware-software integration'? And which code is 'distributed' and where?*
3. *189: I think figure 3 is superfluous. The message conveyed by the caption could be included in the text. No illustration needed.*
4. *277: for people working in the field of micrometeorology (and particularly when involved in CO2 exchange) the terms "turbulence" and "storage" will ring a bell. But for people somewhat more remote, but interested in the proposed collection of tools, it will be less clear (in the context of a strongly IT-oriented paper "storage" might mean something completely different to the reader). For the clarity of the paper the distinction between the turbulence and storage workflows is in fact irrelevant. The message is*

*simply that based on the same code/machinery you can do different things with the same data Unless you want to show that you can have various parallel workflows which could also interact (but then you have to clarify the potential interactions).*

5. *317: I think that figure 7, apparently just a collection of screenshots, is superfluous.*
6. *376: it is unclear to me how the current setup would lead to 50 high-rate data streams.*
7. *385: the 0.1 to 0.2 Gb: I suppose that this refers to the input files (L0 or L0p).*
8. *459: the performance is only relevant if it can be compared to something else*
9. *460: what do you mean with 'file-backed objects'?*
10. *565: what do you mean by defensible?*

*References*

*Mauder, M., Foken, T., Clement, R., Elbers, J. A., Eugster, W., Gr, T., ... & Kolle, O. (2008). Quality control of CarboEurope flux data–Part 2: Inter-comparison of eddy- covariance software. Biogeosciences, 5, 451-462.*

Author intentions for revision

Thank you for your thorough review. We intend to utilize these specifications as concrete areas in the manuscript for revision per our replies to the general comments, and to utilize / address the provided examples to achieve the suggested content extension. We will certainly work to address all the smaller comments, but we wanted to reply to your bigger concerns before beginning the work to ensure our plan of action is satisfactory.

---

## Author Comment (AC3) · 13 Apr 2017

please find replies in supplement

Please also note the supplement to this comment:
http://www.geosci-model-dev-discuss.net/gmd-2016-318/gmd-2016-318-AC3-supplement.pdf

---

## Author Comment (AC4) · 13 Apr 2017

**"eddy4R: A community-extensible processing, analysis and modeling framework for eddy-covariance data based on R, Git, Docker and HDF5"**

**by S. Metzger et al.**

We thank Anonymous Referee #3 for the valuable feedback on this manuscript. To ensure the comments result in the intended improvements of the manuscript, we outline below our plans for addressing them in a revised version. The comments by the reviewer are recited in italics, followed by our reply in upright font.

We would value further feedback and specification, and are happy to incorporate additional suggestions.

| |
|---|
| Reviewer comments |
| *This study present a radically new way to process eddy-covariance data. It combines R-coded EC software that are wrapped in a portable Docker image that can be used on various platforms. It is meant to be scalable and to make use of parallel processing of large quantities of data.* |
| Author intentions for revision |
| Many thanks for this succinct summary. |

| |
|---|
| Reviewer comments |
| *Major comments* |
| *In line with the other reviewers, I think that the paper currently lacks a clear scientific question. I could image that for GMD a clear description of a software environment would suffice, but this paper seems to describe "work in progress".* |
| Author intentions for revision |
| As stated by the reviewer, the aim of manuscript is to introduce the novel eddy4R-Docker software framework to address a methodological rather than scientific question: the portable, reproducible and extensible processing of eddy-covariance data. For this reason, the GMD journal was chosen, and three examples of geoscientific applications are provided in favor of a single in-depth scientific survey. One core component of GMD model description papers is "…evaluation against standard benchmarks…" which is addressed in Sect. 3.3. To |

demonstrate completion of the v1.0.0 development stage we intend to include an application example as suggested below by the reviewer.

Reviewer comments

*I am a big fan of Docker and directly downloaded the Docker image. I was disappointed in the fact that the image did not contain clear examples (e.g. the three examples outlined in the paper). I could see that the eddy4R.base and eddy4R.qaqc packages were part of the Docker image. I think it is a missed opportunity not to provide examples of (simple) data processing and plotting. Now the advantage of Docker images remains untraceable to the readers and remains rather theoretical.*

Author intentions for revision

We could not agree more with the reviewer in that an application example would add much value for the reader and potential user. For this reason, we intend to include an R-vignette example of a (simple) data read-in, processing and plotting workflow. This example will utilize the functionality of both R-packages presented here, eddy4R.base and eddy4R.qaqc.

Reviewer comments

*For instance, the HDF5 section (2.4) is clear but a rather standard description that is available on internet (meta-data, directory structure, self-documenting). Again, this is a missed opportunity to guide users through an example (download raw data, process the data, and HDF5 output and visualization of results). You want to convince the "traditional ASCII" community.*

Author intentions for revision

Agreed. The R-vignette example will include HDF5 read-in, write-out examples.

Reviewer comments

*Section 2.5 presents the way NEON wants to deploy Docker images. Again, this remains rather high level, while the stated goal is to "empower the Science community at large by putting the key to the scientific algorithms into the hand of scientists". Again, a clear running example in a Docker container would convince these scientists more than a NEON brochure.*

Author intentions for revision

We intend to address this concern through the R-vignette example.

Reviewer comments

*Section 2.6 would be an ideal starting point for further "Docker-assisted" data analysis, but unfortunately stops at a reference to the eddy4R wiki pages.*

Author intentions for revision

In response to the reviewer suggestion, we intend to introduce the R-vignette example "Docker-assisted" data analysis in Section 2.6.

Reviewer comments

*In section 5 there is a reference to the raw data, but again unfortunately no examples are given in which a Docker image automatically reads, processes, and presents results. In the remainder of the paper, three examples are given, which is basically fine, but without a traceable and "hands-on" exercise does not add much. It is (and should be) part of the standard software testing.*

Author intentions for revision

We intend to address this concern through the R-vignette example.

Reviewer comments

*In summary, I very much like the concept presented in this paper. However, without more in depth possibilities for potential users of the software, the papers seems more suitable for internal documentation than convincing readers that this is a promising way for the community to process eddy covariance data.*

Author intentions for revision

We intend to address this concern through the R-vignette example.

Reviewer comments

*Minor comments*

*Page 1: line 34: mention where the NEON site is and also where the aircraft data were collected.*

*Page 1, line 38: "streaming generation of science-grade EC fluxes": please explain better what this means.*

*Page 6, line 185: current recent*

*Page 6, Figure 3, introduced at line 191. This hardly adds anything. A link would do here. Also figure 4 and figure 7 seem illustrations that do not add much.*

*Page 7, line 231: CI?*

Author intentions for revision

We will certainly work to address all the smaller comments, but we wanted to reply to your bigger concerns before beginning the work to ensure our plan of action is satisfactory.

---

## Author Comment (AC5) · 20 Apr 2017

**"eddy4R: A community-extensible processing, analysis and modeling framework for eddy-covariance data based on R, Git, Docker and HDF5"**

**by S. Metzger et al.**

We thank Anonymous Referee #2 for the valuable feedback on this manuscript. With regard to one reviewer comment, we would like to provide additional clarification to our earlier response from 2017-04-13. The comments by the reviewer are recited in italics, followed by our reply from 2017-04-13 in upright font, and additional clarification in blue font.

We would value further feedback and specification, and are happy to incorporate additional suggestions.
* * *
Reviewer comments

*General issues:*

*1) One fundamental issue in this paper intended for GMDD is that the work is not even connected to any model or modeling framework. The journal scope does not overlap with what paper is about or at least the connection is not made clear. Because there is no model, there is no model version – a requirement of the journal. There are only two words "model" in the whole paper, one of which is included in the last sentence of conclusions but probably in a different meaning: "We hope this framework can serve as a \*model\* for implementing community-sourced, distributed-development scientific code while combatting the deficiencies of current computational frameworks that limit accessibility, reproducibility, and extensibility."*
* * *
Author intentions for revision (2017-04-13)

The authors considered several journals before deciding where to submit our manuscript, and we came to this decision through taking into account the manuscript types requested on the Geoscientific Model Development (GMD) webpage. Specifically, we felt that our paper provides "…utility tools … such as coupling frameworks … with a geoscientific application".

We intend to clarify in the manuscript: The framework provides modular processing for surface-atmosphere exchange data with quality assurance and quality control as foundation for modelling exercises such as the application example in Sect. 3.2. This includes footprint modeling (GMD: Kljun et al., 2015), evaluation of large eddy simulations (GMD: Maronga et al., 2015), machine learning etc. The result is an end-to-end framework for model building,

parameterization and assessment considering the large amounts of theoretical assumptions in eddy-covariance technique that require corrections to the data. The combination of these tools to address the concern of reproducibility was a major consideration when submitting to GMD.

Per suggestion of referee #2 as well as the executive editor, in addition to Sect. 5 Code and data availability we will include the eddy4R-Docker framework version (0.1.0) also in the manuscript title.

References:

Kljun, N., Calanca, P., Rotach, M. W., and Schmid, H. P.: A simple two-dimensional parameterisation for Flux Footprint Prediction (FFP), Geosci. Model Dev., 8, 3695-3713, doi:10.5194/gmd-8-3695-2015, 2015.

Maronga, B., Gryschka, M., Heinze, R., Hoffmann, F., Kanani-Sühring, F., Keck, M., Ketelsen, K., Letzel, M. O., Sühring, M., and Raasch, S.: The Parallelized Large-Eddy Simulation Model (PALM) version 4.0 for atmospheric and oceanic flows: model formulation, recent developments, and future perspectives, Geosci. Model Dev., 8, 2515-2551, doi:10.5194/gmd-8-2515-2015, 2015.

Additional clarification by the authors (2017-04-20)

We further intend to clarify in the revised manuscript that eddy-covariance data processing consists of employing a sequence of models. These often originate from scientific sub-fields with corresponding publications, and eddy4R-Docker provides an integrative, yet modular and extensible framework for their concerted application and continued development. In its current form eddy4R-Docker v1.0.0 encompasses the following models: plausibility tests (Taylor and Loescher, 2013), de-spiking (Brock, 1986), lag correction, data aggregation, and QA/QC budgeting (Smith et al., 2014).

Additional models are in preparation for future extension of the eddy4R-Docker framework presented here: coordinate rotation (Wilczak et al., 2001), spectral correction (Nordbo and Katul, 2012), turbulent mixing and stationarity (Foken and Wichura, 1996), detection limit (Billesbach, 2011), turbulent sampling error (Lenschow et al., 1994), footprint analysis (Kljun et al., 2015), storage flux term, and uncertainty budgeting.

Please note that e.g. Kljun et al. (2015) is itself published in GMD.

References:

Billesbach, D. P.: Estimating uncertainties in individual eddy covariance flux measurements: A comparison of methods and a proposed new method, Agric. For. Meteorol., 151, 394-405, doi:10.1016/j.agrformet.2010.12.001, 2011.

Brock, F. V.: A nonlinear filter to remove impulse noise from meteorological data, J. Atmos. Oceanic Technol., 3, 51-58, doi:10.1175/1520-0426(1986)003<0051:anftri>2.0.co;2, 1986.

Foken, T., and Wichura, B.: Tools for quality assessment of surface-based flux measurements, Agric. For. Meteorol., 78, 83-105, doi:10.1016/0168-1923(95)02248-1, 1996.

Kljun, N., Calanca, P., Rotach, M. W., and Schmid, H. P.: A simple two-dimensional parameterisation for Flux Footprint Prediction (FFP), Geosci. Model Dev., 8, 3695-3713, doi:10.5194/gmd-8-3695-2015, 2015.

Lenschow, D. H., Mann, J., and Kristensen, L.: How long is long enough when measuring fluxes and other turbulence statistics?, J. Atmos. Oceanic Technol., 11, 661-673, doi:10.1175/1520-0426(1994)011<0661:HLILEW>2.0.CO;2, 1994.

Nordbo, A., and Katul, G.: A wavelet-based correction method for eddy-covariance high-frequency losses in scalar concentration measurements, Boundary Layer Meteorol., 146, 81-102, doi:10.1007/s10546-012-9759-9, 2012.

Smith, D. E., Metzger, S., and Taylor, J. R.: A transparent and transferable framework for tracking quality information in large datasets, PLoS One, 9, e112249, doi:10.1371/journal.pone.0112249, 2014.

Taylor, J. R., and Loescher, H. L.: Automated quality control methods for sensor data: A novel observatory approach, Biogeosciences, 10, 4957-4971, doi:10.5194/bg-10-4957-2013, 2013.

Wilczak, J. M., Oncley, S. P., and Stage, S. A.: Sonic anemometer tilt correction algorithms, Boundary Layer Meteorol., 99, 127-150, doi:10.1023/A:1018966204465, 2001.

---

## Editor Comment (EC1) · C. van Heerwaarden (Editor) · 30 Apr 2017

The paper has received very detailed reviews with excellent suggestions to improve the paper. The reviewers see the novelty an applicability of the presented framework, but demand improvements in order to accept the paper.

The authors have given excellent answers to the reviewers' concerns and by incorporating their suggested improvements, this paper will most likely meet the standards of GMD. Important here is that the reviewers provide a working example as requested by the third reviewer. This will greatly improve the quality of the paper.

I suggest that the authors proceed with improving the manuscript.

---

## Author Comment (AC6) · 3 Jul 2017

**Author reply to the comment by Executive Editor Astrid Kerkeweg of the manuscript**

**gmd-2016-318**

**"eddy4R: A community-extensible processing, analysis and modeling framework for eddy-covariance data based on R, Git, Docker and HDF5"**

**by S. Metzger et al.**

We thank the Executive Editor for catching our oversight regarding "*The main paper must give the model name and version number (or other unique identifier) in the title.*" In response we have adjusted the manuscript title accordingly: "eddy4R **0.2.0**: A DevOps model for community-extensible processing and analysis of eddy-covariance data based on R, Git, Docker and HDF5".

---

## Author Comment (AC7) · 3 Jul 2017

**Author reply to the comments by Anonymous Referee #1 of the manuscript gmd-2016-318**

**"eddy4R: A community-extensible processing, analysis and modeling framework for eddy-covariance data based on R, Git, Docker and HDF5"**

**by S. Metzger et al.**

We thank Anonymous Referee #1 for the valuable feedback, which helped to improve the manuscript. Please find below the Referee comments recited in *blue, italics font*, followed by our point-by-point replies and corresponding changes in the manuscript in black, upright font.

**1 General**

- The paper describes the development of an open source framework for the processing of eddy-covariance data. The framework is developed to deal with the wealth of data that will collected within the National Ecological Observatory Network (which is located in the US, a geographical specification not given in the manuscript).
- The paper both addresses the development process as well as some higher level details of the framework being developed. Little specific information is provided about the actual processing algorithms.
- Three case studies are presented in which the processing framework, in its present state, has been used.
- *My* general impression is that the work presented in itself is very worthwhile, and -at least in our field- quite novel.

Author reply: We thank the Referee for acknowledging the significance of this work.

Changes in the manuscript: We have performed multiple changes resulting from these comments, which are detailed in response to the specific recommendations below.

**However, I do have some comments:**

1. The Development and Operations (DevOps) framework is mentioned as an essential characteristic of the work discussed. However, the DevOps framework is not clearly introduced to the reader (except for one paragraph in the introduction). Even there, most emphasis is placed on the tools, rather than the essential steps or processes that are part of DevOps. I would have expected some sort of a list of steps, or a schematic that shows

general characteristics of the workflow in a generic DevOps development process (Wikipedia already shows some colorful examples). Then these generic characteristics could be translated into the specific characteristics of the project that is the subject of the paper. Furthermore, although a number of problems in current practice of EC-data processing are identified (lines 48-62), it remains unclear why DevOps would be the answer.

Author reply: We agree that the generic DevOps methodology was not introduced in sufficient clarity to the reader: an overview of DevOps steps and how they translate into the specific characteristics of the presented software framework is needed. In addition, we agree that clarification is needed on the problem statement of the paper (see Referee comment 5 below) and why DevOps is the answer.

Changes in the manuscript: To address these points, we added text and a figure in Sect. 2. These provide an overview of the DevOps methodology, and throughout the paper we now tie the eddy4R-Docker software development model to it. The adjusted text reads:

"Figure 1. Stages of the general DevOps workflow (source: Kharnagy via Wikimedia Commons [CC BY-SA 4.0]).

DevOps promotes collaboration and tight integration between software development, testing, and operational deployment by following a core workflow (e.g., Wurster et al., 2015): Plan, Create, Verify, Package, Release, Configure, and Monitor. The text below describes these stages and Figure 1 shows the general sequence and overlap of these stages between software developers (Dev) and operators (Ops).

**Plan** involves focusing and prioritizing new software features or capabilities based on their enhancement of value. **Create** is the activity of designing and writing the code that delivers a new feature. **Verify** tests the new software feature against established standards for accuracy and performance (e.g. does it unexpectedly alter the output of pre-existing features? Does it produce the expected result?). **Package** involves the compilation of the code once it is ready for deployment, including all data and software dependencies, and gathers necessary approvals. The **Release** stage deploys the software into production. **Configure** involves supplying and configuring the IT infrastructure required to operate the code at scale, including storage, database operations, and networking. Finally, **Monitor** observes and tracks the use, performance, and end-user impact of the release.

Variants of this workflow exist (e.g., Chen, 2015), but the general components and sequence are retained. In addition, there is no single set of tools accompanying the DevOps approach. Rather, many tools exist that facilitate the execution of one or more of these workflow steps, often through automation.

NEON's DevOps framework consists of a periodic sequence (Figure 2) that incorporates these workflow steps: The science community contributes algorithms and best practices (1). Implicitly or explicitly, this embodies the DevOps: Plan stage – the algorithms most valued by the community are being incorporated. Together with NEON Science (2), these algorithms are coded in the open-source R computational environment (DevOps: Create stage). DevOps: Verify (testing) and Package (packaging) are performed as the code is compiled into eddy4R packages via the GitHub distributed version control system (3). NEON Science releases an eddy4R version from GitHub, which automatically builds an eddy4R-Docker image on DockerHub as specified in a "Dockerfile" (4; DevOps: Release stage). The eddy4R-Docker image is immediately available for deployment by NEON Cyberinfrastructure (CI; 5; DevOps: Configure & Monitor stages), the Science Community (1) and NEON Science (2) alike. Here the DevOps: Configure (computational resource allocation) & Monitor stages occur. Monitoring of end-user experience is also performed in GitHub (3) via issue-tracking.

Figure 2. NEON-specific DevOps workflow. DevOps workflow steps are called out in parentheses. Please see text in Sect. 2 for detailed explanation."

In addition, we added text throughout Sects. 2.1 - 2.5, referencing the DevOps workflow steps that each presented tool facilitates.

To address why the DevOps methodology is the answer to some of the problems plaguing EC data processing, we first clarified the problem statement in the introduction: "The question we ask in this paper is: How do we collaboratively create portable, reproducible, open-source, scalable, and extensible software that improves reliability and comparability of eddy covariance data products?"

Then, we added the following text later in Sect. 2: "Thus, the DevOps model serves as the framework within which the scientific community can efficiently and robustly collaborate to produce, manage, and iterate community software. Through choosing appropriate tools to implement the DevOps workflow steps, the reproducibility, scalability and extensibility needs of software development communities (including EC) can be met."

**References:**

Chen, L.: Continuous delivery: Huge benefits, but challenges too, IEEE Softw., 32, 50-54, doi:10.1109/ms.2015.27, 2015.

Wurster, L. F., Colville, R. J., and Duggan, J.: Market Trends: DevOps - not a market, but a tool-centric philosophy that supports a continuous delivery value chain, Gartner, Inc., G00274555, Stamford, U.S.A., 14 pp., 2015.

2. The description of the DevOps framework in section 2 is in fact a description of the collection of tools. It is hard to recognize the DevOps characteristics of it.

Author reply: We agree, and believe that the addition of the general DevOps overview and translation into the software development model presented in the paper as discussed above addresses this Referee comment.

Changes in the manuscript: Please see the response to the Referee comment 1 for changes made in the manuscript.

3. The description of the tools in section 2 remains rather vague on one hand (2.1, 2.2), and becomes very (too) specific at other points (2.3). For example, to me the message of section 2.1 is that the authors have implemented regular EC- processing software in R. It remains unclear what are the special properties of this implementation that make it so much better/easier/more flexible/.... than any existing EC-processing toolchain (except that in a number of places terms are used that at least suggest that something special happens in the coding, while it remains unclear what this means in terms of code quality, re-usability etc. e.g. lines 131-134). Similar comments would hold for the other parts of section 2 (see below).

Author reply: The level of detail provided in Sects. 2.1 - 2.5 was tailored to call out the particular elements of the tools that addressed the portability, reproducibility, and extensibility needs of the EC community. For example, in Sec. 2.1 (previously lines 132-

134), we stated "Following best practices, eddy4R is written in controlled and strictly hierarchical terminology consisting of base names and modifiers, which ensures modular extensibility over time." We agree that this was not clear to the reader as a result of an ambiguous problem statement and an insufficient overview of DevOps and how DevOps was translated into the specific software development model presented in the paper (addressed in response to Referee comment 1 above). We believe the changes made in response to these points now highlight the special properties of this implementation that improve upon existing EC processing toolchains.

Changes in the manuscript: Please see changes made in response to Referee comment 1.

4. The various sections in in section 2 (except 2.1) in my view insufficiently address what the role of the different tools is (on a conceptual level, not an implementation level): who/why are these tools part of the proposed system. Furthermore, some of the sections are not very specific for the proposed application of the tool (section 2.3 could be used in nearly any paper that describes the use of Docker). The same holds to a large extent for sections 2.2 and 2.4.

Author reply: We believe that this Referee comment is addressed by the responses above.

Changes in the manuscript: Please see changes made in response to Referee comment 1.

- 5. To summarize my main concern: the paper misses a clear problem statement and as a result it is unclear why the presented software development would be the answer. The paper would be worth reconsidering for publication if the authors would be able to reformulate the paper in such a way that:
  - *there is a clear problem statement (which may, or may not be related directly to the NEON network);*
  - *it is clear that the DevOps methodology is needed to tackle that problem (including a clear general introduction to DevOps, irrespective of the tools used);*
  - that this set of proposed tools and methods would enable a DevOps methodology for the task at hand (EC-data processing);
  - the presented cases (section 3) clearly illustrate why a software infrastructure as proposed in the paper is needed.

Author reply: We believe that our responses to Referee comments, especially comment 1, and the associated changes made to the manuscript, address these concerns.

Changes in the manuscript: Please see changes made in response to Referee comment 1.

Below I will provide detailed commentsNote: in the comments below, the comment is preceded by the line number.2 Detailed comments

1. 56: I do not see why the ultimate goal should be a universal EC processing environment. As long as the software that is used in papers is described in open literature and the source code is openly available, readers will be able to assess the results of the EC-processing used in the described research. Research groups will always have reasons to do things their own way: because of instrument or site specifics, or due to specific requirements in output and analysis. And from software intercomparisons performed in the past it is quite clear in which aspects the main differences between different processing packages occur (e.g. Mauder et al., 2008; Fratini and Mauder, 2014).

Author reply: By universal processing environment, we mean the capability of processing software to be portable, reproducible, and extensible such that it can be reliably and consistently applied at scale across most, if not all, computer platforms. This would better allow research groups to tailor existing software to their needs instead of re-creating code or kludging together multiple software outputs to realize an algorithmic chain for data analytics. Even though source code is available, it does not guarantee reproducibility: The subsequent sentence in the manuscript (previously, line 58) points out that 50% of published scientific code cannot even be run due to missing software dependencies. We adjusted the text to better clarify this point.

The Referee notes:" As long as the software that is used in papers is described in open literature and the source code is openly available". Our claim is that this is generally not the case, and we argue that the DevOps framework helps groups achieve that aim and demonstrate such with our work with eddy4R. Even if source code is provided, code is not sufficiently updated to reflect e.g. new developments in processing or firmware bugs. End-to-end processing from raw data to fluxes is not commonly available except in a few softwares, usually written in compiled programming languages that are not easily modifiable by a general user. Further, it is not easy to modify them for site-specific conditions, and finally, they are rarely inter-compared.

Changes in the manuscript: The passage in question now reads: "Still, large differences in instrumentation, site setup, data format, and operating system stymie the adoption of a universal EC processing environment: one that is portable, reproducible, and extensible to allow tailored workflows that incorporate additional data streams, to automate and scale processing across large compute facilities, or to inject additional algorithms that address specific needs or synergistic research questions. In 50% of published scientific code, one cannot even replicate the necessary software dependencies (Collberg et al., 2014), and even widely used and well-documented EC processing software packages have shown substantial inconsistencies in flux estimates (e.g. Fratini and Mauder, 2014). A universal EC processing software to their needs (and contribute new algorithms) instead of re-creating code or kludging together multiple software outputs to realize an algorithmic chain for data analytics."

References:

Collberg, C., Proebsting, T., Moraila, G., Shankaran, A., Shi, Z., and Warren, A. M.: Measuring reproducibility in computer systems research, University of Arizona, Department of Computer Science, Tucson, USA, 37, 2014.

Fratini, G., and Mauder, M.: Towards a consistent eddy-covariance processing: An intercomparison of EddyPro and TK3, Atmos. Meas. Tech., 7, 2273-2281, doi:10.5194/amt-7-2273-2014, 2014.

2. 71: it is unclear why DevOps is being embraced: why is this methodology so suited for the problem at hand (the problem is not clearly stated, but I assume that the 'strong need' expressed in lines 66-70 is 'the problem')?

Author reply: In response to the points above we clarified the problem statement and provided a general overview of DevOps that links the NEON software framework to DevOps methodology. We believe this has clarified why DevOps is suited to address the problem statement.

Changes in the manuscript: Please see changes made in response to major Referee comment 1.

3. 83: indeed, I agree that the fact that the proposed work enables an exact reconstruction of the system that was used to construct certain derived data is an important asset (having the data and the software is -in some cases- insufficient to enable exact reproducibility). On the other hand, in the application at hand, bit-wise identical results are usually not needed: the statistical errors in the derived quantities are usually orders of magnitude larger than the differences that occur do to differences in details of the computational environment.

Author reply: Our point was that the reproduction of the computational environment, namely the data and software dependencies required to run the software successfully, is the most important. Given that we do not mention bit-wise identical results and the preceding sentence states "The recipe automates the loading of the software including all dependencies so that the most significant hurdle of reproducing the computational environment is overcome.", we do not think the manuscript needs further clarification of this point.

Changes in the manuscript: No changes.

4. 96-106: this 'cycle' suggests some regularity and pace of development. Is that what is intended, or does it simply mean that once someone has a suggestion for a new feature or new implementation, the code has to go through the cycle? My own experience with code development is that many users are using very different versions of the code (from very old to cutting edge) which may yield impractical surprises when these different users supply new code. The advantage of the -apparently well-funded- NEON network

is that paid personal is available to oversee new code submissions. In this section it is not clearly defined who/what is the 'science community' on one hand, and 'NEON science' on the other hand.

Author reply: The subsequent sentence (previously lines 106-108) described the pace of iteration "This DevOps cycle can be repeated for continuous development and integration of requests and future methodological improvements, resulting in the next release." Meaning, the cycle begins anew when new code features are submitted or desired by the scientific community. As this is already explained in the manuscript, we do not think further clarification is needed.

Regarding new code submissions arising from out-of-date versions: The advantage of the NEON-DevOps methodology is that users stay up-to-date with the code-base by 'forking' the repository in GitHub (previously lines 184-186). This avoids the issue of users modifying "offline" versions of the code that easily become out-of-date. As this is already described in the manuscript, we do not think further clarification is needed.

We agree that clarification of 'NEON science' vs. the 'science community' may be helpful to readers.

Changes in the manuscript: No changes were made with regard to the cycle of development and staying up-to-date with the code base.

We added the following text to Sect. 2 to clarify 'NEON Science' vs. the 'Science Community': "Here, we define NEON Science as personnel working directly on the NEON project, and the Science Community as anyone producing or using data, algorithms, or research products related to NEON data themes (Atmosphere; Biogeochemistry; Ecohydrology; Land Cover and Processes; Organisms, Populations, and Communities), regardless of whether they also work on the NEON project."

5. As explained in my main comment: section 2.1 seems to mainly discuss 'just another EC-processing tool'. Please focus on those aspects that are new and make it particularly suited for NEON, and for use in the DevOps methodology. My impression is that the direct implementation of the option for parallel processing of EC-data is one important aspect (which may not be relevant if a group only operates a limited number of towers, but which may be relevant if in 10 years time NEON decides to reprocess all EC-data of all stations according to the latest insights). But if this parallelization is so central, I would like to see some more explanation/example/tests of it (performance, scalability). Another aspect would be indeed the modularity if it allows -as advertised-for the easy adaptation of new hardware. I suppose that you use some sort of instrument abstraction which makes it possible to describe the properties of any thinkable instrument in such a way that the data can be ingested and processed in a correct way. Again, more details on this potentially unique aspect would make the paper worthwhile to read.

Author reply: As detailed above, the DevOps model has been centralized in the paper to demonstrate the novelty of this approach to scientific computation in general, and to EC specifically. An executable example workflow is provided as part of the revised manuscript in order to demonstrate some of the novel aspects of the eddy4R software itself. The example workflow covers read-in and write-out of self-describing HDF5 files, the modular and nested application of data processing models, as well as interactive visualization capabilities.

Changes in the manuscript: We have added additional context to the HDF5 Sect. 2.4, please see our reply to Referee comment 13 below. Furthermore, we have adjusted the installation and operation Sect. 2.6 to incorporate the executable example workflow.

6. 168: indeed git is open source and free, but Github is not (unless you use a public repository). That brings me to the question in which way eddy4R will be open-sourced. Will the github repository be opened up for everyone (by now I cannot find it)? Or will you publish certain stable versions to the public and keep the development closed? Reliance on Github may seem a risk when planning for the processing of data from a project that will run for 30 years. But since git repositories are stored locally as well, no risk exists in case Github would go out of business.

Author reply: The current, stable snapshot of the GitHub repository is published and archived incl. DOI (links provided in Sect. 5 "Code and data availability"). As NEON Construction completes, the GitHub repository itself will be made publicly accessible. The central component with regard to contingency planning is distributed version control, of which Git is one implementation. Should Git fail in the next 30 years, it can be replaced with another version control system.

Changes in the manuscript: We added the following text to the end of Sect. 2.4: "We note that the DevOps workflow is robust to the business viability of the particular tools used for implementation. Git is simply one instance of a version control system, which could be replaced with another similar tool should Git fail at some point in the future."

7. 181: who will do the review and testing? Do you have full-time staff for that? Will the testing be automated in the sense that when the new code is included, all prior test cases should still run without errors, while the new code should also provide it's own test case (if it implements a new feature)?

Author reply: As facilitator of the DevOps implementation of eddy4R, NEON provides the software change control board necessary for continuous development and integration. Testing is automated, and when new code is included, all prior test cases must run without errors and reproduce benchmark results. Including a test case for new code is strongly encouraged, but not mandatory.

Changes in the manuscript: Changed accordingly in Sect. 2.2.: "After (3) 'committing' or creating a new feature, the developer (4) can propose the feature for inclusion in the

official eddy4R source code by issuing a pull request to (5) NEON's change control board. After (6) thorough review and all prior test cases reproducing benchmark results (DevOps: Verify stage), the feature can be 'merged' or integrated into the next release of (1) the official, stable eddy4R source code (DevOps: Package stage). This cycle can be repeated to accommodate requests and future developments, resulting in subsequent releases. Including a test case for new code is strongly encouraged to ensure sustainability, but is not mandatory."

8. 198: The phrase 'minimal context' suggests that Dockers are small. However, in my experience Docker files are usually quite bulky (as they contain a complete operating system + the software needed to run the scripts). Although storage is not an issue nowadays, the wording suggests something different than what readers might expect.

Author reply: From https://www.docker.com/what-docker: "Using containers, everything required to make a piece of software run is packaged into isolated containers. Unlike VMs, containers do not bundle a full operating system - only libraries and settings required to make the software work are needed. This makes for efficient, lightweight, self-contained systems...". In consequence, the eddy4R Docker image is approximately one order of magnitude smaller compared to a virtual machine with comparable functionality.

Changes in the manuscript: Clarified in Sect. 2.3: "Compared to the similar but more cumbersome virtual machine approach, a Docker image is an order or magnitude smaller (eddy4R-Docker: 2 GB without example data). Also, by running as native processes it bypasses the virtual machine overhead."

9. 211: it is unclear to what extent hub.docker.com provides versioning (in the sense that I would be able to exactly reproduce a docker that was produced 5 years ago (with the same Debian version and the same R version with the same package versions). In lines 324 and further it becomes clear that indeed you can specify a version of the docker.

Author reply: As mentioned by the Referee, versioning of Docker is addressed in Sect. 2.6: "The release version of the Docker image can be specified, or alternatively the specifier latest provides the most up-to-date development image." In addition, it is even possible to pull and run a specific digest using the docker run stefanmet/eddy4r@sha256: command.

Changes in the manuscript: Added early reference to versioning in Sect. 2.3: "Using e.g. a cloud hosting platform like DockerHub (https://hub.docker.com/), the image build, versioning and distribution can be automated." And added in Sect. 2.6: "In addition, it is possible to pull and run a specific digest using the docker run stefanmet/eddy4r@sha256: command." 10. I do understand that a docker image provides a machine that can do the data processing in a way that is independent of the underlying hardware and software (OS). But you do not specify how this machine (which I still would consider 'virtual') talks to the local file system (outside the docker) to read the raw EC data and write the results. Or is the data flow always supposed to pass through the NEON databases (which are accessed over a network connection).

Author reply: The Docker container can be mounted to a local file system when initially calling the docker run command. We have updated to the manuscript to clarify the ability to mount a local file system.

Changes in the manuscript: In Sect. 2.6 (Installation and operation) we have added: "If data is not directed from/to cloud hosting, a physical file system location on the host computer (local/dir) can be mounted to a virtual file system location inside the Docker container (docker/dir). This is achieved with the docker run option -v local/dir:docker/dir."

11. 249: does 'uncompressed' mean 'ASCII', or does it mean 4 or 8 byte binary real values. In the first case a 1:10 compression ratio is not unexpected, in the second case I would be surprised.

Author reply: The data was saved in comma-delimited or .csv files in ASCII format. Changes in the manuscript: In Sect. 2.4 we added for clarity: "For the tower datasets analyzed in this study, including sonic anemometer, infrared gas analyzer and mass flow controller data, file sizes ranged from 1 GB for the uncompressed data in commadelimited ASCII files to 0.1 - 0.2 GB in HDF5 format, depending on the amount of missing data."

12. 251: I am surprised that only the variable name and units are added as meta data. Since an important reason for the proposed methodology is reproducibility and traceability, I would expect that also metadata on the processing itself (software versions, tool chain, processing configurations) would be included. In that way a file with processed EC-data, when 'found in the wild' would still tell the story of how it was produced.

Author reply: These software configuration metadata are also included in the HDF5 files. In Sect. 2.5 they are referred to as processing parameters and described in more detail "This includes EC raw data (Level 0, or L0 data) alongside contextual information on measurement site (ParaSite), environment (ParaEnv), sensor (ParaSens), calibration (ParaCal), as well as processing parameters (ParaProc)."

Changes in the manuscript: We have added mention of this additional metadata, now already in the HDF5 Sect. 2.4 of the manuscript: "Additional metadata are attributed

to various hierarchical groups throughout the file, including environmental parameters, sensor metadata, and processing parameters."

13. 253: I would say that text-based data storage of raw EC-data is already something of the distant past. NetCDF has gained significant usage since 15 years. When properly used, NetCDF files are self-descriptive as well (the same reservation would hold for HDF files: the meta data have to be filled). Otherwise many people use the binary file formats produced by their data logging infrastructure. So again, please indicate the real advantages of the HDF format as compared to others, vis-a-vis the requirements of your application, in combination with the DevOps framework. I could think of two aspects: compression of data (which is not possible in NetCDF) and the hierarchical data structure.

Author reply: While the use of NetCDF has gained in popularity for some use cases, it does not appear to be prevalent throughout the community. On the other hand, storing data in binary formats necessitates converting the data before interacting with it for data analysis, which can be very cumbersome. We agree that the data compression, hierarchical data and metadata structure are important proponents for the choice of HDF5. Both, NetCDF-4 and HDF5 build on the HDF data model and are interoperable.

Changes in the manuscript: We have restructured Sect. 2.4 and added several sentences to make this clearer:

"The capability to process large data sets is reliant upon efficient input and output of data, data compressibility to reduce compute resource loads, and the ability to easily package and access metadata. The Hierarchical Data Format (HDF5) is a file format that can meet these needs, and is a key tool aiding the DevOps: Configure (computational resource allocation) stage. A NEON standard HDF5 file structure and metadata attributes allow users to explore larger data sets in an intuitive "directory-like" structure that is based upon the NEON data product naming convention (see Figure 4). Group level 1 separates data by site and site level metadata are attributed at that level. Group level 2 separates data by data product level (DPL) and DPL metadata are attributed at that level, where DPLs correspond to the amount of processing performed. DPL1 are calibrated descriptive statistics, DPL2 are temporally interpolated, DPL3 are spatially interpolated, and DPL4 are further-derived quantities. Group level 3 are the individual data products, for instance CO2 concentration. Lastly, replicates in the horizontal and vertical are separated as individual data tables.

This provides a streamlined data-delivery mechanism for the eddy4R-Docker processing framework. For the tower datasets analyzed in this study, including sonic anemometer, infrared gas analyzer and mass flow controller data, file sizes ranged from 1 GB for the uncompressed data in comma-delimited ASCII files to 0.1 - 0.2 GB in HDF5 format, depending on the amount of missing data. The HDF5 files can be written in a simple format where data are stored

as single 1-dimensional arrays to maximize compression and efficiency, or the data can be stored as compound datatables that allow multiple datatypes to be written together in columnar format for ease of navigation when data size is not an issue.

Another important function of the HDF5 file format is the ability to attach metadata as attributes, further promoting reproducibility. The data in this study has the units and variable names as metadata attached to the data tables in the HDF5 file. Additional metadata are attributed to various hierarchical groups throughout the file, including environmental parameters, sensor metadata, and computational workflow parameters. As a result, HDF5 and similar self-documenting hierarchical data formats are gaining traction in a community that has traditionally relied on ASCII text column or comma-delimited files, especially as tools for viewing, manipulating, and extracting data from HDF5 become more commonplace. The utility of HDF5 file format is demonstrated in the executable example workflow that accompanies this manuscript (see Sects. 2.6, 5)."

14. 258: in the figure it is unclear if this depicts one file, or multiple files. Furthermore, the terminology ('group', 'Level 1,2,3', 'DPL', ....) is not explained. It is insufficient to keep the reader in the dark and just refer to the 'NEON data product naming convention'. To summarize: I do not understand what I am looking at.

Author reply: Agreed. We have updated the manuscript to give a better description of the file structure.

Changes in the manuscript: Please see our reply to Referee comment 13 (above).

15. 268 and related text: in principle, the configuration shown here is not much different from the scripted use of a compiled program (or interpreted script): what is done by the docker images is described in the parameter files. What may be interesting is that the first (left-most) docker-image spawns the second (right-most) images, which are identical in implementation, but serve a different purpose be- cause they are instructed to do so (either the 'turbulence' task or the 'storage' task). Another important asset of the workflow is that apparently meta-data on the processing is included in the HDF files (although line 251 suggests otherwise). It would be of interest to give more details about that (how self-contained are the HDF-files?).

Author reply: Many thanks for pointing out the one-to-many spawning property. Regarding the inclusion of metadata, please see our response to Referee comment 12.

Changes in the manuscript: We added in Sect. 2.5: "It should be noted that the "turbulence", "storage" and "combined" Docker containers (Figure 5 right) are all spawned from the same eddy4R Docker image (Figure 5 center): each container

includes the same underlying functionality (eddy4R packages), but serves a different purpose by being fed the "turbulence", "storage" or "combined" workflow files."

Regarding the inclusion of metadata, please see our response to Referee comment 13.

16. 283: I wonder why apparently a single day is used as the unit of storage when it comes to storage of L1-L4 data. In many applications I would think that longer time series are needed. Or does the data-portal glue these daily datafiles together when the datarequest to the portal asks for e.g. a full year of data?

Author reply: One day was chosen as a simply digestible and scalable unit for data processing. The NEON data portal also provides monthly concatenated files.

Changes in the manuscript: Added in Sect. 2.5: "In addition to the daily output files, monthly concatenated files are also available for download from the NEON data portal (https://w3id.org/smetzger/Metzger-et-al\_2017\_eddy4R-Docker/portal/0.2.0)."

17. 284: As it is shown here, the workflow is tightly integrated with the data portal. It seems to be the only use case. But what if a given researcher wants to (re-)process her/his data on her/his desktop for private use only? Is that possible as well? And how would that alter the use of the presented infrastructure?

**Author reply:**

The Sect. 2.5 in question addresses "Modular compatibility with existing compute infrastructure", of which the NEON application is one example. This integration is optional, and each user can utilize eddy4R-Docker for their own purposes and in their own configuration. To highlight this aspect of extensibility, we created an executable example workflow accompanying the revised manuscript. It utilizes the functionality of both R-packages presented here, eddy4R.base and eddy4R.qaqc, and contains a user-extensible data read-in, processing and plotting workflow.

Changes in the manuscript: Sect. 2.6 now introduces the executable example workflow. Therein we demonstrate the utility of the HDF5 file format incl. example HDF5 input and output files that are already pre-compiled into the Docker image. The following paragraph was added to Sect. 2.6:

"To demonstrate some basic capabilities and provide a template for potential eddy4R-Docker users, an executable example workflow and data are included in the eddy4R-Docker image. Once the eddy4R container is started, the example workflow, input data (NEON dp0p HDF5 file) and output data (NEON dp01 HDF5 file) are available from the Docker-internal directory example workflow is located /home/eddy/. The at /home/eddy/flowExmp/flow.turb.tow.neon.exmp.dp01.R, and provides a selection of the processing steps that yield the EC dp01 data on the NEON data portal (https://w3id.org/smetzger/Metzger-et-al 2017 eddy4R-

Docker/portal/0.2.0). The example workflow is fully documented to guide readers through the various processing steps. These include data and metadata import from the input HDF5 file, data assignment to file-backed objects, processing of 1 minute and 30 minute data statistics and data quality, and writing the output HDF5 file. In addition, outputs from the quality flag and quality metric model are visualized."

18. 294-297: this is the first point where I clearly see a reference to DevOps that links the presented software infrastructure to the advantages that DevOps could provide.

Author reply: We thank the Referee for providing an example of how to link the DevOps methodology and associated advantages to the presented software infrastructure. As addressed in other Referee comments, we added text throughout the manuscript that provides an overview of DevOps and how it forms the basis for our software development model.

Changes in the manuscript: Changes regarding this topic have been addressed in previous replies.

19. 300: again, this figure contains a number of unexplained acronyms and terms. Furthermore, I wonder what the added value of this figure is for the paper as a whole. Finally, a part of the text is very hard to read.

Author reply: The added value of this figure is to demonstrate the modular compatibility of the eddy4R-Docker framework with existing compute infrastructure, which is the focus of Sect. 2.5. We double-checked, and all acronyms are explained in the figure itself. The figure resolution appears sufficient for all text to be clearly legible.

Changes in the manuscript: No changes.

20. 303: section 2.6 reads as a user manual, rather than the description of the logic of operation. To me it is unclear why you would need to login to the docker machine (unless you want to develop and test new R-code. On the other hand, figure 5 suggests that the docker-images can be instructed 'from the outside' using parameter files.

Author reply: The GMD instructions for "model description papers" require a "user manual"-like component.

The eddy4R Docker image can be used from the command line to perform batch processing. Alternatively, the RStudio interactive development environment (IDE) can be used for code development through accessing a running eddy4R Docker container via a web browser. To clarify access to the Docker container we prepared a executable example of a (simple) data read-in, processing and plotting workflow accompanying the revised manuscript. Readers can interactively run this example workflow, which

utilizes the functionality of both R-packages presented here, eddy4R.base and eddy4R.qaqc.

Changes in the manuscript:

We addressed the scaled deployment from command line options in Sect. 2.6 and linked it to the DevOps cycle:

"The eddy4R Docker image can be used for code development (DevOps: Create stage) through accessing a running eddy4R Docker container via a web browser, as described above. Alternatively, the eddy4R Docker image can be used from the command line to perform scaled batch processing (DevOps: Configure & Monitor stages). Deployment from the command line consists of passing the R workflow file to the Docker container being deployed. This is achieved by using the docker run Command with the additional argument Rscript docker/dir/filename.R. Thus, the eddy4R Docker image can be used to simultaneously deploy multiple Docker containers to process data for multiple days or sites to the capacity of the computational platform."

In addition, we introduce the executable example workflow in Sect. 2.6. Please see our reply to Referee comment 17 for details.

21. 41 and further: to me it is unclear what the function of the three case studies is. In the introductory paragraph details on the computing infrastructure is given. This suggests that some benchmarking in terms of performance will be done. However, this only makes sense to me if there is a standard against which the new setup can be compared (e.g. current practice of reading ASCII files on a single processor machine?).

Author reply: The purpose of the test applications in Sect. 3 is to demonstrate the applicability and usefulness of the DevOps software development model to various EC use cases. Software benchmarking is one of several aspects of such evaluation, and absolute performance results are provided in Sect. 3.1.1. As explained in more detail in our response to Referee comment 34, a direct performance intercomparison from these results is not meaningful, as hardware and software process implementation differed among and within test applications.

Changes in the manuscript: Added clarification in Sect. 3 "In the following we present three test applications of eddy4R-Docker to evaluate whether the NEON DevOps model can indeed produce collaborative, portable, reproducible, and extensible eddy covariance software."

In addition, throughout Sect. 3 we now use our results to evaluate the desired DevOps software properties of portability, reproducibility and extensibility.

22. 349 and further: if indeed the performance of the software (in terms of speed, and perhaps also ease of use, flexibility . . . ) is the objective of section 3, I do not see why so much attention needs to be paid to the experimental setup (including photographs!)

as well as the interpretation of the results (section 3.1.2). Any EC dataset with a length of 1 to 2 weeks would have sufficed.

Author reply: As explained in our response to Referee comment 21, software performance is one of several aspects to evaluate the usefulness of the DevOps software development model for EC applications. More importantly, the modular applicability of eddy4R-Docker to substantially different site setups, measurement platforms and even end-to-end scientific hypothesis testing sets it apart from existing software solutions. As such, clearly identifying the differences in setups is an important component of demonstrating the value of the DevOps approach. Moreover, Referee #3 has inquired additional detail, and we feel that the current presentation strikes a balance between these viewpoints.

Changes in the manuscript: Please see our response to Referee comment 21.

23. 381: what kind of additional complexity? It could be that the presented processing infrastructure makes it easy to logically define different processing levels (L1 to L4). In that case it would be of interest to the reader to clarify which are the differences between the various processing levels, and how, conceptually, they can be defined and configured with the current setup. Perhaps this is an important added flexibility? In line 400 it becomes at least clear what is the distinction between L1 and L4.

Author reply: Complexity to test assumptions of the EC theory. The full sentence reads "The algorithmic processing for the L4 flux calculations requires additional scientific and procedural complexity to test assumptions of the EC theory."

Changes in the manuscript: For additional definition of the processing levels, please see our reply to Referee comment 13.

24. 386: what is the difference between a simple data format and a compound data format. Since the use of the HDF5 file format is such an important aspect, I would expect a clearer description of the way in which the added possibilities of HDF5 files are used (which meta-data, mixes of L1, L2... data....).

Author reply: We are using the simple HDF5 format to produce 1-dimensional arrays, each with a single datatype. In contrast, HDF5 compound datatables allow multiple datatypes to be written together in columnar format.

Changes in the manuscript: In Sect. 2.4 we added information to describe this functionality, please see our response to Referee comment 13 above.

25. 389-390: does this mean that the calibration step (to go from L0 to L0p) takes 4.8 PU minutes? Compared to the actual processing this seems long. But perhaps there is a good explanation for it. Please provide the reader with one (or perhaps my interpretation of the numbers is incorrect).

Author reply: In the processing from L0 to L0p the plausibility tests per Taylor and Loescher (2013) are performed. Of these, the step-test is the main resource consument.

Changes in the manuscript: Added in Sect. 3.1.1 "This difference arises mainly from application of plausibility tests per Taylor and Loescher (2013) in the transition from L0 to L0p."

References:

Taylor, J. R., and Loescher, H. L.: Automated quality control methods for sensor data: A novel observatory approach, Biogeosciences, 10, 4957-4971, doi:10.5194/bg-10-4957-2013, 2013.

26. 399 and further: only include these results if they show something that only this software can do, and other EC-processing methodologies cannot.

Author reply: To our knowledge, no other software methodology currently permits modularly producing results for the tower and aircraft test applications. Please also see our reply to Referee comment 21.

Changes in the manuscript: No changes.

27. 423: the fact that the presented methodology apparently is equally well able to deal with aircraft data as it is can deal with tower data is an interesting added value, worth advertising! However, I would have appreciated it if the authors would have clearly shown which parts of the processing would work the same for tower and aircraft data, and which steps would be different (e.g. by referring to figure 5). Again, the details of the particular data set (conditions of collection) is less important than the type of data and instruments involved.

Author reply: In contrast to existing flux processors, eddy4R-Docker enables any arbitrary data analysis. This is accomplished by spawning a Docker container with user-defined workflow files. These R-files define not only the parameter settings for the various processing steps and models, but the presence, absence and sequence of the individual processing steps themselves. In comparison to existing methodologies, the underlying eddy4R definition and wrapper functions are pre-compiled into the eddy4R-Docker image, and modularly shared and identical among the various applications. As such, each container spawned from the eddy4R-Docker image includes the same underlying functionality (eddy4R packages), but can serve a different purpose by being fed a different workflow file. Please also see our reply to Referee comment 15. The functions contained in the eddy4R.base and eddy4R.qaqc packages are published here, alongside an executable example workflow. How these functions are used together depends on the specific application and is user-defined. The example workflow

accompanying the revised manuscript serves as a template for user-specified implementations such as tower, aircraft, and various other potential use cases.

For the description of the datasets and site conditions, please see our reply to Referee comment 22.

Changes in the manuscript: We now introduce the executable example workflow in Sect. 2.6. Please see our reply to Referee comment 17 for details.

28. 442: it is unclear whether the  $100Hz \rightarrow 20Hz$  reduction is done by the same software or was part of the preparation of the L0 data (i.e. from L(-1) to L0.

Author reply: These steps were performed in pre-processing, and were not part of the eddy4R processing.

Changes in the manuscript: Added clarification in Sect. 3.2 "These steps were performed prior to import into eddy4R processing, but could equally well be performed therein."

29. 445: it is unclear whether the software is able to perform all the corrections that are related to the accelerations of the aircraft (the wind speeds derived from the pressure readings are not the wind speeds). I could imagine that this is part of the  $L0 \rightarrow L0p$  conversion, but this is not specified.

Author reply: In the presented case, derivation of the 3-D wind vector was part of the  $L0 \rightarrow L0p$  processing prior to ingest into eddy4R. Functionality to convert pressure and inertial motion readings to wind speeds exists in eddy4R (e.g., Metzger et al, 2011), but are not part of this release.

Changes in the manuscript: Clarified in Sect. 3.2 "The input data used in this study included the pre-derived 3-D wind vector from 5-hole probe and aircraft position, velocity and attitude."

References:

Metzger, S., Junkermann, W., Butterbach-Bahl, K., Schmid, H. P., and Foken, T.: Measuring the 3-D wind vector with a weight-shift microlight aircraft, Atmos. Meas. Tech. Discuss., 4, 1303-1370, doi:10.5194/amtd-4-1303-2011, 2011.

30. 449: it was earlier suggested that the workflow is based on HDF5 files, whereas here gzipped ASCII files are used. I understand that R is able to read all kinds of files, but is reading the ASCII files a standard feature of the methodology or did you first convert the gzipped ASCII files to HDF5 (to produce real L0 data) before ingesting them into the processing software?

Author reply: The data ingest mechanism into eddy4R is user-definable. For the aircraft test application in Sect. 3.2, gzipped ASCII files were directly imported (without prior conversion to HDF5). In contrast, for NEON's test application in Sect. 3.1 HDF5 files are used. While HDF5 is conceptually preferable because self-describing and read/write efficient, the usability of various file formats highlights the modularity of eddy4R. As Sect. 3.2.1 explains "…standard R capabilities for data ingest can be used to read data in various formats, frequencies and units."

Changes in the manuscript: No changes.

31. 451-460: This is an interesting, perhaps non-standard, chain of processing steps. For the reader this would be an interesting example to see how easily this can be configured in your software. Is it just a matter of setting a number of flags in your configuration files? How easily can you change the order of various steps that are applied to high rate data or reduced data?

Author reply: Please see our reply to Referee comment 27.

Changes in the manuscript: Please see our reply to Referee comment 27.

32. Section 3.2.2: again (like for 3.1.2): presentation of the results is only relevant if your methodology can do something that existing software cannot (or if yours can do it better/faster/easier/more insightful). For instance, if figure 11 would be output that can be produced automatically it could be of added value. In that respect it would also be interesting to know if the software could be used in a scenario where certain processing has already been done, and the user decides that he/she wants additional output. Would the software be able to figure out which data (L1...L4) is already there, and which is the minimum set of additional processing steps needed to produce the additional output. Again, in that sense it would be worthwhile for the reader (and potential user) to know which output your software can produce, based on which type of input data (type as in L0, L0p, etc). A well-organized table would be more informative than three case studies. Similarly a clear definition of which types of input data can be ingested would be helpful.

Author reply: Please see our replies to Referee comments 21, 22 and 27. One key attribute of the eddy4R-Docker methodology is its user-extensibility per requirements of the desired application. As such, no default inputs exist. To demonstrate the complementarity of eddy4R-provided functions and user-supplied workflow files, input data is instead defined individually for each test case in Sects. 3.1, 3.2 and 3.3.

Changes in the manuscript: Please see our replies to Referee comments 21 and 27.

*33.* 499: this is an interesting application! Does the script automatically select the EddyPro settings that would match the eddy4R settings? If so, providing the difference statistics

with your L1-L4 output in the HDF5 file would be useful added value, since users of your L1-L4 data would have an indication how sensitive the resulting aggregated data are to details of the processing (he/she does not know who is wrong and who is right, but a sense of the sensitivity can be helpful).

Author reply: The comparison between EddyPro and eddy4R outputs is a test application and not done automatically. The results have been calculated based on the same input dataset and processing settings, but by separate institutions and on different computer platforms. Please see our reply to Referee comment 34 for details.

Changes in the manuscript: No changes.

34. 519: Do you run EddyPro in the same docker as the R-framework? It would be interesting to know what the performance difference (if any) is between EddyPro and your R-based software.

**Author reply:**

EddyPro was run natively in Windows. The calculations were performed independently at LI-COR (EddyPro) and U Wisconsin (eddy4R-Docker), with identical settings and based on the same input dataset as specified in Sect. 3.3. Absolute benchmarking is possible and summarized in below table. However, a direct performance intercomparison from these results is not meaningful, as both, the hardware and the software process implementation differed substantially: EddyPro used two independent processes and required 33 s to process one day of data on a laptop with a solid-state drive. On the other hand, eddy4R was run as a single process and required 68 s to analyze one day of data on a server with 100 Mbit interconnect speed to a RAID Level 5 data share. Put against a rapid-access solid-state drive, the interconnect speed alongside the single/dual process implementation is likely dominating the difference in processing times.

| Metric             | EddyPro                 | eddy4R                           |
|--------------------|-------------------------|----------------------------------|
| File size per day  | 182 MB in ASCII         | 182 MB in ASCII format           |
| and format         | format                  |                                  |
| Utilized processor | 2 x Intel (R) Core (TM) | 1 x Intel(R) Xeon(R) CPU E5-2650 |
| type and number    | i7-4600U CPU            | v2 @ 2.60GHz                     |
|                    | @2.1GHz                 |                                  |
| Available memory   | 16 GB                   | 200 GB                           |
| Storage type and   | 512 GB SSD              | 40 TB in RAID Level 5            |
| size               |                         | configuration                    |
| Interconnect speed | NA                      | 100 Mbit/s                       |
| (if server)        |                         |                                  |
| Operating system   | Windows 10 enterprise   | Linux 2.6.32-                    |
|                    | 64-bit                  | 642.1.1.el6.centos.plus.x86_64   |

| CPU time per day | 33 s   | 68 s   |
|------------------|--------|--------|
| of data          |        |        |
| Maximum memory   | 120 MB | 200 MB |
| use              |        |        |

Changes in the manuscript: No changes, as a direct performance intercomparison from the benchmarking results is not meaningful.

35. 550-560: it is OK for me that in the paper you present the workflow as it is envisaged. But it would be good to already clarify in the main text which steps have already been implemented and which are underway (for the eddy4R part you clearly made this distinction, but for the other parts not). Furthermore, it remains unclear to me to what extent the workflow you propose is strongly tied to the NEON infrastructure. Or could I, as a scientist not working within NEON, be able to adopt your methodology as well. To pose it differently: where does your methodology end and where does the NEON infrastructure start?

Author reply: The eddy4R-Docker methodology is a generalized implementation of an EC software using the DevOps model. eddy4R-Docker is thus not tied to NEON, and openly available to everyone. Rather, NEON's implementation is one example of a scaled application of tower EC processing using the eddy4R-Docker methodology (Sect. 3.1). As such, the current status of NEON's implementation does not contribute to the objective of the manuscript, to "demonstrate the applicability of the DevOps model to science code development". eddy4R-Docker can be easily adopted for various use cases and applications by directly adjusting its workflow file. Please see our detailed replies to Referee comments 15, 17 and 27.

Changes in the manuscript: Please see our replies to Referee comments 15, 17 and 27.

**3 Very detailed comments**

1. 63: please add that NEON is being developed in the US.

Author reply: We agree this detail is appropriate to include.

Changes in the manuscript: The sentence in question now reads: "The U.S.-based National Ecological Observatory Network"

2. 68: what do you mean by 'tight hardware-software integration'? And which code is 'distributed' and where?

Author reply: By 'distributed and dynamic community developed code', we mean that code is written by multiple people in multiple places. By tight integration of software

and hardware we mean that improvements in data collection (e.g. spectral characteristics) and data processing (e.g., time-frequency-based spectral correction) can be closely aligned. This yields an overall optimal performance.

Changes in the manuscript: The sentence in question now reads: "This capability is accompanied by a strong need for a flexible and scalable processing framework that can incorporate specific data streams, take advantage of close alignment of hardware and software for problem tracking and resolution, provide traceability and reproducibility of outputs, and seamlessly integrate distributed and dynamic community-developed code (written by multiple people in multiple places) within existing cyberinfrastructure.

*3. 189: I think figure 3 is superfluous. The message conveyed by the caption could be included in the text. No illustration needed.*

Author reply: We agree that Figure 3 can be removed without losing much information. Changes in the manuscript: Removed Figure 3.

4. 277: for people working in the field of micrometeorology (and particularly when involved in CO2 exchange) the terms "turbulence" and "storage" will ring a bell. But for people somewhat more remote, but interested in the proposed collection of tools, it will be less clear (in the context of a strongly IT-oriented paper "storage" might mean something completely different to the reader). For the clarity of the paper the distinction between the turbulence and storage workflows is in fact irrelevant. The message is simply that based on the same code/machinery you can do different things with the same data Unless you want to show that you can have various parallel workflows which could also interact (but then you have to clarify the potential interactions).

Author reply: Turbulent exchange and storage exchange computations are basic micrometeorological capabilities of the eddy4R family of R-packages for EC data processing, and introduced in Sect. 2.1. In the paragraph in question in Sect. 2.5 these are set in relation to each other (Fig. 5), incl. their interface through the Docker container "derived". Lastly, they are set in context once more during the outlook in Sect. 4 "Summary and conclusions".

Changes in the manuscript: Given that the terminology is central to the developed EC capability and described in text, no changes.

5. 317: I think that figure 7, apparently just a collection of screenshots, is superfluous.

**Author reply:**

The GMD instructions for "model description papers" require a "user manual"-like component, and this figure presents our user interface: it permits to show how easy the Docker approach makes preparing the development environment. We are under the impression that retaining this figure provides clarity for some readers.

Changes in the manuscript: No changes.

6. 376: it is unclear to me how the current setup would lead to 50 high-rate data streams.

Author reply: Including sensor diagnostics, these are  $\sim 10$ ,  $\sim 10$ ,  $\sim 20$  and  $\sim 10$  data streams from the sonic anemometer, attitude and motion reference system, infrared gas analyzer, and mass flow controllers and solenoids, respectively.

Changes in the manuscript: No changes

7. 385: the 0.1 to 0.2 Gb: I suppose that this refers to the input files (L0 or L0p).

Author reply: Correct.

Changes in the manuscript: Clarified: "...the L0p input file sizes ranged from 0.1 - 0.2 GB..."

8. 459: the performance is only relevant if it can be compared to something else

Author reply: Please see our reply to Referee comment 34 (above).

Changes in the manuscript: Please see our reply to Referee comment 34 (above).

9. 460: what do you mean with 'file-backed objects'?

Author reply: R's ff file-backed object (https://cran.r-project.org/package=ff) facilities.

Changes in the manuscript: Now introduced in more detail in Sect. 2.1: "In addition, eddy4R is fully parallelized and memory efficient leveraging R's snowfall parallelization (https://cran.r-project.org/package=snowfall) and ff file-backed object (https://cran.r-project.org/package=ff) facilities, respectively."

10. 565: what do you mean by defensible?

Author reply: Thanks for pointing this out.

Changes in the manuscript: Changed to "...for generating reproducible net ecosystem exchange data products..."

**References**

Mauder, M., Foken, T., Clement, R., Elbers, J. A., Eugster, W., Gr, T., ... & Kolle, O. (2008). Quality control of CarboEurope flux data–Part 2: Inter-comparison of eddy-covariance software. Biogeosciences, 5, 451-462.

---

## Author Comment (AC8) · 3 Jul 2017

**Author reply to the comments by Anonymous Referee #2 of the manuscript**

**gmd-2016-318**

**"eddy4R: A community-extensible processing, analysis and modeling framework for eddy-covariance data based on R, Git, Docker and HDF5"**

**by S. Metzger et al.**

We thank Anonymous Referee #2 for the valuable feedback, which helped to improve the manuscript. Please find below the Referee comments recited in *blue, italics font*, followed by our point-by-point replies and corresponding changes in the manuscript in black, upright font.

*Metzger et al. describe a data processing framework which is illustrated on ad- hoc examples from NEON's eddy covariance tower and airborne measurement datasets. Overall this technical concept seems potentially valuable for streamlining automation of specific data processing steps from different measurement stations but it is extremely difficult to recognize the broader scientific values in the current version of the paper as written. I must admit that I was rather disappointed to find the description of the tools to be fragmented and poorly supported by the scientific results and conclusions. The whole analysis is very descriptive and in many cases misleading as to what is possible. There is little effort to synthesize what an be actually learnt from using the tool other than what its potential applications might be in the future. Most importantly, the paper does not specify scientific goals and does not even address the scope of modeling which is what the main interest of the journal's audience is. The manuscript seems to need much work to make the results and discussion useful for the scientific community but could be worthwhile to reconsider after major clarifications. The other reviewer has already provided a useful detailed guidance how this could be achieved and I agree with her/him. I also have other concerns which hopefully can be addressed in the revision.*

Author reply: Many thanks for your summary. Please find specific replies below. As detailed in the responses to Referee #1, we have better clarified the problem statement of the paper and why the DevOps approach and specific tools used to implement are the answer. The problem statement is: "How do we collaboratively create portable, reproducible, open-source, scalable, and extensible software that improves reliability and comparability of eddy covariance data products?" We believe this clarification addresses the main concern of this Referee – that there is little of use to the scientific community presented by the description of software tools – by better communicating that the focus of the paper is not on the specific software implemented, but a model of

how the EC community can go about creating portable, reproducible, and extensible software.

As such the manuscript addresses a methodological rather than scientific question. For this reason, the GMD journal was chosen, and three tests of geoscientific applications are provided in favor of a single in-depth scientific survey. These applications serve not as scientific results, but as tests that the software is portable, reproducible, and extensible. One core component of GMD model description papers is the "…evaluation against standard benchmarks…" which is addressed in Sect. 3.3.

Changes in the manuscript: Please see responses below and also to Referee #1.

*General issues:*

*1) One fundamental issue in this paper intended for GMDD is that the work is not even connected to any model or modeling framework. The journal scope does not overlap with what paper is about or at least the connection is not made clear. Because there is no model, there is no model version – a requirement of the journal. There are only two words "model" in the whole paper, one of which is included in the last sentence of conclusions but probably in a different meaning: "We hope this framework can serve as a \*model\* for implementing community-sourced, distributed-development scientific code while combatting the deficiencies of current computational frameworks that limit accessibility, reproducibility, and extensibility."*

Author reply: The authors considered several journals before deciding where to submit our manuscript, and we came to this decision through taking into account the manuscript types requested on the Geoscientific Model Development (GMD) webpage. Specifically, we felt that our paper provides "…utility tools … such as coupling frameworks … with a geoscientific application".

In addition, as detailed in the replies to Referee #1, we clarified the problem statement of the paper: "The question we ask in this paper is: How do we collaboratively create portable, reproducible, open-source, scalable, and extensible software that improves reliability and comparability of eddy covariance data products?" We then introduce the DevOps approach in more detail and how it, along with the specific tools implemented in the eddy4R-Docker development model, solves this problem. The framework provides modular processing for surface-atmosphere exchange data with quality assurance and quality control as foundation for modelling exercises such as the test application in Sect. 3.2. This includes footprint modeling (GMD: Kljun et al., 2015), evaluation of large eddy simulations (GMD: Maronga et al., 2015), machine learning etc. The result is an end-to-end framework for model building, parameterization and assessment considering the large amounts of theoretical assumptions in eddy-covariance technique that require corrections to the data. The combination of these tools to address the concern of reproducibility was a major consideration when submitting to GMD.

Per suggestion of Referee #2 as well as the executive editor, in addition to Sect. 5 Code and data availability we now include the eddy4R-Docker development model version (now: 0.2.0) also in the manuscript title.

We further clarify in the revised manuscript that eddy-covariance data processing consists of employing a sequence of model algorithms. These often originate from scientific sub-fields with corresponding publications, and eddy4R-Docker provides an integrative, yet modular and extensible framework for their concerted application and continued development. In its current form eddy4R-Docker 0.2.0 encompasses the following models: plausibility tests (Taylor and Loescher, 2013), de-spiking (Brock, 1986), lag correction, data aggregation, and QA/QC budgeting (Smith et al., 2014).

Additional models are in preparation for future extension of the eddy4R-Docker framework presented here: coordinate rotation (Wilczak et al., 2001), spectral correction (Nordbo and Katul, 2012), turbulent mixing and stationarity (Foken and Wichura, 1996), detection limit (Billesbach, 2011), turbulent sampling error (Lenschow et al., 1994), footprint analysis (Kljun et al., 2015), storage flux term, and uncertainty budgeting.

Please note that e.g. Kljun et al. (2015) is itself published in GMD.

Changes in the manuscript: We clarify the objective of the paper in the introduction: "The question we ask in this paper is: How do we collaboratively create portable, reproducible, open-source, scalable, and extensible software that improves reliability and comparability of eddy-covariance data products?"

[revised manuscript text omitted]

*2) It is not apparent how exactly this technical set of workflows adopted by NEON can be useful for a broader scientist/modeler community and what scientific problems it can solve as the idea wraps around different open-source products dedicated essentially to crunching of eddy covariance measurement data. In the abstract, it is promised that the framework is applicable beyond EC but it is completely unclear how. Maybe one way to overcome this issue would be to make a strong connection to a modeling framework where measurement and model outputs are evaluated together or elucidate aspects where this data processing framework would add to novelty and usefulness for the broader GMD community*

Author reply: This suite of packages and Docker image is meant to provide a modularly extensible flux processing platform as foundation for modeling exercises (see reply above). The presented framework is motivated by a lack of collaborative coding and processing code development in the eddy-covariance community.

To address the Referee comment and demonstrate the ease of applied modelling, we prepared an executable example workflow that accompanies the revised manuscript and executes the QA/QC model by Smith et al. (2014).

The underlying functions are already included in the eddy4R.qaqc package. In addition, we highlight the extensibility that can be achieved with the modular packaging of the eddy4R-Docker framework: the eddy4R family of packages already includes the Environmental Response Function (ERF) model for flux upscaling to the landscape (manuscript Sect. 3.2.2), scheduled for future release.

Changes in the manuscript: We now introduce the executable example workflow in Sect. 2.6. Please see our reply to Referee #1, comment 17 for details.

References:

Smith, D. E., Metzger, S., and Taylor, J. R.: A transparent and transferable framework for tracking quality information in large datasets, PLoS One, 9, e112249, doi:10.1371/journal.pone.0112249, 2014.

*3) There are no clear scientific objectives of the paper and the title does not help either "eddy4R: A community-extensible processing, analysis and modeling framework for eddy-covariance data based on R, Git, Docker and HDF5". The use of "modeling framework" is misleading (see also comment 1) because the paper fails to present any modeling or prediction which could be achieved from this framework.*

Author reply: The aim of manuscript is to introduce the novel eddy4R-Docker software framework to address a methodological rather than scientific question: the portable, reproducible and extensible processing of eddy-covariance data. For this reason, the GMD journal was chosen, and test applications to three geoscientific use cases are provided in favor of a single in-depth scientific survey.

Based on the GMD manuscript types specifications, as well as existing papers from our community (e.g., Kljun et al, 2015; Maronga et al., 2015), we are under the impression that scientific hypothesis testing is not a typical component of a GMD model / framework description paper. On the other hand a core component of GMD model description papers is "…evaluation against standard benchmarks…" which is addressed in Sect. 3.3.

Changes in the manuscript: Please see reply to Referee comment 1) (above).

References:

Kljun, N., Calanca, P., Rotach, M. W., and Schmid, H. P.: A simple two-dimensional parameterisation for Flux Footprint Prediction (FFP), Geosci. Model Dev., 8, 3695-3713, doi:10.5194/gmd-8-3695-2015, 2015.

Maronga, B., Gryschka, M., Heinze, R., Hoffmann, F., Kanani-Sühring, F., Keck, M., Ketelsen, K., Letzel, M. O., Sühring, M., and Raasch, S.: The Parallelized Large-Eddy Simulation Model (PALM) version 4.0 for atmospheric and oceanic flows: model formulation, recent developments, and future perspectives, Geosci. Model Dev., 8, 2515-2551, doi:10.5194/gmd-8-2515-2015, 2015.

*4) The story basically presents a rather ambitious idea of automating data processing including quality control. The latter is not shown yet that it already works well so the product is not yet ready to be fully useful for the community. Once QC is implemented it could be interesting to see how it is done and how flexible the options are for the user. For instance, on page 14 L32 it is concluded "Once scientific QA/QC and uncertainty budget is implemented, the computational expense will likely increase by a factor of two to three. This suggests that eddy4R performs comparably to other flux processors." As presented, the value from another EC flux processor tool is unclear in where it would really help but what is interesting is that the development is directed to a modeling audience who might also be able to use this tool if it was better explained. However, without clearly stated goals and sufficient supporting material to assess its quality and usefulness, it is difficult to evaluate the code framework for all its ambitious features. The paper is incoherent in its presentation (e.g. different components, datasets are presented separately without a clear thread creating multiple fragmented methods and results) and in many places the quality is diverging from the standards of a scientific paper.*

Author reply: We agree that the implementation of the QA/QC framework substantially adds to the novelty of the methods included in this initial release of the eddy4R software. The QA/QC framework to deal with plausibility tests on the data is now fully implemented. Additional flux QA/QC tests are still be refined to accompany the full suite of eddy4R packages that are being released with the completion of NEON Construction. We hope to address the main concern by providing an example workflow accompanying the revised manuscript, which include the Taylor and Loescher (2013) high-frequency plausibility test model alongside the Smith et al. (2014) model for consolidating the results to a final quality flag. This highlights some of the capabilities of the eddy4R.base and eddy4R.qaqc packages, and provides a user-accessible and

modifiable workflow template. Please also see our detailed QA/QC replies to Referee comment 6).

The different test applications are central to proving the flexibility of the eddy4R-Docker framework to process both tower and aircraft flux data. They demonstrate that the DevOps approach can be used in scientific software development.

Changes in the manuscript: We now introduce the executable example workflow in Sect. 2.6. Please see our reply to Referee #1, comment 17 for details.

Author reply:

We agree that Figure 3 can be removed without losing much information.

The intent of this paper has been clarified to demonstrate the applicability of the DevOps model to EC science code development (please see our reply to Referee comment 3 for details). One key attribute of the eddy4R-Docker methodology is its

user-extensibility per requirements of the desired application. As such, no default workflow or settings exist that could be easily tabulated across applications. As mentioned by the Referee, specifics differ substantially e.g. among the tower and aircraft use cases. To demonstrate the complementarity of the eddy4R-provided functions and user-supplied workflow files, the corresponding workflows and settings are thus documented individually for each test case in Sects. 3.1, 3.2 and 3.3. The test applications in Figures 9 – 13 are central evidence to the claim of adjustability and expansibility.

We agree with the Referee that tower and aircraft data require careful QA/QC and interpretation, and in no part of these test applications airborne and tower data are automatically compared.

Rather, a comparison takes place as part of the DevOps **Verify** step: reference datasets generated with EddyPro have been stored and automated tests are performed prior to new code being incorporated. The results are shown in Sect. 3.3: calculations were performed independently at LI-COR (EddyPro) and U Wisconsin, with identical settings and based on the same input dataset as specified in the manuscript. Four significant digits were specified in the plotting routine for Figure 13, and the output of any uninformative zeroes is consequently suppressed. As discussed e.g. in Mauder et. al. (2008), discrepancies exist among software implementations also for the calculation of averages and variances, which we thus show alongside their corresponding fluxes.

Changes in the manuscript: Removed Figure 3.

The results in this manuscript are shown identical to directly displaying the data downloadable from the NEON data portal, without sub-setting for quality or uncertainty. This highlights the need for the data user to determine and select the acceptable level of quality and uncertainty based on the particular use case (analogous to e.g. MODIS quality flags). During the remaining software development steps throughout NEON Construction, dedicated flux QA/QC metrics are being added to the already implemented plausibility tests. These are currently residing in the eddy4R.turb package, which is not released, and hence not applied here. Please also see our reply to Referee comment 4).

To address your last question about the temperature trend and sensible heat flux, we see the DOY 117 night was a very turbulent night with a lot of mixing resulting in less radiative cooling. This may explain some of the decorrelation.

Changes in the manuscript: In Sect. 3.1.2 we have clarified that the full flux QA/QC and uncertainty budget needs to be applied: "The spiky results preceding and following periods with >10% invalid data highlight the need for applying the full flux QA/QC and uncertainty budget to provide science-grade fluxes."

Many thanks for catching the mal-formatted unit of the $CO_2$ flux. We have corrected the figure axis label to $\mu mol\,m^{-2}\,s^{-1}$.

Additionally, please also see replies to above Referee comments for changes to the manuscript.

---

## Author Comment (AC9) · 3 Jul 2017

**Author reply to the comments by Anonymous Referee #3 of the manuscript gmd-2016-318**

**"eddy4R: A community-extensible processing, analysis and modeling framework for eddy-covariance data based on R, Git, Docker and HDF5"**

by S. Metzger et al.

We thank Anonymous Referee #3 for the valuable feedback, which helped to improve the manuscript. Please find below the Referee comments recited in *blue, italics font*, followed by our point-by-point replies and corresponding changes in the manuscript in black, upright font.

This study present a radically new way to process eddy-covariance data. It combines R-coded EC software that are wrapped in a portable Docker image that can be used on various platforms. It is meant to be scalable and to make use of parallel processing of large quantities of data.

Author reply: Many thanks for this succinct summary.

Changes in the manuscript: No changes performed.

**Major comments**

In line with the other reviewers, I think that the paper currently lacks a clear scientific question. I could image that for GMD a clear description of a software environment would suffice, but this paper seems to describe "work in progress".

Author reply: As stated by the Referee, the aim of manuscript is to introduce the novel eddy4R-Docker software development model to address a methodological rather than scientific question: the portable, reproducible and extensible processing of eddy-covariance data. For this reason, the GMD journal was chosen, and three tests of geoscientific applications are provided in favor of a single in-depth scientific survey. One core component of GMD model description papers is "…evaluation against standard benchmarks…" which is addressed in Sect. 3.3. To demonstrate completion we created an executable example workflow accompanying the revised manuscript.

In addition, as detailed in the replies to Referee #1, we clarify the problem statement of the paper: "The question we ask in this paper is: How do we collaboratively create portable, reproducible, open-source, scalable, and extensible software that improves reliability and comparability of eddy covariance data products?" We then introduce the DevOps approach in more detail and how it, along with the specific tools implemented in the eddy4R-Docker development model, solves this problem. In doing so, we clarify that although the specific software implemented by this developmental model is a work in progress, the developmental model itself is complete and robust, as shown by the test applications.

Changes in the manuscript: Please see the detailed responses below as well as the responses to Referee #1 for changes made in the manuscript.

I am a big fan of Docker and directly downloaded the Docker image. I was disappointed in the fact that the image did not contain clear examples (e.g. the three examples outlined in the paper). I could see that the eddy4R.base and eddy4R.qaqc packages were part of the Docker image. I think it is a missed opportunity not to provide examples of (simple) data processing and plotting. Now the advantage of Docker images remains untraceable to the readers and remains rather theoretical.

Author reply: We could not agree more with the Referee in that an application example would add much value for the reader and potential user. For this reason, we created an executable example workflow accompanying the revised manuscript. It utilizes the functionality of both R-packages presented here, eddy4R.base and eddy4R.qaqc, and contains a user-extensible data read-in, processing and plotting workflow.

Changes in the manuscript: Sect. 2.6 now introduces the executable example workflow:

"To demonstrate some basic capabilities and provide a template for potential eddy4R-Docker users, an executable example workflow and data are included in the eddy4R-Docker image. Once the eddy4R container is started, the example workflow, input data (NEON dp0p HDF5 file) and output data (NEON dp01 HDF5 file) are available from the Docker-internal directory The example workflow is located /home/eddy/. at /home/eddy/flowExmp/flow.turb.tow.neon.exmp.dp01.R, and provides a selection of the processing steps that yield the EC dp01 data on the NEON data portal (https://w3id.org/smetzger/Metzger-et-al\_2017\_eddy4R-Docker/portal/0.2.0). The example workflow is fully documented to guide readers through the various processing steps. These include data and metadata import from the input HDF5 file, data assignment to file-backed objects, processing of 1 minute and 30 minute data statistics and data quality, and writing the output HDF5 file. In addition, outputs from the quality flag and quality metric model are visualized."

For instance, the HDF5 section (2.4) is clear but a rather standard description that is available on internet (meta-data, directory structure, self-documenting). Again, this is a missed opportunity to guide users through an example (download raw data, process the data, and HDF5 output and visualization of results). You want to convince the "traditional ASCII" community.

Author reply: Agreed. The executable example workflow includes HDF5 read-in, write-out examples with attributed metadata to demonstrate the utility of having metadata attached to the data.

Changes in the manuscript: Please see our replies to the Referee comment above. Among others we demonstrate the utility of the HDF5 file format in the executable example workflow, and example HDF5 input and output files are already pre-compiled into the Docker image.

Section 2.5 presents the way NEON wants to deploy Docker images. Again, this remains rather high level, while the stated goal is to "empower the Science community at large by putting the key to the scientific algorithms into the hand of scientists". Again, a clear running example in a Docker container would convince these scientists more than a NEON brochure.

Author reply: We believe that this concern is addressed through the executable example workflow, which is described in Sect. 2.6.

Changes in the manuscript: No changes to Sect. 2.5.

Section 2.6 would be an ideal starting point for further "Docker-assisted" data analysis, but unfortunately stops at a reference to the eddy4R wiki pages.

Author reply: In response to the Referee comment, we introduce the executable example workflow incl. "Docker-assisted" data analysis in Sect. 2.6.

Changes in the manuscript: Please see our replies to the Referee comments above.

In section 5 there is a reference to the raw data, but again unfortunately no examples are given in which a Docker image automatically reads, processes, and presents results. In the remainder of the paper, three examples are given, which is basically fine, but without a traceable and "hands-on" exercise does not add much. It is (and should be) part of the standard software testing.

Author reply: We address this concern through providing the executable example workflow.

Changes in the manuscript: Please see our replies to the Referee comments above.

In summary, I very much like the concept presented in this paper. However, without more in depth possibilities for potential users of the software, the papers seems more suitable for internal documentation than convincing readers that this is a promising way for the community to process eddy covariance data.

Author reply: We thank the Referee for sharing the positive impression of the paper's concept. We agree that more in-depth possibilities for potential users of the software will help demonstrate the utility of the software development approach.

Changes in the manuscript: We have created an executable example workflow accompanying the revised manuscript.

**Minor comments**

Page 1: line 34: mention where the NEON site is and also where the aircraft data were collected.

Author reply: This information is provided as part of the test applications in Sects. 3.1 and 3.2.

Changes in the manuscript: Added "USA" for the aircraft test application in Sect. 3.2.

*Page 1, line 38: "streaming generation of science-grade EC fluxes": please explain better what this means.*

Author reply: Adjusted.

Changes in the manuscript: Changed to "...automated generation of science-grade EC fluxes..."

Page 6, line 185: current recent

Author reply: Adjusted.

Changes in the manuscript: Changed to "...most recent eddy4R source code..."

Page 6, Figure 3, introduced at line 191. This hardly adds anything. A link would do here. Also figure 4 and figure 7 seem illustrations that do not add much.

Author reply: We agree that Figure 3 can be removed without losing much information. The GMD instructions for "model description papers" require a "user manual"-like component: Figure 4 introduces the HDF5 format and structure used in this study and for NEON data portal downloads of EC data (https://w3id.org/smetzger/Metzger-etal\_2017\_eddy4R-Docker/portal/0.2.0). Figure 7 presents the development environment user interface. As both of these are fairly new to the EC community we are under the impression that retaining Figure 4 and Figure 7 provides clarity for some readers.

Changes in the manuscript: Removed Figure 3.

**Page 7, line 231: CI?**

Author reply: Cyberinfrastructure, as introduced in Sect. 2.

Changes in the manuscript: No changes.

---

## Author Comment (AC10) · 3 Jul 2017

please find final replies in supplement

Please also note the supplement to this comment:
https://www.geosci-model-dev-discuss.net/gmd-2016-318/gmd-2016-318-AC10-supplement.pdf